



# Estimation of effective porosity in large-scale groundwater models by combining particle tracking, auto-calibration and $^{14}$C dating

Rena Meyer[1], Peter Engesgaard[1], Klaus Hinsby[2], Jan A. Piotrowski[3], and Torben O. Sonnenborg[2]

[1]University of Copenhagen
[2]Geological Survey of Denmark and Greenland
[3]Aarhus University

**Correspondence:** Rena Meyer (reme@ign.ku.dk)

**Abstract.** Effective porosity plays an important role in contaminant management. However, the effective porosity is often assumed constant in space and hence either neglected or simplified in transport model calibration. Based on a calibrated highly parametrized flow model, a three-dimensional advective transport model (MODPATH) of a $1300\,km^2$-large coastal area of southern Denmark and northern Germany is presented. A detailed voxel model represents the highly heterogeneous geological composition of the area. Inverse modelling of advective transport is used to estimate seven, spatially distributed, effective porosity units based on apparent groundwater ages inferred from 11 $^{14}$C measurements in Pleistocene and Miocene aquifers, corrected for the effects of diffusion and geochemical reactions. The match between the observed and simulated ages is improved significantly by the calibration of seven porosity units by a reduction of ME of 99% and RMS of 82% compared to a uniform porosity approach. Groundwater ages range from a few hundred years in the Pleistocene to several thousand years in Miocene aquifers. The advective age distributions derived from particle tracking at each sampling well show unimodal (for younger ages) to multimodal (for older ages) shapes and thus reflect the heterogeneity that particles encounter along their travel path. The estimated effective porosity field, with values ranging between 4.3% in clay and 45% in sand formations, is used in a direct simulation of distributed mean groundwater ages. Although the absolute ages are affected by various uncertainties, a unique insight into the complex three-dimensional age distribution pattern and potential advance of young contaminated groundwater in the investigated regional aquifer system is provided, highlighting the importance of estimating effective porosity in groundwater transport modelling and the implications for groundwater quantity and quality assessment and management.

*Copyright statement.* TEXT

## 1 Introduction

The age of groundwater, i.e. the time elapsed since the water molecule entered the groundwater (Cook and Herczeg 2000; Kazemi et al. 2006) is useful (i) to infer recharge rates (e.g. Sanford et al., 2004; Wood et al., 2017) and hence to sustainably exploit groundwater resources, (ii) to evaluate contaminant migration, fate and history (Bohlke and Denver 1995; Hansen et al. 2012) and predict spread of pollutants and timescales for intrinsic remediation (Kazemi et al. 2006), (iii) to analyze aquifer



vulnerability or protection to surface-derived contaminants (e.g. Manning et al. 2005; Bethke and Johnson, 2008; Molson and Frind, 2012; Sonnenborg et al., 2016) and indicate the advance of modern contaminated groundwater (Hinsby et al. 2001a; Gleeson et al. 2015; Jasechko et al. 2017) and groundwater quality in general (Hinsby et al. 2007), and (iv) to contribute to the understanding of the flow system, e.g. in complex geological settings (Troldborg et al. 2008; Eberts et al. 2012). Three
different approaches with specific benefits and disadvantages are commonly applied to simulate groundwater age (Castro and Goblet 2005; Sanford et al. 2017). Particle-based advective groundwater age calculation utilizing travel time analysis is computationally easy, but neglects diffusion and dispersion. The full advection-dispersion transport simulation of a solute or an environmental tracer is computationally expensive and limited to the specific tracer characteristics (McCallum et al. 2015), but accounts for diffusion, dispersion and mixing. The tracer independent direct simulation of groundwater mean age (Goode
1996; Engesgaard and Molson 1998; Bethke and Johnson 2002) includes advection, diffusion and dispersion processes and yields a spatial distribution of mean ages.

A comparison of ages simulated using any of these methods with ages determined from tracer observations, referred to as apparent ages, is desirable as it can improve the uniqueness in flow model calibration and validation (Castro and Goblet 2003; Ginn et al. 2009) and it potentially informs about transport parameters such as effective porosity, diffusion and dispersion, that
are otherwise difficult to estimate. However, the approach is far from straight forward as environmental tracers undergo non-linear changes in their chemical species (McCallum et al. 2015) and groundwater models only represent a simplification and compromise on structural and/or parameter heterogeneity. In a 2D synthetic model, McCallum et al. (2014) investigated the bias of apparent ages in heterogeneous systems systematically. McCallum et al. (2015) applied correction terms, e.g. diffusion correction for radioactive tracers, on apparent ages to improve the comparability to mean advective ages. They concluded that
with increasing heterogeneity the width of the residence time distribution increases and that apparent ages would only represent mean ages if this distribution is narrow and has a small variance.

Flow and transport parameters such as hydraulic conductivity, conductance of streambeds and drains, recharge and dispersivities have gained more and more focus in calibration of groundwater models, recently also on large scales, where the combination of head, flow and tracer observations are widely used as targets (McMahon et al. 2010). However, effective porosity has
not received nearly as much attention and especially its spatial variability is often neglected. The lack of focus on calibrating distributed effective porosity on a regional scale might be related to the common assumption that recharge in humid climates can be precisely estimated and porosity of porous media is relatively well known from literature (Sanford 2011). However, for steady state flow in a layered aquifer system, Bethke and Johnson (2002) concluded that the groundwater age exchange between flow and stagnant zones is only a function of the volume of stored water. Thus, the groundwater age exchange is directly
related to the porosity. Yet, the calibration of a spatially distributed porosity field and its application to simulate groundwater ages and infer capture zones has not gained much attention.

The uniqueness of the presented study lies in the calibration of a three-dimensional, spatially distributed, effective porosity field in a regional-scale complex multi-layered heterogeneous coastal aquifer system. The aim is (i) to use apparent ages inferred from dissolution- and diffusion-corrected $^{14}C$ measurements from different aquifer units as targets in auto-calibration with
PEST of seven unit-specific effective porosities in an advective (particle tracking) transport model. A particle-based simulation



scheme (MODPATH) was evaluated as suitable in terms of the computational time while neglecting dispersion effects seemed to be acceptable at large scale using radiogenic old-age tracer ($^{14}C$) (Sanford 2011). (ii) to assess the advective age distributions at the sampling locations to obtain information on the age spreading; (iii) to apply the estimated seven porosities in a direct age simulation to gain insight in the three-dimensional age pattern of the investigation area and (iv) to assess the effect of using the heterogeneous porosity model compared to a homogeneous porosity model for differences in capture zones via particle back-tracking, which is a water management approach to define wellhead protection areas or optimize pump-and-treat locations for remediation of pollution (Anderson et al. 2015).

## 2   Study area

The $1300\,km^2$-large investigation area is located adjacent to the Wadden Sea in the border region between southern Denmark and northern Germany (Fig. 1). During the Last Glacial Maximum (LGM; 22 ka to 19 ka ago, Stroeven et al. 2016), the area was the direct foreland of the Scandinavian Ice sheet. The low-lying marsh areas (with elevations below mean sea level) in the west were reclaimed from the Wadden Sea over the last centuries and protected from flooding by a dike for the last $\approx 200$ a. A dense network of drainage channels keeps the groundwater level constantly below the ground surface, thus, mostly below sea level. The water divide near the Jutland ridge with elevations of up to 85 m a.s.l. defines the eastern boundary of the study area.

The aquifer systems are geologically complex and highly heterogeneous spanning Miocene through Holocene deposits. The bottom of the aquifer system is defined by low-permeability Palaeogene marine clay. The overlying Miocene deposits consist of alternating marine clay and deltaic silt and sand (Rasmussen et al. 2010). The Maade formation, an upper Miocene marine clay unit, with a relatively large thickness in the west while thinning out to the east, is located below the Pleistocene and Holocene deposits. Buried valleys filled with glacial deposits, mainly from the Saalian glaciation, cut through the Miocene and reach depths up to 450 m. They are important hydrogeological features as they may constitute preferential flow paths and locally connect the Pleistocene and Miocene aquifers.

In our previous studies (Jørgensen et al., 2015; Høyer et al., 2016a), the available geological and geophysical information including borehole lithology, Airborne Electro Magnetic (AEM) and seismic data were assembled into a heterogeneous geological voxel model comprising 46 geological units with raster sizes of 100 x 100 x 5 m. Manual and automatic modelling strategies, such as clay fraction (CF), multi-point simulation (MPS) and cognitive layer approach, were complementarily applied. Meyer et al. (2018a) investigated the regional flow system and identified the most dominant mechanisms governing the flow system comprising geological features and land management that are visualized in a conceptual model in Figure 2. Extensive clay layers separate the Miocene and Pleistocene aquifers, buried valleys locally cut through the Maade formation and connect Miocene and Pleistocene aquifers allowing groundwater exchange and mixing. The large drainage network, established in the reclaimed terrain keeping the groundwater table constantly below the sea level, acts as a large sink for the entire area. In the deeper aquifers, significant inflow from the ocean occurs at the coast near the marsh area as a result of a landward head gradients induced by the drainage.



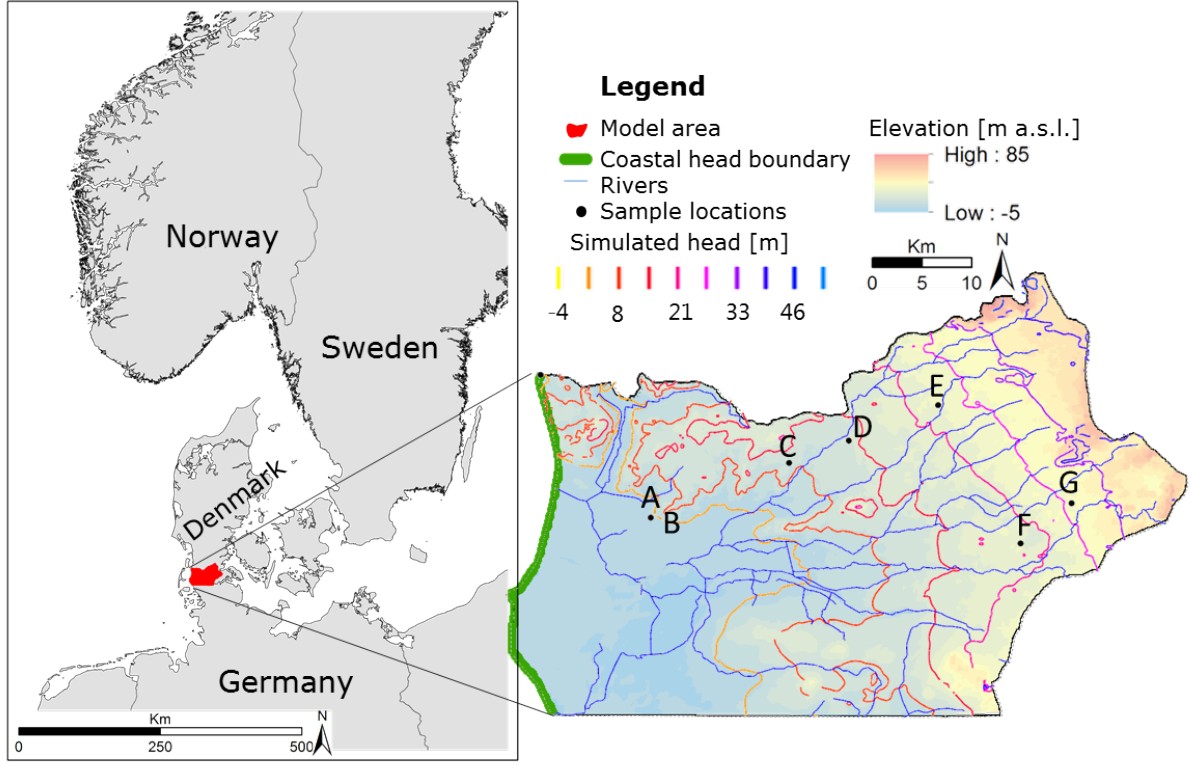

**Figure 1.** Investigation area at the border between Denmark and Germany. Simulated hydraulic heads are from the shallow aquifer (Meyer et al. 2018a). Topography, $^{14}C$ sample locations (A-G), river network and coastal head boundary are indicated.

## 3 Methods

The age simulation and calibration of effective porosities builds upon the calibrated regional-scale groundwater flow model (MODFLOW) of a highly heterogeneous coastal aquifer system by Meyer et al. (2018a). First, advective transport simulation using MODPATH (Pollock 2012) was used for the calibration of effective porosities of seven different geological units. $^{14}C$

5 observations were corrected for carbon dissolution and diffusion and subsequently used as calibration targets during inverse modelling with PEST. Secondly, the analysis of advective age distributions at $^{14}C$ sampling locations provided an insight in the ranges of travel times and distances and hereby the complexity of groundwater age mixing. Thirdly, the estimated effective porosities were used in a direct age simulation in order to investigate the spatial groundwater age distribution in the regional aquifer system. Finally, the impact of using a seven-porosities model compared to a constant porosity model on capture zone

10 delineation at two well locations was assessed.





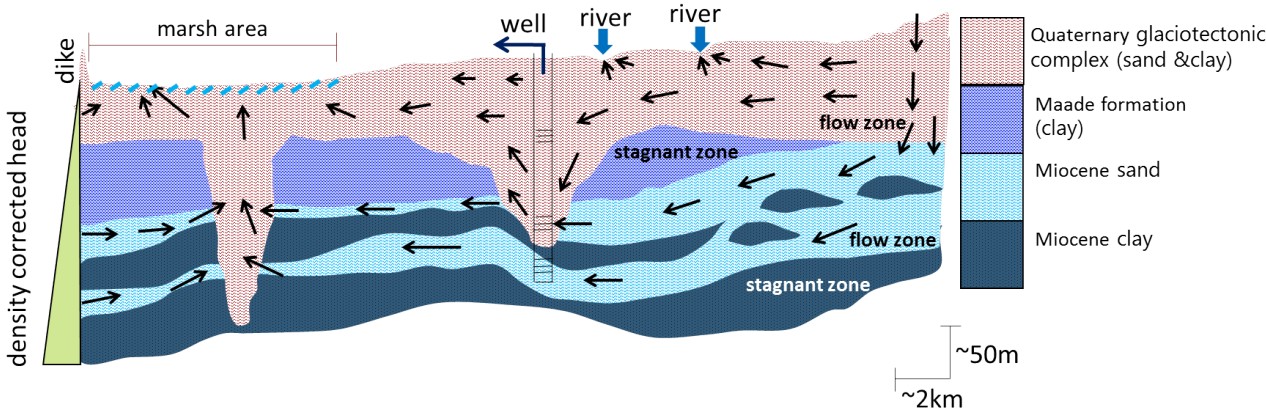

**Figure 2.** Conceptual regional model showing a simplified geology featuring buried valleys and groundwater flow and stagnant zones (as used for the diffusion correction). Arrows indicate general flow field of groundwater. Also shown are the boundary conditions, i. e. density corrected coastal boundary, drained marsh area, rivers.

## 3.1 $^{14}$C measurements

During a field campaign in February 2015, 18 groundwater samples were collected from wells at seven sites with screes at different depths and in different aquifers (Figure 1, Table 1). The wells were pumped clean three times their volume to prevent the influence of mixing with stagnant water. In situ parameters (pH, EC, O2) were measured and after they stabilized

samples for radiocarbon analyses were collected in 1-liter opaque glass bottles. The 18 groundwater samples were analyzed for $\delta^{13}C_{‰VPBD}$ with an isotope ratio mass spectrometer (IRMS) and for $^{14}C$ with an accelerator mass spectrometer (AMS) at the AGH University of Science and Technology, Krakow, Poland and in the Poznań Radiocarbon Laboratory, Poznań, Poland, respectively, in September 2015.

### 3.1.1 $^{14}$C correction for dissolution and diffusion

The $^{14}C$ activity ($A_m$) was measured in the dissolved inorganic carbon (DIC) content of the groundwater. Uncertainties arise from geochemical and hydrodynamic processes that change the $^{14}C$ content in the aquifer (e.g. Bethke and Johnson, 2008; Sudicky and Frind, 1981). The dissolution of fossil "dead" ($^{14}C$-free) carbon dilutes the $^{14}C$ content in groundwater and results in lower $^{14}C$ concentrations (Appelo and Postma 2005). Diffusion into aquitards also reduces the $^{14}C$ concentration in the aquifer (Sanford 1997). Both processes reduce the $^{14}C$ concentration and result in a groundwater age that is older than the

true age. Consequently, the measured $^{14}C$ activities were corrected for carbonate dissolution as well as aquitard diffusion prior to use in the calibration.





The initial $^{14}C$ activities were corrected for fossil carbon dissolution (Pearson and Hanshaw 1970) assuming an atmospheric $^{14}C$ activity ($A_0$) and soil $\delta^{13}C$ concentration in the soil $CO_2$ of 100 pMC (percent Modern Carbon) and -25‰, a dissolved carbonate concentration of 0 pMC and 0‰. With a decay rate constant ($\lambda$) of $1.21 \times 10^{-4}$ 1/year for the $^{14}C$ decay the dissolution-corrected age $\tau_C$ was calculated as (e.g. Bethke and Johnson, 2008)

$$\tau_c = \frac{-1}{\lambda} ln(\frac{A_m}{A_0}) \tag{1}$$

with

$$A_0 = \frac{\delta^{13}C}{-25} * 100 \tag{2}$$

A modified chemical correction was applied that takes into account the effect of dissolution as described by Boaretto et al. (1998). This method was successfully used in Danish geological settings similar to those investigated in the present study (Boaretto et al. 1998; Hinsby et al. 2001b).

Subsequently, a diffusion correction was made to take into account diffusion loss into low-permeability layers (Sanford, 1997). Aquitard diffusion is sensitive to porosity, diffusion coefficient and the thicknesses of the active flow (aquifer) and stagnant (aquitard) zones (Sudicky and Frind 1981). Because of the geological complexity, the sand-to-clay ratio based on voxel lithology was used to calculate the relative aquifer/aquitard (a/b = 0.72) thicknesses. Diffusion-corrected groundwater ages were calculated for three different diffusion coefficients: $1.26x10^{-9}$ m$^2$/s (Jaehne et al. 1987) representing the $CO_2$ diffusion in water, $1x10^{-10}$ m$^2$/s as an average for clay deposits (Freeze and Cherry 1979; Sanford 1997), and $2.11x10^{-10}$ m$^2$/s as calculated by Scharling (2011), using aquifer effective porosities ($n_e$) ranging from 0.16 to 0.35 and aquitard (b) thicknesses between 10 m and 50 m. Based on the ranges of variables, an average corrected age and the corresponding standard deviation were obtained for each sample (Table 1). Corrected groundwater sample age ($\tau_D$), also referred to as the apparent age, was calculated as:

$$\tau_D = \tau_C * \left( \frac{\lambda}{\lambda + \lambda'} \right) \tag{3}$$

with

$$\lambda' = 2 * \tanh\left[ \left(\frac{b}{2}\right) * \left(\frac{\lambda}{D}\right)^{\frac{1}{2}} \right] * \frac{(\lambda D)^{\frac{1}{2}}}{n_e a} \tag{4}$$

## 3.2 Groundwater flow model

Meyer et al. (2018a) simulated the 3D steady state regional groundwater flow using MODFLOW 2000 (Harbaugh et al. 2000). A brief description of the model set up and calibration results are presented here, further details can be found in Meyer et al. (2018a). The model was discretized horizontally by 200 m in the west and 400 m in the east and vertically by 5 m above 150 m





**Table 1.** Sampling wells, uncorrected and corrected groundwater ages. Gray shade indicates samples used for calibration. Note that lower numbers of the wells indicate deeper locations (m b.s. = meter below ground surface, std = standard deviation, pMC = percent Modern Carbon).

| well | DGU no. | filter depth [m b.s.] | aquifer geology | measured $^{14}$C [pMC] | uncorrected $^{14}$C [years] | $\Delta^{13}C_m$ [‰VDPD] | age corrected for dissolution and diffusion (std)[years] |
|---|---|---|---|---|---|---|---|
| A1 | 166.761-1 | 246-252 | Buried valley | 46.44 | 6161 | -13.2 | 344 (59) |
| A2 | 166.761-2 | 204-210 | Buried valley | 49.95 | 5576 | -13 | 108 (19) |
| B1 | 166.762-1 | 160-166 | Buried valley | 49.84 | 5593 | -13.9 | 293(50) |
| B2 | 166.762-2 | 102-108 | Buried valley | 51.9 | 5268 | -13.2 | 46 (8) |
| C1 | 167.1545-1 | 306-312 | Buried valley | 0.48 | 42889 | -5.9 | 10429 (1789) |
| C2 | 167.1545-2 | 273-276 | Buried valley | 1.03 | 36755 | -7.7 | 9097 (1569) |
| C3 | 167.1545-3 | 215-218 | Buried valley | 0.16 | 51714 | -11 | 15038 (2593) |
| C4 | 167.1545-4 | 142-149 | Buried valley | 33.84 | 8703 | -13.2 | 1191 (205) |
| C5 | 167.1545-5 | 116-123 | Buried valley | 43.18 | 6746 | -13.1 | 518 (89) |
| D1 | 159.1335-1 | 290-295 | Miocene | 1.8 | 32271 | -7.9 | 7671 (1323) |
| D2 | 159.1335-2 | 277-282 | Miocene | 1.35 | 34582 | -10.6 | 9229 (1591) |
| E1 | 159.1444-1 | 194-200 | Buried valley | 31.34 | 9320 | -12 | 1141 (197) |
| E3 | 159.1444-3 | 81-87 | Buried valley | 40.29 | 7302 | -12.8 | 642 (111) |
| F1 | 168.1378-1 | 372-378 | Miocene | 46.12 | 6216 | -12.3 | 173 (30) |
| F2 | 168.1378-2 | 341-345 | Miocene | 2.85 | 28580 | -13.3 | 7836 (1351) |
| F3 | 168.1378-3 | 208-214 | Miocene | 25.73 | 10904 | -12.6 | 1800 (310) |
| G1 | 168.1546-1 | 110-120 | Miocene | 42.57 | 6860 | -12.3 | 388 (67) |
| G2 | 168.1546-2 | 74-84 | Pleistocene/ Miocene | 45.33 | 6355 | -12 | 153 (26) |

b.s.l. and 10 m below 150 m b.s.l. resulting in 1.2 million active cells. The voxel geology was interpolated to the MODFLOW grid and 46 hydrogeological units were defined. No-flow boundaries were used in the north, east, south and at the bottom, where the Palaeogene clay constitutes the base of the aquifer system. At the western coast a density-corrected constant head boundary was applied (Figure 1; Guo and Langevin 2002; Post et al. 2007; Morgan et al. 2012). Distributed net recharge, averaged over

5 the years 1991-2010 was extracted from the national water resources model (Henriksen et al. 2003) and included as a specified flux condition. Internal specified boundaries were abstraction wells with a total flux of $26x10^6 \, m^3/year$ (averaged over the years 2000-2010, corresponding to 4% of the total recharge), rivers and drains.

Horizontal hydraulic conductivities, one for each hydrogeological unit, two anisotropy factors (Kh/Kv), one for sand and one



for clay units, as well as river and drain conductances were calibrated, using a multi-objective regularized inversion scheme (PEST; Doherty, 2016a), using head and mean stream flow observations as targets. The resulting head distribution is shown in Figure 1. Horizontal hydraulic conductivities were estimated in a range of 1-83 m/d for Pleistocene sand units, 0.028-0.19 m/d for Pleistocene clay units, 0.008-0.016 m/d for the Maade formation, 16-46 m/d for Miocene Sand and 0.14-0.23 m/d for Lower

Miocene Clay. The vertical anisotropy factor ($K_h/K_v$) was estimated to 25 and 85 for sand and clay units, respectively. The steady-state MODFLOW flow solution forms the basis for the advective transport simulation using MODPATH.

### 3.3  Advective transport model

Advective transport simulation was performed using MODPATH (Pollock 2012) in particle back-tracking mode. Hereby, the travel time of a particle (t), released in a cell, is calculated based on the MODFLOW cell-by-cell flow rates (q). The advective

travel time (t) is calculated as

$$t(x,y,z) = \int\limits_{x_0,y_0,z_0}^{x,y,z} \frac{\mathbf{n}_e(x,y,z)}{\mathbf{q}(x,y,z)} dx dy dz \tag{5}$$

In addition to the input data required by MODFLOW to generate the flow solution, MODPATH requires a value for effective porosity ($n_e$) to calculate the seepage velocity.

The groundwater age can be seen as the backward integration of travel times along the travel path back to its recharge location.

Hence, the simulated groundwater age is a function of the ratio of flux to effective porosity and the travel distance. In this study, the total flux is controlled by prescribed recharge and heterogeneous distribution of hydrogeological parameters (e.g. hydraulic conductivity, porosity).

In order to ensure stability (Konikow et al., 2008) 1000 particles were distributed evenly in the cell of the well screen and their average simulated particle age was compared with apparent groundwater ages (derived from equation 3).

The corrected $^{14}C$ ages were used as targets in the objective function (see below) of the simulated average travel time during calibration. According to Sanford (2011), neglecting hydrodynamic dispersion in advective transport simulations on a regional scale is a reasonable approach when old-age tracers, such as $^{14}C$, are used as dispersion might not be crucial for these tracers. On the other hand, diffusion into stagnant zones can create a significant loss in old-age tracer concentration which was taken into account by correcting the $^{14}C$ (paragraph 3.1.1) before calibration.

### 3.3.1  Calibrating porosity

The flow solution of the calibrated flow model (Meyer et al. 2018a) constitute the base for the 3D advective transport model. Depending on the depositional environment and clay/sand content, seven porosities corresponding to two Pleistocene sand, two Pleistocene clay, one Miocene sand and two Miocene clay units, were estimated using regularized (Tikhonov) inversion with PEST (Tikhonov and Arsenin 1977; Doherty 2016). As the calibration approach is similar to the one of Meyer et al. (2018a)

only additional characteristics are described in the following. Average corrected $^{14}C$ groundwater ages from 11 samples with a $^{14}C$ activity higher than 5 pMC (Table 1) were used as calibration targets. $^{14}C$ activity lower than 5 pMC were not used as it



was assumed that the boundary conditions of the flow model (e.g. sea level, recharge, head gradients) were not representative for pre-Holocene conditions. Moreover, the data from well F1 was excluded from calibration as an age inversion with F2 was observed here (Table 1), probably due to local heterogeneity or contamination of water with higher $^{14}C$ concentration, which is not possible to reproduce by the model. The average uncertainty of apparent ages was estimated to about 102 years. This value was based on the average of the standard deviation of the diffusion correction for the selected 11 samples and was used for weighting of the individual ages.

When Tikhonov regularization is applied a regularized objective function ($\Phi_r$) is added to the measurement objective function ($\Phi_m$) in form of the weighted least-squares of the residuals of preferred parameter values and parameter estimates. Within the limits of the user-defined objective function (PHIMLIM) and the acceptable objective function (PHIMACCEPT), the weight of the regularized objective function ($\mu$) increases and the parameter estimates are directed towards the preferred values. Calibration settings such as initial and preferred values and final parameter estimates are shown in Table 2. Values for PHIM-LIM and PHIMACCEPT were set to 60 and 100, respectively. The total objective function ($\Phi_{tot}$), minimized by PEST is then the sum of the measurement objective function ($\Phi_m$) and the regularized objective function ($\Phi_r$)

$$\Phi_{tot} = \Phi_m + \mu^2 \Phi_r \qquad (6)$$

with

$$\Phi_m = \sum \left( \omega_a \left( a_{obs} - a_{sim} \right) \right)^2 \qquad (7)$$

where $a_{obs}$ and $a_{sim}$ are observed and simulated groundwater ages, respectively, and the weight $\omega_a$ is the inverse of the standard deviation of the observed age. The calibration is evaluated based on the mean error (ME) and the root mean square (RMS) between apparent (corrected $^{14}C$ ages) and advective groundwater ages. Parameter identifiability (Doherty and Hunt 2009) is used to investigate to what extent the porosities were constrained through model calibration. Identifiability close to one means that the information content of the observations used during calibration can constrain the parameter. Parameters with an identifiability close to zero cannot be constrained.

### 3.4 Direct age

To visualize the mean groundwater age pattern in the regional 3D aquifer system, direct simulation of groundwater age was performed with MT3DMS (standard finite difference solver with upstream weighting) chemical reaction package using a zeroth-order production term (Goode 1996; Bethke and Johnson 2008). Hereby, mean groundwater age is simulated in analogy to solute transport as an "age mass" (Bethke and Johnson 2008). For each elapsed time unit (day) the water "age mass" increases by one day in each cell. Increase or decrease of ages is a results of diffusion, dispersion and advection (Bethke and Johnson 2008). The transient advection-dispersion equation of solute transport of "age mass" in three dimensions and with varying density and porosity is given by Goode (1996)

$$\frac{\partial a n_e \rho}{\partial t} = n_e \rho - \nabla a \rho \mathbf{q} + \nabla n_e \rho \mathbf{D} * \nabla a + F \qquad (8)$$





**Table 2.** Calibration settings: parameters with initial, preferred and estimated values for effective porosity.

| parameter ($n_e$) | Initial/preferred value | estimated value | % of cells | objective function | |
|---|---|---|---|---|---|
| Pleistocene sand 1 | 0.3 | 0.130 | 24.4 | PHIMLIM | 60 |
| Pleistocene sand 2 | 0.3 | 0.263 | 2.5 | PHIMACCEPT | 100 |
| Pleistocene clay 1 | 0.1 | 0.085 | 11.6 | $\phi_m$ achieved | 74 |
| Pleistocene clay 2 | 0.05 | 0.043 | 4.8 | | |
| Miocene sand | 0.3 | 0.450 | 15.1 | | |
| Miocene clay | 0.1 | 0.102 | 22.8 | | |
| Miocene clay (Maade formation) | 0.05 | 0.049 | 18.8 | | |

where F is an internal net source of mass age, $\mathbf{q}$ the Darcy flux (m/d), a the mean age (d), $n_e$ the effective porosity, $\rho$ the density of water ($kg/m^3$) and $\mathbf{D}$ the dispersion tensor ($m^2/d$), including molecular diffusion and hydrodynamic dispersion. The initial concentration of the "age mass" was set to zero, while a constant age of zero was assigned to the recharge boundary and the constant head boundary at the coast. Steady state conditions were evaluated based on the change in mass storage in a 40000

5    year simulation. The age mass storage (m) in the whole model was calculated for each time step as the sum of mass in each cell ($m_i$). The latter was calculated by multiplying the cell dimensions ($\Delta z, \Delta x, \Delta y$) with porosity ($n_e$) and age ($a_s$)

$$m = \sum m_i \tag{9}$$

with

$$m_i = \Delta z * \Delta x * \Delta y * n_e * a_s \tag{10}$$

10    The percentage change in mass storage ($\Delta m_t$) per time step ($\Delta t$) was calculated as

$$\frac{\Delta m_t}{\Delta t} = \left( \frac{m_t - m_{t-1}}{m_{t-1}} \right) * 100 \tag{11}$$

The integral of the change in mass storage over time was used to define quasi-steady state conditions. This was reached when

$$\int_{t1}^{t} \Delta m(t)dt \geq 0.95 * \int_{t1}^{\infty} \Delta m(t)dt \tag{12}$$

Dispersion experiments were carried out for longitudinal dispersivity $\alpha_L$ values of $0\,\mathrm{m}$, $5\,\mathrm{m}$, $20\,\mathrm{m}$, $50\,\mathrm{m}$, $500\,\mathrm{m}$, while the

15    horizontal transversal $\alpha_{TH}$ and vertical transversal $\alpha_{TV}$ dispersivities were specified to $10\%$ and $1\%$ of $\alpha_L$, respectively. A diffusion coefficient of $1x10^{-9}\,\mathrm{m^2/s}$ was used to account for self-diffusion of the water molecule at about $10\,^{\circ}\mathrm{C}$ (Harris and Woolf 1980).



### 3.5 Capture zones

Well capture zones are used in water management to define areas of groundwater protection, where human actions, such as agricultural use, are restricted. Simulated by the single-porosity and seven-porosities model the capture zones of one existing well (Abild, abstraction rate $27\,m^3/d$) located in a buried valley and one virtual well (AW, abstraction rate $280\,m^3/d$) located

in a Miocene sand aquifer were evaluated and compared for different back tracking times using 100 particles per well. No differences in the area of the whole capture zone are expected as porosity does not impact the trajectory of the particle path (Hill and Tiedeman 2007) and only affects the travel time. Hence, the capture zone area at different times were compared.

## 4 Results

### 4.1 $^{14}C$ corrections

Figure 3 shows the corrected and uncorrected $^{14}C$ ages over depth. Except for well F, ages increase with depth at each multi-screen location. Otherwise, no clear trend between age and depth can be identified on the regional scale. Uncorrected ages range from 5000 to 50000 years (Table 1). After correction, all ages decrease and the relative difference between the corrected ages increase, now within a range from 46 to 15000 years. Hence, it is expected that the oldest water recharged the groundwater at the end of the last glacial period. The majority of the samples represent younger waters with 12 out of 18 samples being less

than 2000 years.

### 4.2 Calibration results

The match between the average of simulated groundwater ages (particle tracking with MODPATH) and corrected $^{14}C$ ages is shown in Figure 4a.

Results from the seven-porosities model were compared to those from a single porosity model with an effective porosity of

0.3 which is a typical textbook value for porous media (Holting and Coldewey 2013; Anderson et al. 2015) and often used in groundwater modelling studies (e.g. Sonnenborg et al., 2016). The seven-porosities model is able to match all the observations reasonably. This is not the case for the single porosity model where especially one sample is poorly simulated with an estimate of more than 5500 years whereas the corresponding observation only reach about 1200 years. The ME and RMS of the calibrated seven-porosities model were -2.3 years and 267 years, respectively, which correspond to a reduction in ME of 99%

and RMS of 82% compared to the single porosity model. Considering the uncertainties involved in estimation of apparent age, see uncertainty estimates in Table 1, column to the right, the match is found acceptable. Comparing the average uncertainty on apparent ages used for calibration of 102 years with the achieved RMS of 267 years indicate that no overfitting occurred and mismatches can be a result of small scale heterogeneity below grid resolution.

The estimated porosities of the seven hydrogeological units are listed in Table 2. Realistic values are found for all parameters

and the values of the sand units are generally higher than those of the clay units. However, the estimate of 0.13 for Pleistocene sand 1 is relatively low. This may be explained by the fact that this unit do not represent sand exclusively everywhere. The




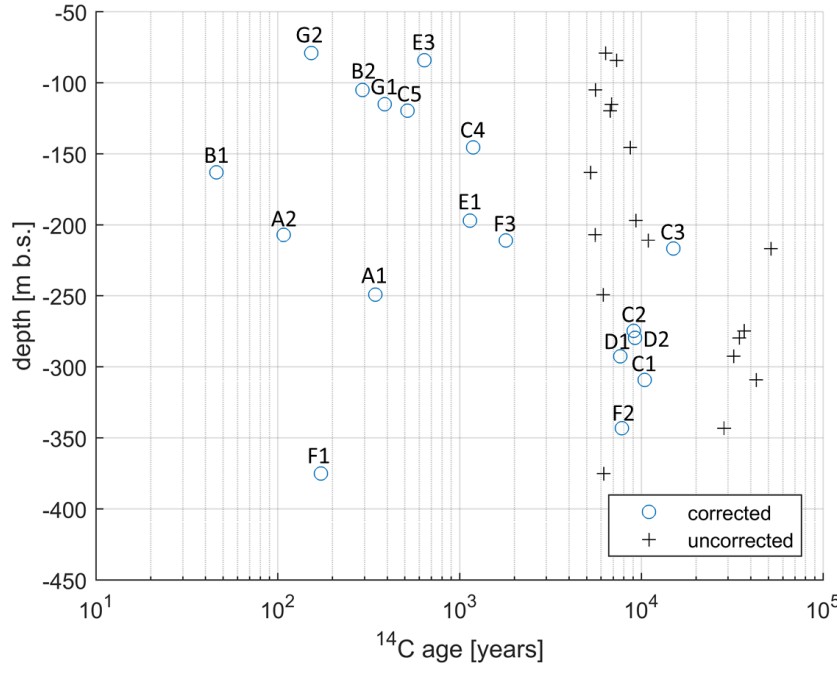

**Figure 3.** Apparent groundwater $^{14}C$ ages as a function of groundwater sampling depth: black crosses indicate ages without correction for dissolution and diffusion, blue circles show ages with correction. Labels indicate well location and filter number (cf. Fig 1 and Table 1).

Pleistocene deposits in the area are highly heterogeneous (Jørgensen et al. 2015) and it is therefore difficult to identify units exclusively composed of sand, partly due to the difficulties in using AEM data to guide the distinction between sand and clay at a relatively small scale. Hence, Pleistocene sand 1 may to some extent represent a mixture of sand and clay. Additionally, uncertainties in the estimates of hydraulic conductivity from Meyer et al. (2018a) will translate into errors in seepage flux and

5 hence ages. Uncertainties and errors in hydraulic conductivity may therefore be partly compensated by estimates of effective porosity that are somewhat different from the expected value.

The parameter identifiability (Figure 4b) shows that the corrected $^{14}C$ ages may constrain four out of seven estimated effective porosities, i.e. of Pleistocene sand 1, Pleistocene clay 2, Miocene sand and Miocene clay. The warmer colors (red-yellow) indicate that the parameter is less influenced by measurement noise (Doherty, 2015, Figure 4). Where the parameter identifia-

10 bility is relatively low ($< 0.8$), i.e. for effective porosities of Pleistocene sand 2, Pleistocene clay 1 and Miocene clay (Maade), the estimated parameter value is more constrained by the regularization and hence stay close to the preferred value (Table 2). The low identifiability is a result of the distribution (or density) of observations compared to the particle travel paths. Figure 5 shows the pathlines of particle back-tracking (for better visualization only one path line is shown per well screen). As mentioned above, only $^{14}C$ observations with an activity higher than 5 pMC (Table 1) were used, which excludes results from

15 wells C, D and F. The recharge area is mostly located to the east (Figure 5). The Maade formation is more dominant towards

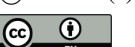



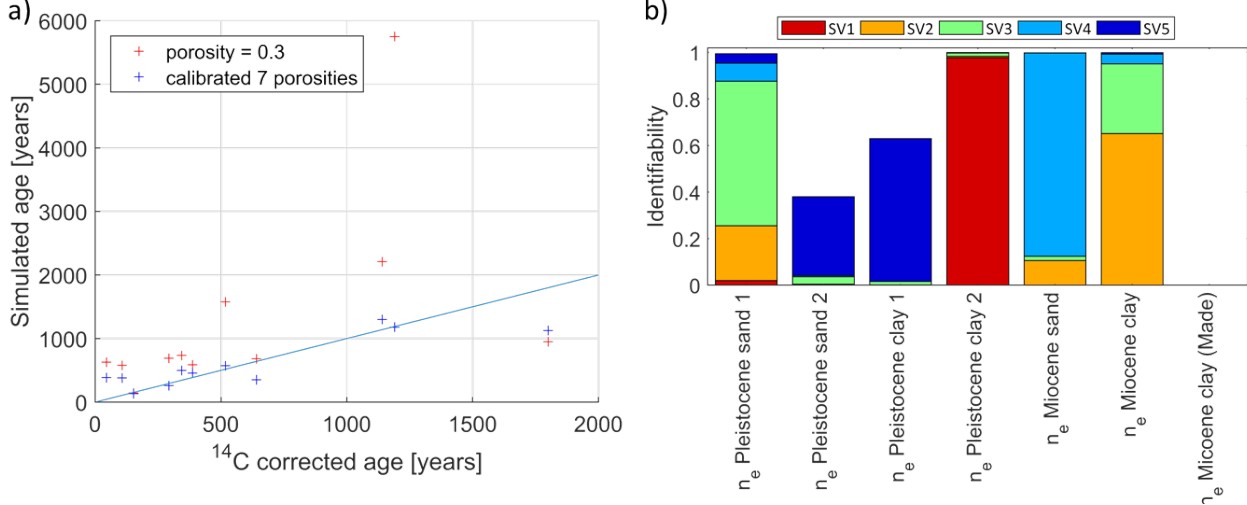

**Figure 4.** Calibration results: a) red crosses show apparent ages simulated with a porosity of 0.3 as often used in porous media models and blue crosses are ages simulated based on 7 calibrated porosities (Table 2). b) parameter identifiability of effective porosities (warmer colors correspond to singular values (SV) of a lower index, cooler color to SV of higher index) of the different geological formations; the identifiability of the Maade porosity is close to zero.

the west while it is patchy in the east. Consequently, it does not affect the particle tracking as much in the east. Only effective porosities of geological units through which particles actually travel (primarily sand) are well informed by the observations. The low-permeability Maade unit acts as an obstacle to the travel paths and since the particles circumvent the Maade formation the actual value of porosity has no impact on the age. The Maade unit significantly affects the age distribution due to its

influence on travel paths, but no sensitivity to the porosity of the unit is found.

Pleistocene sand 2 represents less than 5% of the total amount of cells (Table 2) and it occurs mostly in the west. Pleistocene clay 1 is mostly shallow, patchy and located far away from the well locations. Hence, the impact of these two geological units on the particle tracks is also relatively small and results in low identifiability (Figure 4b).

### 4.3 Advective age distribution at observation wells

Figure 6 shows the simulated advective age distribution at the sampling locations (A-G, Figure 1), split up in three different age groups.

The results show a wide variety of mean ages (Table 3) and the shape of the age distributions is very different. The well screens with mean particle ages less than 1000 years (Figure 6a) show particle age distributions that are mostly narrow and unimodal (except E3), which is also reflected in a small standard deviation (smaller than 20% of the mean age, except E3), see

column B in Table 3. The particle age distributions of older waters with a mean particle age larger than 1000 years (Table 3) tend to have broader and/or multi-modal shapes (Figures 6b,c) and large standard deviations (Table 3, column B).



**Figure 5.** Horizontal geological cross-section at an elevation of -100 m a.s.l. and a SW-NE cross-section through sampling well locations (A-E). Bluish (cold) colors represent pre-Pleistocene sediments (dark blue = clay, light blue = sand), while warm colors represent Pleistocene deposits (red = sand, brown = clay). Also shown are MODPATH back-tracking lines (1 per cell) and groundwater flow velocity vectors.

The mean distance that particles travel from their recharge points to the sampling well (Table 3, column D) ranges between





**Table 3.** Results of the analysis of particle age distributions and path lengths.

| | A | B | C | D | E |
|---|---|---|---|---|---|
| well | mean particle age [years] | std particle age [years] | median particle age [years] | mean path length [km] | std path length [km] |
| A1 | 536 | 72 | 503 | 7.50 | 0.22 |
| A2 | 392 | 16 | 387 | 6.17 | 0.36 |
| B1 | 400 | 71 | 367 | 6.28 | 0.41 |
| B2 | 272 | 31 | 277 | 3.69 | 0.61 |
| C1 | 7232 | 2814 | 6503 | 26.68 | 1.64 |
| C2 | 3654 | 2816 | 2818 | 27.52 | 0.94 |
| C3 | 2640 | 608 | 2542 | 27.83 | 0.86 |
| C4 | 1038 | 45 | 1036 | 3.24 | 0.07 |
| C5 | 542 | 116 | 512 | 3.19 | 0.17 |
| D1 | 14122 | 7563 | 13479 | 22.12 | 1.05 |
| D2 | 5028 | 4498 | 3064 | 21.94 | 0.97 |
| E1 | 1306 | 1508 | 908 | 13.56 | 0.82 |
| E3 | 404 | 448 | 300 | 11.50 | 1.30 |
| F1 | 6649 | 1405 | 6394 | 17.01 | 0.47 |
| F2 | 2950 | 1584 | 2768 | 16.34 | 0.78 |
| F3 | 1129 | 60 | 1120 | 14.83 | 0.57 |
| G1 | 470 | 258 | 514 | 6.82 | 0.62 |
| G2 | 135 | 17 | 130 | 6.02 | 0.46 |

3 km and 28 km. The younger waters ($<$1000 years) show path lengths less than 10 km (except at well location E3 and F), Figure 7, while most of the older waters travel more than 20 km. However, the relation between path length and travel time is far from linear. At some well locations (e.g. well locations A2, B2, C4, C5) the relation between path length and travel time forms a few distinct small clouds without much spread, indicating that the particles follow alternative large-scale preferential

5 flow paths. At other locations a larger and more diffusive spread is found, either in travel times (e.g. well locations C1, C2, C3, D1, E1) or path lengths (e.g. well locations F3, G2, E3). The large spread in travel times indicate that some particles travel slowly through clay units of various thicknesses. The large spread in path lengths originates from long and quick or short and slow travel paths through or around clay units and reflects the geological heterogeneity.

## 4.4 Regional age distribution based on direct age simulation

10 Figure 8 shows the ME and RMS of the direct age and the apparent age (corrected $^{14}C$) for different $\alpha_L$ values ($\alpha_{TH}$ and $\alpha_{TV}$ are tied to $\alpha_L$, see section 3.4). Minimum ME and RMS values are achieved for longitudinal dispersivities $\alpha_L$ $<$5 m. For lower $\alpha_L$ the effect on ME and RMS is insignificant as numerical dispersion is expected to dominate at this scale. With higher $\alpha_L$



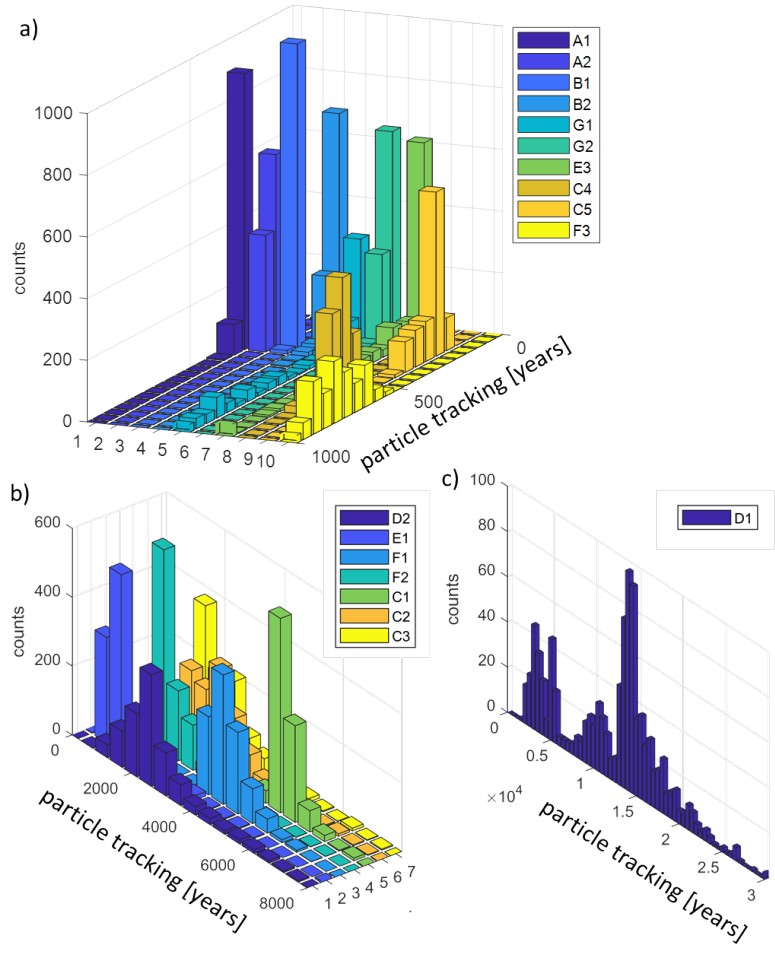

**Figure 6.** Particle age distributions at sampling wells A-G (see Figure 1 for locations). a) young waters (bin size = 50 years) show a narrow, unimodal distribution; b) old waters (bin size = 500 years) have broader and often multimodal distributions; c) multi-modal age distribution at sample location D1, which shows the longest travel times.

values ME and RMS increase significantly. Other regional-scale studies (e.g. Sonnenborg et al., 2016) have used longitudinal dispersivity in the magnitude of tens of meters or more to account for geological heterogeneities at formation scale. The very detailed voxel geological model that resolves heterogeneities at a scale of 200 x 200 m justifies the use of a relatively small $\alpha_L$. Hence, the dispersivity only describes the effect of heterogeneity at the grid scale, 200 m. In accordance with Gelhar et al.

5 (1992) this results in $\alpha_L$ with a magnitude of a few meters. Since there is no sensitivity for lower $\alpha_L$ (numerical dispersion dominates at this scale), a macrodispersivity of 0 m was used in the following simulations of direct age.

The age distribution on a regional scale (Figure 9) shows a general age evolution from young water in the recharge area in the east towards older water in the west (Figure 9 b, e, f). Young water also enters the system through the coastal boundary



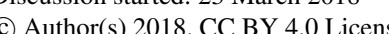

**Figure 7.** Particle tracking time over path length for the different well locations (cf. Figure 1.; the screen depth is indicated in parentheses).

in the west (Figure 9 b, e, f). The age distribution is strongly affected by geology and is therefore in good agreement with the interpretation of the flow system by Meyer et al. (2018a). Two main aquifers are present on a regional scale: a shallow Pleistocene sand aquifer and a deep Miocene sand aquifer, separated by the Maade formation and locally connected through




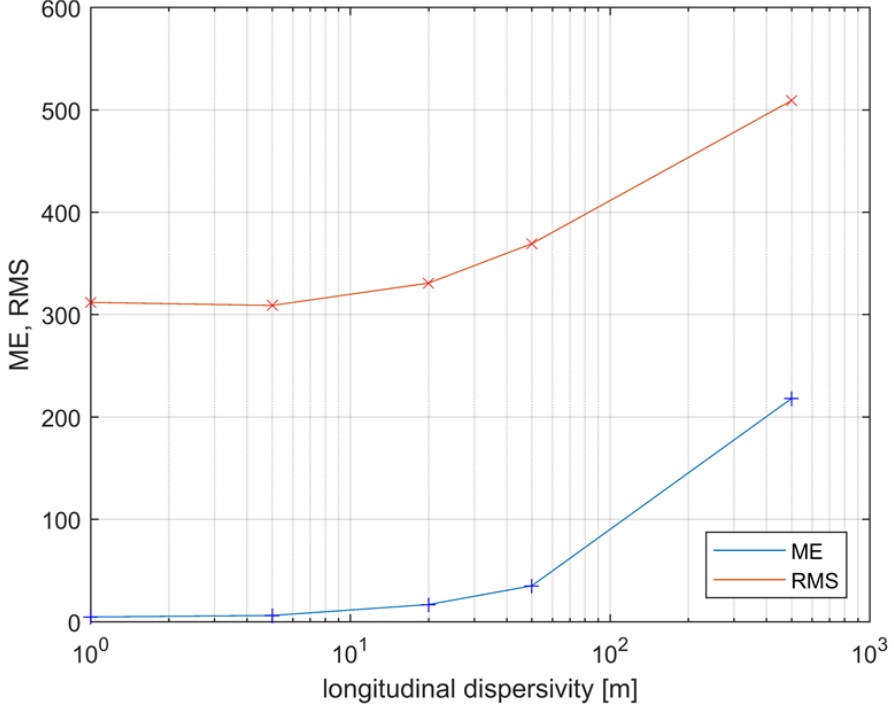

**Figure 8.** Mean error (ME) and root mean square (RMS) between corrected $^{14}C$ (shaded in gray in Table 1) and directly simulated ages as a function of longitudinal dispersivity.

buried valleys (conceptual model in Figure 2, Figure 9 g, h). The regional age distribution also reflects this system. Younger waters dominate the shallow Pleistocene aquifers (Figure 9 a, e, f), where the flow regime can be described as mostly local and intermediate (cf. Tóth, 1963). The separating Maade formation with its increasing thickness towards the west (Figure 9 d) acts as a stagnant zone where groundwater age increases (Figure 9 c). The underlying Miocene sand shows the age evolution from

5 young water in the recharge areas in the east to older water towards the discharge zones in the west (Figure 9 b, e, f). Here the flow regime is dominated by regional flow (cf. Tóth 1963). Special features are the buried valleys where downward flow of young waters, upwelling of old waters and mixing occurs (Figure 9 e, f, g, h). At the coastal boundary in the west young water enters the system and due to the density-corrected head boundary a wedge is formed with young waters in the wedge and old water accumulating in the transition zone (Figure 9 e, f). Another feature is the human land use change including an extensive

10 drainage network with drain elevations below the sea level in the marsh area. There, old groundwater is forced upward, partly through buried valleys, before it can discharge to the sea.

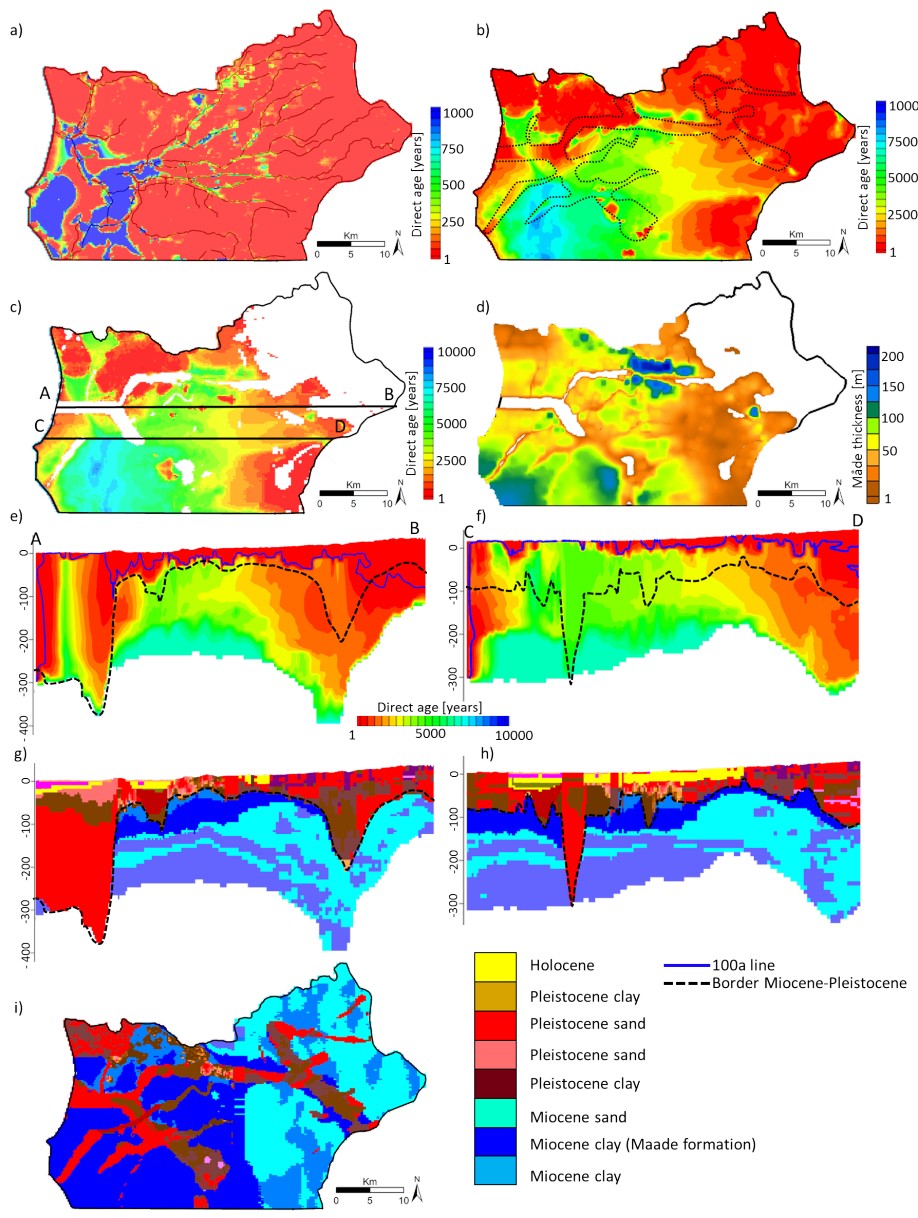

**Figure 9.** Directly simulated ages and velocity vectors presented at: a) horizontal section at layer 2, also showing river network; b) horizontal section at a depth of 100 m a.s.l. (buried valleys indicated with dotted lines); c) horizontal section at the top of the Maade formation; d) extent and thickness of the Maade formation; e) cross-sections A-B and f) C-D; Pleistocene-Miocene boundary indicated with dashed lines (buried valleys), 100 year lines; g) and h) geological cross-section and i) horizontal geological section, main geological units indicated (a detailed geological description is given in Meyer et al. 2018a)


### 4.4.1 Direct simulated mean age distribution in geological units

The steady state distribution of average groundwater age was reached after 26000 years. In Figure 10 the normalized direct age distributions are shown for a) the whole model, b) the Pleistocene aquifer, c) the Maade clay formation that acts as an aquitard, and d) the Miocene sand aquifer (compare the geological setting with conceptual model in Figure 2). The directly

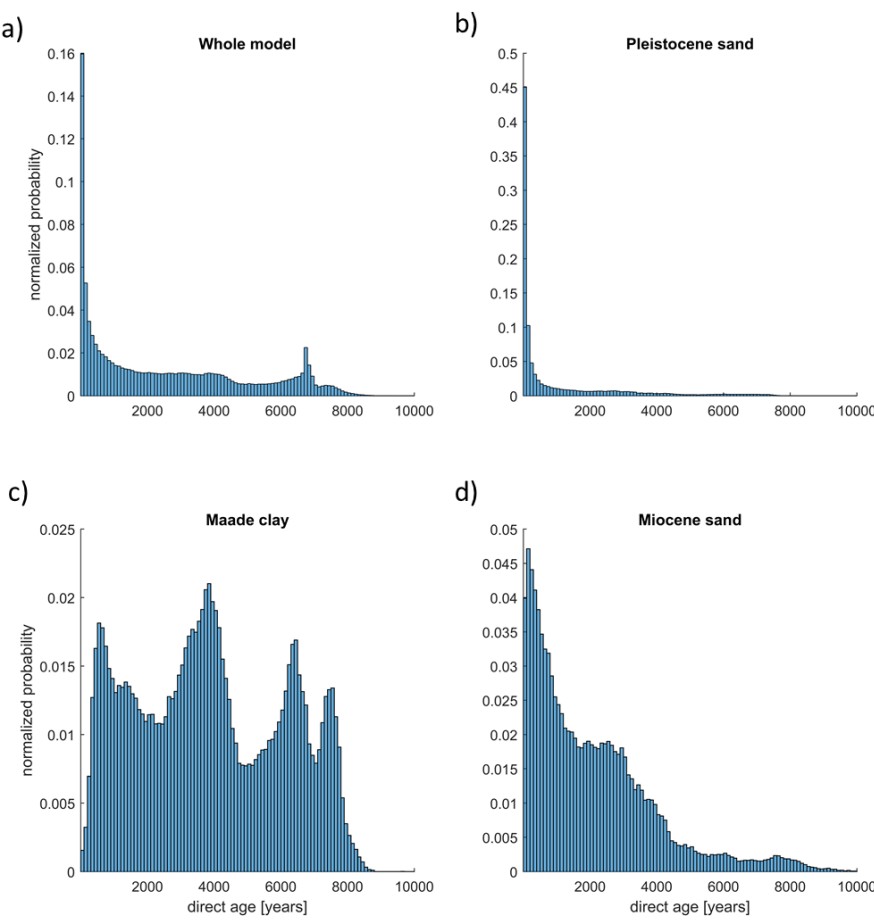

**Figure 10.** Normalized probability distribution (bin size = 100 years) of directly simulated groundwater ages in a) the whole model, b) the shallow Pleistocene aquifer, c) the separating Miocene clay (Maade formation) and d) the deep Miocene aquifer.

5    simulated mean groundwater ages for the whole model, the Pleistocene sand, the Maade formation and the Miocene sand were determined by a moment analysis (Levenspiel and Sater 1966) as 2574, 1009, 3883, and 2087 years, respectively. The shape of the age distribution in these units varies significantly. The Pleistocene sand shows a unimodal distribution with one peak at 100 years and a tail (Figure 10b). The age distribution is governed by recharge of young water and discharge through rivers





and drains, which are fed by the upwelling older groundwater (Figure 9a). The age distribution in the Maade formation is multi-modal with five peaks at about 600, 1400, 3900, 6500 and 7600 years (Figure 10c). Comparison of Figures 9c and 9d reveals a positive relation between age and thickness of the Maade formation. The age distribution in the underlying Miocene sand has one peak at 200 years followed by a plateau between 1600-3100 years and a small peak at 7800 years (Figure 10d).

5      This distribution is controlled by the overlying and separating Maade formation in the west and the interlayering with Miocene clay.

### 4.4.2   Advective and directly simulated ages

The comparison of the advective ages with the direct simulated ages at the sampling well locations shows a good match for advective ages with a small variance and worsens when the variance increases (Figure 11). Older ages are generally associated

10     with larger variances. Where the mismatch between advective and direct ages is large, the direct simulated mean ages are consistently lower than mean ages derived from particle back tracking (see discussion below) because of diffusion into clay units. However, most of them lie within one standard deviation.

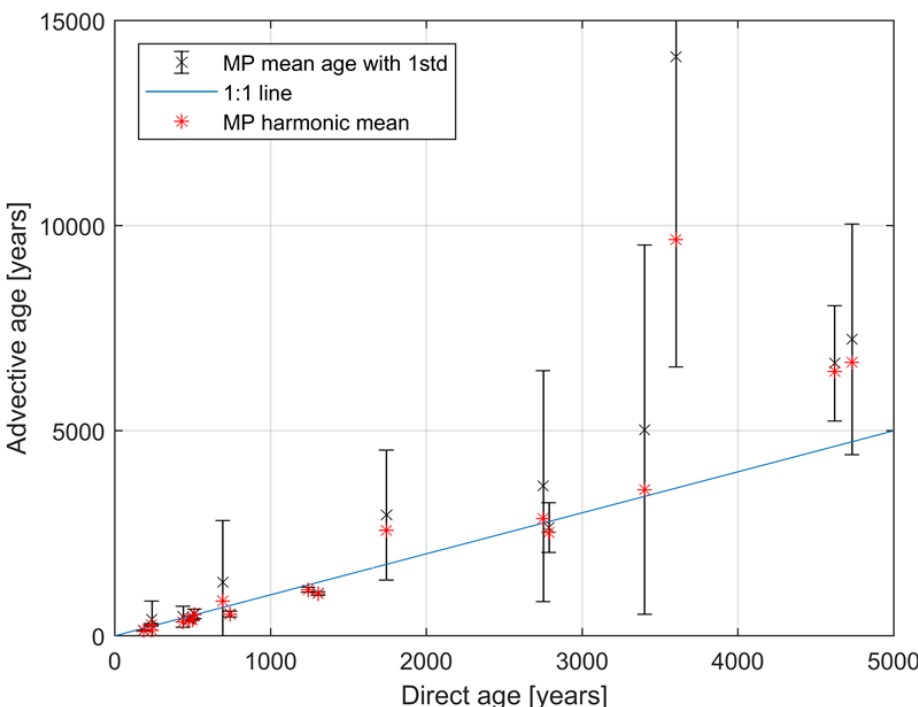

**Figure 11.** Mean advective age (MODPATH (MP) particle backtracking) compared to directly simulated mean groundwater age at sampling well locations; error bars on advective age represent 1 standard deviation; 1:1 line in blue; harmonic mean of advective age in red.





### 4.4.3 Capture zones: effect of porosity

Figure 10 shows the capture zones at the Abild well for 1500 and 2000 years and for the virtual well (AW) for 1000, 2000 and 3000 years for a constant porosity of 0.3 (solid line) and the calibrated seven-porosities model (dashed line), respectively. The capture zones of the two models vary both in extent and shape. The areas of the capture zone differ by up to 50%. Interestingly, it is not always the same porosity model that has the smaller capture zone, but it changes due to the heterogeneity in the geological model and the assigned porosities. However, the results illustrate the importance of reliable estimates of effective porosity when delineating the capture zone of an abstractions well.

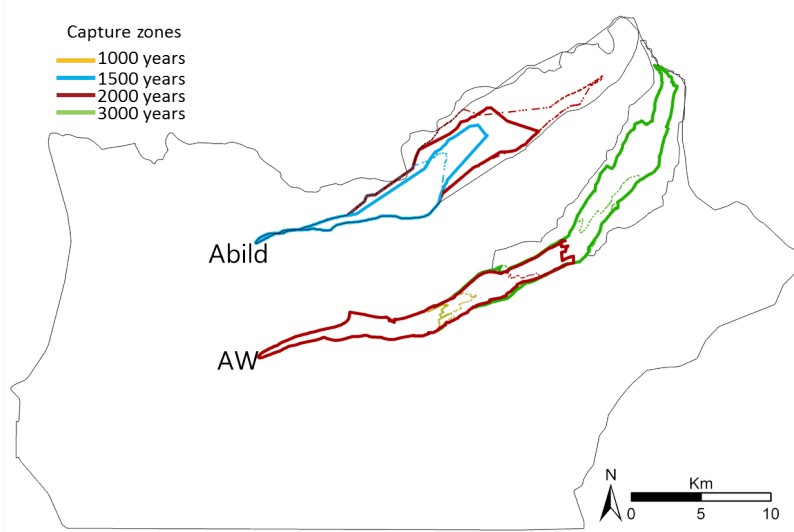

**Figure 12.** Capture zones at a well in Abild and a virtual well (AW) with a comparison of capture zones for a model with homogeneous porosity of 0.3 in all geological units (solid lines) and one with seven different porosities (dashed lines).

## 5 Discussion

$^{14}C$ observations were used to constrain the estimation of effective porosities of a large-scale coastal aquifer system using an approach similar to Konikow et al. (2008) and Weissmann et al. (2002). Advective transport modelling and direct age simulations were applied to gain insight into the regional age structure of this highly heterogeneous geological system. In the following, limitations, uncertainties and simplifications of the model structure, estimated parameters and resulting interpretations are discussed. A detailed description of the age distribution is provided to highlight the relevant physical processes and their interactions.



## 5.1 Uncertainties

### 5.1.1 Boundary conditions

Uncertainties in model results originate partly from simplifications in boundary conditions and geological heterogeneities that are not resolved at the grid scale. Groundwater recharge, drain levels, well abstractions and sea levels were assumed constant over time for practical reasons and to reduce computational time. However, Karlsson et al. (2014) showed that recharge has changed significantly in Denmark during the last centuries. Changes in recharge could result in different age patterns (cf. Goderniaux et al. 2013). Similarly, sea level changes that were disregarded in this study would have an effect on the groundwater age distribution in the coastal areas. Prescribing a vertical coastal age boundary of zero years is another simplification that neglects the vertical mixing and dispersion, which would result in an increase of age with depth (Post et al. 2013). However, since these physical processes were difficult to quantify, estimating age at this boundary would be highly uncertain. Thus, a constant age of zero years was applied.

### 5.1.2 Apparent age as calibration target

Uncertainties in the use of $^{14}C$ as a groundwater dating tool and as calibration target arise at different levels. First, sampling of well screens with a length of 6 -10 m would encompass a range of groundwater ages as a result of mixing of groundwater of different ages. Hereby younger waters, corresponding to DIC with a higher $^{14}C$ content, would dominate older ages (Park et al. 2002). The $^{14}C$ content is measured in the DIC of the groundwater. In order to obtain a reliable age estimate, the origin of DIC in groundwater is important. For the different processes that can affect the DIC and change its $^{14}C$ content (e.g dissolution, precipitation, isotopic exchange) a variety of correction models exists (see overview of correction models in IAEA 2013). For the investigated system, corrections for carbonate dissolution and diffusion were applied, but it cannot be ruled out that also other chemical processes might have changed the $^{14}C$ content over the past thousands of years. The $^{14}C$ correction for diffusion into stagnant zones is sensitive to aquifer porosity, aquitard thickness and diffusion constant. The geology is highly complex and aquitard thickness and porosity distribution change spatially over the entire region, whereas the correction terms were based on the general behaviour of the voxel system. Hence, average values of diffusion corrections were applied with parameters varying in ranges realistic for an aquifer system at this scale. However, in reality a groundwater particle would have been exposed to a variety of aquifer/aquitard thicknesses and porosities along its flow path implying smaller or larger diffusion. The correction results show that both carbonate dissolution and diffusion into stagnant zones reduce the apparent groundwater age considerably, both at a similar magnitude as observed by Scharling (2011) and Hinsby et al. (2001).

Finally, the calibration of effective porosity using an advective transport model relies on a calibrated 3D flow solution that already bears uncertainties with respect to structure and parameters, as addressed by Meyer et al. (2018a). The number and position of the released particles contribute to the uncertainty especially in heterogeneous systems as pointed out by Konikow et al. (2008) and Varni and Carrera (1998). The use of a high number of particles – here 1000 particles were distributed in one cell – generally reduces the uncertainty and enhances stability of the solution. The arithmetic mean of the 1000 released particles evenly distributed in the sampling cells resulted in estimates of effective porosities in the range of 0.13 to 0.45 for



sand and 0.043 to 0.1 for clay units, which is significantly different to porosities of 0.25 or 0.30 that are often used in porous media(e.g. Sonnenborg et al., 2016). The reliability of the estimated effective porosities was assessed through the identifiability that depends on the observation density (see section 4.2) and is high for four out of the seven estimated porosities.

### 5.1.3   Mean age

The differences between mean advective ages and directly simulated mean ages as described in section 4.4 can be related to the simulation methods. While the direct age corresponds to the flux-averaged mean, the particle tracking age is resident-averaged (Varni and Carrera 1998). Hence, the age distribution of the 1000 simulated particles, especially when it is broad and multi-modal, shifts the mean age towards older ages. By using the harmonic mean of travel times of particles back-tracked from one cell (Konikow et al. 2008) more weight is given to younger ages which would more closely correspond to a flux-weighted mean.

This approach improves the comparison (Figure 11; red stars), especially at wells, where the variances are large. Nonetheless, this approach is empirical and do generally not guarantee a better result. Hence, there are still some mismatches that can be related to the diffusion and dispersion processes (here represented by numerical dispersion as physical dispersion was set to zero), which are included in the direct age approach, but neglected in simulating advective ages.

## 5.2   Flow system and age distribution interpretation

### 5.2.1   Advective age distribution

The analysis of the advective age and travel distance distributions (Figures 6 and 7, Table 3) revealed a larger variance of ages for waters with a higher mean age. Following the pathlines of wells with younger waters (e.g. Figure 5, well locations A, B, C4, C5 and G), recharge areas are more proximal (path length <10 km, Figure 7, Table 3). Consequently, the particles pass through fewer hydrogeological units and hence the flow path is less influenced by heterogeneous geology, which results in

a smaller variance in ages and path lengths (Figure 7, Table 3). Particles traveling to well locations C1, C2, C3, D, E and F (e.g. Figure 5) have to travel through a variety of hydrogeological units, characterized by different hydraulic conductivities and effective porosities, hence showing a broader age distribution and larger variance as well as longer travel distances. Their broad and multi-modal age distributions reflect the up-gradient heterogeneity in fluxes, related to hydraulic conductivity and effective porosity. This behavior is in accordance with conclusions by Weissmann et al. (2002) who investigated groundwater

ages in a heterogeneous 3D alluvial aquifer based on particle tracking and CFC-derived ages.

### 5.2.2   Regional age pattern

The regional age pattern derived from direct age simulation is consistent with the findings of Meyer et al. (2018a) about the flow system. The two-aquifer system is separated by a confining aquitard in the west. The shallow aquifer system consisting of glaciotectonically disturbed Pleistocene sands mixed with clays is dominated by local and intermediate flow regimes and

contains water of younger ages. The confining aquitard (Maade formation) shows older waters and a positive relation between ages and aquitard thickness what agrees with Bethke and Johnson (2008). In the deep Miocene sand aquifer that is interbedded





with Miocene clay, regional flow regimes dominate and groundwater ages vary from young waters in the recharge areas in the east, where the overlying confining aquitard does not exist, to very old waters (up to 10000 years) in the west. The confining Miocene aquitard (Maade formation) influences the age distribution pattern in the underlying Miocene sand in two ways. First, it limits deeper groundwater to seep upward and mix with the younger waters in the shallow aquifer. Secondly, the age flux

from the aquitard to the aquifer shows a positive correlation with the ratio between aquitard thickness and aquifer thickness (Bethke and Johnson 2008).

At the buried valleys, groundwater exchange and hence age mixing occurs. Upwelling of the older groundwater from the deeper aquifer happens preferentially through these buried valleys. The dense drainage network in the west close to the coast

acts as a regional sink, with younger groundwater flowing horizontally and older water vertically and discharging to the drains. At the coastal boundary in the west, where a constant concentration of an "age mass" zero was assigned to the density-corrected constant head boundary, an age wedge characterized by waters of contrasting ages is established as a result of intruding young ocean water that meets old waters in the transition zone. This agrees with the findings by Post et al. (2013) based on simulation of synthetic groundwater age patterns in coastal aquifers using density-driven flow.

The results of our study differ significantly from findings by Sonnenborg et al. (2016) who investigated a regional aquifer system with a similar geological setting located a few hundred kilometers north of the present study area. Their direct simulation of groundwater ages shows a pattern of much younger water than here, rarely exceeding 700 years even in the deepest aquifers, while in our study ages exceeding 10000 years occur. The discrepancies may arise from differences in the geological models. In the area of Sonnenborg et al. (2016) the thickness of the Miocene sand units decreases towards west and disappears

before reaching the west coast. Sonnenborg et al. (2016) conclude that rivers control the age distribution even in deep aquifers. Based on particle tracking they found that the flow regimes were dominated by local and intermediate flow (cf. Tóth, 1963) with flow lengths not exceeding 15 km. In contrast, in the study presented here, the Miocene sand extends to the coast and probably beyond. While the age pattern in the shallow aquifers is controlled by rivers and drains (similarly to Sonnenborg et al., 2016), the age pattern in the deep aquifers is dominated by the extend and thickness of the Maade formation, the Marsh

area as a location of preferred discharge and the occurrence of buried valleys as locations of groundwater exchange, especially upwelling of old groundwater. Particle path lengths reach up to 30 km and regional flow dominates in the Miocene aquifer.

### 5.3   Perspectives of using spatial and temporal groundwater age distributions in groundwater quantity and quality assessment and management

The groundwater age distribution in aquifers is closely related to the distribution of physical (e.g. to hydraulic conductivity

and porosity) and chemical parameters (concentrations of contaminants and natural geogenic elements) of the aquifers and aquitards. Hence, tracer and model estimated groundwater age distributions provide important information for the assessment of the hydraulic properties of the subsurface as demonstrated in this study, and as an indicator of groundwater quality and vulnerability (Hinsby et al. 2001a; Sonnenborg et al. 2016) including contaminant migration (Hinsby et al. 2001a), contents of harmful geogenic elements such as Arsenic and Molybdenum (Edmunds and Smedley 2000; Smedley and Kinniburgh 2002,



2017) and the risk of saltwater intrusion (MacDonald et al. 2016; Larsen et al. 2017; Meyer et al. 2018b). Groundwater age distributions in time and space are therefore important information for groundwater status assessment and the development of proper water management strategies that consider and protect both water resources quality and quantity (MacDonald et al. 2016). Water quality issues are often related to human activities such as contamination or overabstraction (MacDonald et al.

2016) and is typically found in waters younger than 100 years to depth of about 100 m (Seiler and Lindner 1995; Hinsby et al. 2001a) although deep subsurface activities may threaten deeper and older resources (Harkness et al. 2017). Deeper and older water is generally not contaminated or affected by human activities, but the impact of natural processes and contents of dissolved trace elements increases with depth and transport times (Edmunds and Smedley 2000). Similarly, the risk of salt water intrusion from fossil seawater in old marine sediments increase with depth in inland aquifers and reduce the amount of

available high quality groundwater resources (MacDonald et al. 2016; Larsen et al. 2017; Meyer et al. 2018b). Furthermore, old groundwater resources which are only slowly replenished are more vulnerable to over-exploitation, which lead to declining water tables, increasing hydraulic gradients and long-term non-steady state conditions that change the regional flow pattern (Seiler and Lindner 1995) and potentially result in contamination of deeper groundwater resources by shallow groundwater leaking downward. The presented modelling results show that the Miocene sand aquifer is protected by the overlying Maade

formation over a wide area. The aquifer bears old waters (>100 a, cf. Figures 9e,f) of high quality (Hinsby and Rasmussen 2008), especially in the east and the central part of the area, as the risk of seawater intrusion increase towards the west. However, caution should be shown as the shallow and the deep aquifers are naturally connected through buried valleys, where groundwater exchange occurs in both direction (Meyer et al. 2018a). In these geological features, young and possibly contaminated water can be found to greater depth (Seifert et al. 2008, Figure 9e). Moreover, deep, old waters are vulnerable to

contamination by modern pollutants as a result of the construction of wells with long screens, connecting different aquifers separated by aquitards (Seiler and Lindner 1995; Jasechko et al. 2017, Figure 2).

## 6 Conclusions

The originality of this study comes from a 3D multi-layer coastal regional advective transport model, where heterogeneities are resolved on a grid scale. The distributed effective porosity field was found by parameter estimation based on apparent ages

determined from $^{14}C$ activities, corrected for dissolution and diffusion. Based on regularized inversion seven porosities were estimated. Four of these were found to have high identifiability indicating that they are well constrained by the age data. The remaining three have moderate to low identifiability implying that they are less or poorly constrained by the data. In the latter case, parameter estimates close to the preferred values were obtained because of the use of Tikhonov regularization. By using a distributed effective porosity field, it was possible to match the observed age data significantly better than if porosity was

assumed to be homogeneous and represented by a single value.

The advective age distributions at the well locations show a wide range of ages from few hundreds to several thousand years. Younger waters show narrower unimodal age distribution with small variances while older waters have wide age distributions, often multi-modal with large variances. The variances in age distribution reflect the spatial heterogeneity encountered by the



groundwater when travelling from the recharge location to the sampling point.

The estimated porosity field was subsequently applied in a direct age simulation that provided insight into the 3D ground-water age pattern in a regional multi-layered aquifer system and the probable advance of modern potentially contaminated
5   groundwater. Large areas in the shallow Pleistocene aquifer is dominated by young recharging groundwater ($< 200\,\mathrm{a}$) while older water is upwelling into rivers and drains in the marsh area. Hence, the upper aquifer is prone to contamination. In large areas the deeper Miocene aquifer is separated and protected by the Maade formation bearing old water, whereas young and possibly contaminated water is located in the recharge area in the East and in the buried valleys where the shallow and deep aquifer systems are shortcut.

The study clearly demonstrate the governing effect of the highly complex geological architecture of the aquifer system on the age pattern. Even though there are multiple uncertainties and assumptions related to groundwater age and its use in calibration, the results demonstrate that it is possible to estimate transport parameters that contain valuable information for assessment of groundwater quantity and quality issues. This can be used in groundwater management problems in general, as demonstrated
15   in an example of capture zone delineation where a heterogeneous porosity field resulted in a $50\%$ change in the capture zone area compared to the case of homogeneous porosity. The adopted approach is easy to implement even in large-scale models where auto-calibration of transport parameters using models based on the advection-dispersion equation might be restricted by computer run time.

*Data availability.* The data is available from the authors.

20   *Competing interests.* No competing interests are present

*Acknowledgements.* This study stems from the SaltCoast project generously funded by GeoCenter Denmark. The authors extend sincere thanks to all individuals and institutions whose collaboration and support at various stages facilitated completion of this study.





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
