# Peer review of "Estimation of effective porosity in large-scale groundwater models by combining particle tracking, auto-calibration and $^{14}$C dating"

_Hydrology and Earth System Sciences, 2018_

## Short Comment (SC1) · 5 Apr 2018

Joost Delsman

Gualbert Oude Essink

Joeri van Engelen

Deltares, Subsurface and Groundwater Unit and Utrecht University, The Netherlands

Meyer et al describe the estimation of the porosity parameter in a steady-state groundwater flow model of a coastal region in Denmark, using 14C dated groundwater samples as a calibration target. We find this a very interesting work and welcome the effort

to calibrate groundwater models on targets other than groundwater heads. Such efforts are crucial to improve our understanding of groundwater flow in coastal environments. However, their paper did spark two comments we could not resist to raise.

1. The groundwater flow system cannot be assumed stationary over the timescales considered Meyer et al use a stationairy groundwater flow model to calculate the age of groundwater at measurement locations, and compare this with corrected 14C dates at these locations. Their model represents the present-day groundwater system, and is forced with present-day boundary conditions. However, the historical trajectory of the measured water droplets has likely been far more complex than assumed in the stationary flow model. Sea level changes, shifting of coastlines, marine transgressions and subsequent infiltration of sea water, drainage of arable land, land subsidence, development of well fields all significantly alter groundwater flow patterns over the timescales considered. See e.g. Delsman et al. (2014) in HESS (sorry to cite our own work), where we show – in a very similar hydrogeological setting – massive changes in groundwater flow patterns occurring over millennia, and even over the very last decades.

The authors do acknowledge the non-stationarity of groundwater flow patterns at larger timescales, by discarding all samples over 5 pMC of activity. But that still leaves samples with a corrected age of 1800 years in the data set, a timeframe in which a lot can happen. For example, as described by Meyer et al., "low-lying marsh areas (with elevations below mean sea level) in the west were reclaimed from the Wadden Sea." With profound effects on groundwater flow patterns: "the large drainage network, established in the reclaimed terrain keeping the groundwater table constantly below the sea level, acts as a large sink for the entire area." And while this dominant flow-defining feature has only been present for the last 200 years, the analysis presented by Meyer et al assumes the present-day groundwater flow pattern to have existed for at least 1800 years. Furthermore, the North sea level has risen about 2 m over the past 1800 years (Van de Plassche, 1982). Given the very shallow local bathymetry , the coastline

of 1800 years ago may have been located as far as 25 km westward of its present-day position. Such significant changes should in our opinion be accounted for (for instance by paleo-hydrogeological modeling) before attempting to use age data as a calibration target.

2. Density effects may significantly affect groundwater flow patterns and should be addressed Our second point concerns the use of a constant-density groundwater flow model in the analysis. In this specific coastal groundwater system, saline groundwater has clearly been detected from an airborne electro-magnetic survey (Støvring Harbo, 2011; Jørgensen et al., 2012). This means density variations will significantly affect groundwater flow patterns and should have been addressed in the analysis, e.g. by using the computer code SEAWAT. Correcting the seaward boundary for density effects will unfortunately do little to improve modeled inland flow patterns (and hence calculated age distributions) affected by density variations. Simmons (2005) has a nice way of showing the importance of density variations, by equating a typical head gradient of 10-3 to the density effect caused by a density difference of only 1 kg/m3 (5% seawater). In addition, we wonder if the seaward boundary condition accurately represents the connection of the groundwater flow system to the groundwater flow system underlying the North Sea. The boundary condition is located directly next to the system of interest, and seems to be applied without taking into account the likely seaward extension of the clay layers that are depicted in Figure 2.

Therefore, given the issues outlined above, we wonder if the conceptualization of the groundwater flow model used by Meyer et al is indeed sufficient to accurately model groundwater age, and if the obtained results are not merely a case of "The right result, for the wrong reasons" (Beven, 1993).

References

Beven, K., 1993. Prophecy, reality and uncertainty in distributed hydrological modelling. Adv. Water Resour. 16, 41–51.

Delsman, J.R., Hu-a-ng, K.R.M., Vos, P.C., De Louw, P.G.B., Oude Essink, G.H.P., Stuyfzand, P.J., Bierkens, M.F.P., 2014. Paleo-modeling of coastal saltwater intrusion during the Holocene: an application to the Netherlands. Hydrol. Earth Syst. Sci. 18, 3891–3905.

Jørgensen, F., Scheer, W., Thomsen, S., Sonnenborg, T.O., Hinsby, K., Wiederhold, H., Schamper, C., Burschil, T., Roth, B., Kirsch, R., Auken, E., 2012. Transboundary geophysical mapping of geological elements and salinity distribution critical for the assessment of future sea water intrusion in response to sea level rise. Hydrol. Earth Syst. Sci. 16, 1845–1862.

Post, V.E.A., Vandenbohede, A., Werner, A.D., Maimun, Teubner, M.D., 2013. Groundwater ages in coastal aquifers. Adv. Water Resour. 57, 1–11.

Simmons, C.T., 2005. Variable density groundwater flow: From current challenges to future possibilities. Hydrogeol. J. 13, 116–119.

Støvring Harbo, M., Pedersen, J., Johnsen, R., & Petersen, K. (2011). Groundwater in a Future Climate: The CLIWAT Handbook Interreg IV B (North Sea Region Programme), http://cliwat.eu/xpdf/groundwater_in_a_future_climate.pdf.

Van de Plassche, O., 1982. Sea-level change and water-level movements in the Netherlands during the Holocene (PhD thesis). Faculty of Earth Sciences, VU University Amsterdam.
* * *

---

## Referee Comment (RC1) · Timothy Ginn (Referee) · 20 Apr 2018

General comments

The paper details a significant modeling effort demonstrating the importance of carbon-14 dating in the calibration of spatially-distributed porosity. The study utilizes a previously calibrated 3D groundwater flow model of the site and selects 11 of 18 carbon-14 data as targets. I have two major concerns and several other concerns about the implementation of the inverse method and the conceptualization of the apparent ages. The latter are detailed in the specific comments and the former are: 1. the model assumes the conductivity field inherited from the (unpublished, at the time of this review) Meyer

et al. (2018a, and b which is in preparation); and 2. the data are prefiltered (e.g., eliminated) based on their coherence with the inherited model prior to the analyses. While I highly respect the authors' work in the field and I believe this work has a substantive contribution in the rarely touched world of porosity estimation, I think there are important elements that require consideration and careful address in the discussion. Details of my concerns follow.

The very significant reliance on the unpublished groundwater flow model, and its fixed hydraulic conductivities, raises concerns about the current study. The current study seeks to identify porosities of 7 geological units by fitting them so that the mean ("direct") ages match the apparent ages from carbon-14 corrected for dissolution and diffusion; however, there is no allowance for departures from the originally calibrated conductivities (from the unpublished Meyer et al. 2018a). Thus the porosities are treated as if they are independent of the hydraulic conductivities. This is not conventional and disagrees with current understanding of the properties of natural porous media, and needs to be addressed by the authors.

Multiple aspects of the inversion done in Meyer et al. are important here since that work laid the foundation flow model; for instance, the vertical anisotropy factors assigned from that work are 25 for sand and 85 for clay units, which are quite high, and qualitatively at least would seem to restrict vertical migration of water in a way that would definitely affect age.

A more robust approach would have been to do a wholistic inversion, where the conductivity (and other flow and transport parameters) were calibrated at the same time as the porosity (and other transport parameters, including the dispersivity, set to zero here based on a brief local sensitivity), to the collective head and apparent age data. Why this is not done, and the potential constraints on the resulting two-stage inverse, should be discussed. There are no error plots from the prior head-inversion of Meyer et al so the success of the calibration of the flow equation is unknown. More importantly for a subsquent inversion for porosity, there is no indication of the uniqueness of that

first inversion. Even if that inversion gave good results, it may be nonunique, and it seems that there may be a different set of hydraulic conductivities and porosities which together might fit both the available head and carbon-14 data.

The elimination of dispersivity appears not only somewhat arbitrary but also contradictory to the authors' overall argument for the importance of porosity (cf. specific comment on page 8 line 21). It appears they have replaced the modeling of mobile-immobile domain mass transfer in the model with the approximate diffusion-correction applied to the data. This could be justified based on pragmatic grounds but the discussion in this regard is lacking. The alternative to use effective mobile-immobile domain mass transfer seems potentially useful and pragmatic as well but is not discussed.

Very important is the unsupported elimination of 7 pesky carbon-14 data (cf specific comment on page 8 line 30). The focus only on the data which are consistent with the already partly calibrated model brings the entire study into question.

Why the recently developed methods for full distribution of age (e.g., several articles in J Hydrology, December 2016) are not used is not described; however, this may be attributed to the reliance on single radiometric tracer (carbon-14) concentration measurements, which precludes any inferrence of age distribution.

Specific comments.

page 2 line 4. "Three different apporaches with specific benefits and disadvantages are commonly applied to simulate groundwater age..." The given list of commonly-used methods is not complete (there are also the lumped-parameter approach, and the mixing cell model approach), and equally important are the new methods which are generally more robust [solving the actual equation of groundwater age, either by the Laplace method of Cornaton (WRR 2012) or by using reduced dimensions as in Woolfenden and Ginn (Groundwater, 2009)]. The review by Turnadge and Smerdon (J Hydrology 2014) provides a more complete listing and assessment.

page 2 line 12. "A comparison of ages simulated using any of these methods with ages determined from tracer observations, referred to as apparent ages is desireable..." This is true but omits the very important point that "ages determined from tracer observations" are not equal to mean ages, especially as in the present case of decaying environmental tracers (e.g., carbon-14). The rest of this paragraph summarizes part of the way that "apparent ages" are misled by old carbonate dissolution, by diffusion, and by heterogeneity, following McCallum's work; however, it should also point out the fundamental difference between mean ages and radiometric ages described explicitly by equation 16 of Varni and Carerra (WRR 1998), and the general relation between distribution of age and the radiometric age given in Massoudieh and Ginn (WRR 2011).

page 2 line 238. "Bethke and Johnson (2002) concldued that the groundwater age exchange... is only a function of the volume of stored water." This is misleading because it is valid only for the mean groundwater age, and requires steady-state as detailed in Ginn et al. (Tranpsort in Porous Media, 2009). Also this point is made earlier and more precisely in Varni and Carerra (op. cit., page 3272), who points out that it is actually a result of Haggerty. The overall point by the authors that porosity is important to age modeling is valid.

page 3 line 1. "neglecting dispersion effects seemed to be acceptable at large scale" is unsupported for the present application, results of cited Sanford and later Gelhar notwithstanding. See comments below (re: page 8 line 21 and the reliance on Sanford; page 10 lines 14-17 and Figure 8) for more discussion.

page 6 line 27. "Meyer et al. (2018b) simulated ....further details can be found in Meyer et al. (2018a)." Actually they cannot because Meyer et al. (2018a) is in submitted state (page 30 line 28). This is quite important because the present authors have chosen to rely upon the hydraulic conductivity field previously calibrated in that work, and here do not allow the conductivity values to be modified in the inversion using carbon-14 inferred ages (page 8 line 26).

page 8 line 2. "The resulting head distribution is shown in Figure 1." Figure 1 shows (it seems to me) only the shallow aquifer heads. It is well-known that the quality of an inversion of the flow equation (to determine hydraulic conductivities) depends on a broad spatial distribution of the heads, and it is unclear that such head data were available to Meyer et al. Also, there are no error plots showing the goodness-of-fit of the flow inversion to the measured heads, so it is impossible for the reader to evaluate how good was the flow equation inversion. Also it is impossible for the reader to evaluate the uniqueness of the flow equation inversion, which is commonly very poor.

page 8 line 21 "According to Sanford (2011), neglecting hydrodynamic dispersion... on a regional scale is a reasonable approach when old-age tracers, such as carbon-14, are used as dispersion might not be crucial for these tracers." This sentiment is unclear because it suggests that there is something particular to the carbon molecule that frees it from dispersion, which is quite incorrect. It is also directly in opposition with the authors' claim (page 2 line 28ff) that porosity is important for groundwater mean age determination because "groundwater age exchange between flow and stagnant zones is only a function of the volume of stored water."

page 8 line 30ff. The authors removed 7 data from their 18 carbon-14 measurements becaue the values did not match their conceptual model; 6 were deleted because the carbon-14 activities were below 5pmc, and one due to proximity to another sample with different value. The justification given for the first 6 is "it was assumed that the boundary conditions of the flow model ... were not representative for pre-Holocene conditions." This justification is unclear at best; the model is steady state so the initial conditions do not matter, and the boundary conditions are necessarily (by the steady-state assumption) constant. Thus the elimination of the low carbon data is unsupported. The elimination of the 7th datum is only weakly justified, as there appears to be nothing wrong with it other than its troubling value.

page 9 line 4-6. The weights on the data used in the inversion were all the same. They were based on an average uncertainly of apparent ages of ∼102 years, as per "average

of the standard deviation of the diffusion correction for the selected 11 samples..." This defeats the purpose of calculating individual standard deviations for individual data in the first place. The individual standard deviations (Table 1, last column) show a range of 8 to 310 years, so individual weights based on these values would have led to significantly different weights. Individualized weighting is rarely possible in groundwater flow model inversion but is often possible in transport inversion, and it seems to me that the authors have unintentionally limited the inversion by assigning equal weights to all apparent age data. The importance and utility of weighting is amply described in the books by John Doherty and Mary Hill, and could have been used to condition the data per their individual certainties; moreover it could have been used to condition - perhaps to good end - the pesky 7 data that were eliminated instead. In fact, the standard deviations of the 6 eliminated data range from 1323 to 2593 years, which would have led to quite significant reduction in the importance of these data as the weights are generally taken as the reciprocals.

page 9 line 27. "mean groundwater age is simulated in analogy to solute transport as an "age mass" (Bethke and Johnson 2008)." This "age mass" requires mathematical and physical definition; as pointed out in Ginn et al (2009, op. cit., section 2.2) the definitions of Goode and of Bethke and Johnson are not clear or consistent. The example of Bethke and Johnson involves an aquifer and an aquiclude with only immobile water, so that diffusion is the only mechanism by which exchange can take place. If it is eliminated, then the argument collapses.

page 10 lines 14-17 The numerical experiments to evaluate dispersion effects, described here, with results summarized on page 15 lines 10ff and in Figure 8, are apparently done on one model, that is, on one assignment of hydraulic conductivities and porosities. It is not clear which porosities were used. In any case, this is at best a local parameter sensitivity analysis and it would be more accurate to include the dispersivity values in the inversion. The argument that the 200mx200m grid cell size is sufficiently resolved to allow ignoring dispersion is unconvincing, because there are multiple modeling exercises where the effective dispersivity is proportional to the grid cell size, not zero. Figure 8 does not tell how the errors grew but only the total error - did the errors go biased ? If one were to guess, I would bet they did, because the dispersion would allow mass transfer laterally, causing generally older ages.

page 11 line 6 "...as porosity does not impact the trajectory of the particle path..." this is true only via the assumption that the porosity and hydraulic conductivity are independent, which is not common.

page 13 Figure 4a. The plot demonstrates in my view limited improvement for two reasons. First, the 5 older water samples (with carbon-14 corrected ages greater than 500 years) show significantly improved fitting in 3 cases, with one getting worse. Second, the plot is absent of confidence intervals (compare for instance to Figure 11) which could be it seems to me estimated based on the standard deviations of the corrected carbon-14 ages (Table 1), with additional uncertainty based on equation 16 of Varni and Carrera (op. cit.). The recognized uncertainty in the apparent ages should it seems be used to condition the results of Figure 4a.

page 16 line 4ff "Hence, the dispersivity only describes the effect of heterogeneity at the grid scale, 200m. In accordance with Gelhar et al. (1992) this results in (dispersivity) with a magnitude of a few meters." I am unaware that Gelhar suggested this dispersivity value given (only) the size of the grid, please provide the page. Also in the intervening 25 years there has been extensive research and articles on the effective dispersivity for regional groundwater models, and more up to date referencing is called for. Notably, the model (including its effective parameters) at the 200m grid block scale tells only the expected or mean concentration in the grid block, that is, the concentration in the model is treated as a constant on the 200m x 200m x 5m grid block, while the carbon-14 data are collected from sampling wells on much smaller spatial scales - this issues should also be addressed or at least noted.

page 17 line 1. "The age distribution is strongly affected by geology and is therefore in

good agreement with the interpretation of the flow system by Meyer et al. (2018)." This statement is unclear: the age distribution is always strongly affected by geology.

Figure 10 caption "Normalized probability distributions..." These are frequency distributions becasue there is no randomness in the model or its parameters.

page 21 line 12 (regarding Figure 11) "However, most of them lie within one standard deviation." Seven of the standard deviations here span several thousands of years while the means for all but one are less than 7000 years, so this is not a comforting result.

page 23 section 5.1.2. This discussion clearly identifies the ways that individual particle path history of exposure to different geologic units differentiates the actual true correction of the carbon-14 from the simplified correction done in the paper; however, it still does not tell about the fundamental difference between the apparent age and the mean age (cf. comment on page 2 line 12). That is, even if the correction were perfect, the apparent age would not equal the mean age.

page 24 line 6. "While direct age corresponds to the flux-averaged mean, the particle tracking age is resident-averaged (Varni and Carrera, 1998)." I do not see where this statement is given in the cited reference, please clarify if so; furthermore, I do not believe the statement is correct. The mean age of the model of Goode is an Eulerian quantity, just like a solute resident concentration. The relation between resident and flux-averaged concentrations is given in a number of papers by Parker and van Genuchten and coworkers (1984) but the governing equations that result are mainly restricted to 1D cases.

page 24 line 9. The use of harmonic mean for particle ages is absent of a rational basis other than it seems to fit the data well, and a generic reference to Konikow (2008). The specific manner of averaging the particle ages should be physically-based and independent of how well it fits the data in a particular setting.

References

Cornaton, F. J. (2012), Transient water age distributions in environmental flow systems: The time‐marching Laplace transform solution technique, Water Resour. Res., 48, W03524, doi:10.1029/2011WR010606.

Ginn, T. R., H. Haeri, L. Foglia, and A. Massoudieh (2009), Notes on groundwater age in forward and inverse modeling, Transport in Porous Media,79:117-134.

Massoudieh, A., and T. R. Ginn (2011), The theoretical relation between unstable solutes and groundwater age, Water Resour. Res., 47, W10523, doi:10.1029/2010WR010039.

Parker, J. and M. Th. van Genuchten (1984), Flux-Averaged and Volume-Averaged Concentrations in Continuum Approaches to Solute Transport, Water Resour. Res., 20(7):866-872.

Turnadge, C., and B. D. Smerdon (2014), A review of methods for modelling environmental tracers in groundwater: Advantages of tracer concentration simulation, Journal of Hydrology 519, 3674-3689.

Woolfenden, L., and T. R. Ginn (2009), Modeled ground water age distributions, Ground Water, 47(4), 547–557.

Varni, M., and J. Carrera (1998), Simulation of groundwater age distributions, Water Resour. Res., 34, 3271–3282, doi:10.1029/98WR02536.
* * *

---

## Referee Comment (RC2) · Anonymous Referee #2 · 28 Jun 2018

(1) Scientific significance. The paper presents a case study in which inferences about the regional distribution of groundwater travel times are based on 18 measurements of 14C at 7 locations. In addition to the measurements, an existing groundwater flow model and a voxel-based geologic model were available and used. Only porosity (in 7 zones) was optimized, using the existing flow model with advection-only particle tracking.

The resulting porosity field was used in a direct age simulation to generate the mean travel time distribution throughout the aquifer system. The distribution of travel times was explained in the context of the geologic structure of the system, which in turn was

extended to a discussion of the general vulnerability of the system to various forms of contamination (natural, sea water intrusion, and anthropogenic).

The paper largely uses concepts and methods that are well known. Groundwater models have been calibrated to travel times (many examples). One aspect of the paper that is not well represented in the literature is the sequential calibration of an existing flow model to travel times using only porosity, but this this too has been used before, as for example in Starn, J.J., C.T. Green, S.R. Hinkle, A.C. Bagtzoglou, and B.J. Stolp. 2014. Simulating water-quality trends in public-supply wells in transient flow systems. Groundwater 52(S1): 53-62.

(2) Scientific quality. The methods and analyses are sound. The discussion of travel times in the context of the geology is especially good. Although the researchers reach a different conclusion than another study in the same geographic area, the differences are explained well and make good sense. Once the relation of travel time and geology was established (in this paper), the geologic voxel model was used to make broad statements about the susceptibility of groundwaters in the area. The paper is a good example of using relatively few data points, along with existing data, in a thoughtful way that should enhance proper management of the resource.

(3) Presentation quality. The graphs and tables could easily be made clearer. Suggestions on how to do that are included in an attached PDF document.

Please also note the supplement to this comment:
https://www.hydrol-earth-syst-sci-discuss.net/hess-2018-99/hess-2018-99-RC2-supplement.pdf

**Supplement:**

[revised manuscript text omitted]

---

## Author Comment (AC2) · 18 Jul 2018

*(1) Scientific significance. The paper presents a case study in which inferences about the regional distribution of groundwater travel times are based on 18 measurements of 14C at 7 locations. In addition to the measurements, an existing groundwater flow model and a voxel-based geologic model were available and used. Only porosity (in 7 zones) was optimized, using the existing flow model with advection-only particle tracking. The resulting porosity field was used in a direct age simulation to generate the mean travel time distribution throughout the aquifer system. The distribution of travel times was explained in the context of the geologic structure of the system, which in turn was extended to a discussion of the general vulnerability of the system to various forms of contamination (natural, sea water intrusion, and anthropogenic).The paper largely uses concepts and methods that are well known. Groundwater models have been calibrated to travel times (many examples). One aspect of the paper that is not well represented in the literature is the sequential calibration of an existing flow model to travel times using only porosity, but this this too has been used before, as for example in Starn, J.J., C.T. Green, S.R. Hinkle, A.C. Bagtzoglou, and B.J. Stolp. 2014. Simulating water-quality trends in public-supply wells in transient flow systems. Groundwater 52(S1): 53-62.*

*(2) Scientific quality. The methods and analyses are sound. The discussion of travel times in the context of the geology is especially good. Although the researchers reach a different conclusion than another study in the same geographic area, the differences are explained well and make good sense. Once the relation of travel time and geology was established (in this paper), the geologic voxel model was used to make broad statements about the susceptibility of groundwaters in the area. The paper is a good example of using relatively few data points, along with existing data, in a thoughtful way that should enhance proper management of the resource.*

*(3) Presentation quality. The graphs and tables could easily be made clearer. Suggestions on how to do that are included in an attached PDF document. Please also note the supplement to this comment:*

*https://www.hydrol-earth-syst-sci-discuss.net/hess-2018-99/hess-2018-99-RC2-*

*supplement.pdf*
*Interactive comment on Hydrol. Earth Syst. Sci. Discuss., https://doi.org/10.5194/hess-2018-99, 2018.*

Specific comments:

**AC: Starn et al. (2014) is added to the introduction**

Page 2:

*RC#1: It would have been nice to include a sensitivity analysis – how sensitive are travel times to changes in porosity.*

**AC#1: We did not include a formal sensitivity analysis, but instead we show the effect of a distributed effective porosity field compared to a uniform one on a capture zone delineation application, which shows a significant change based on the effective porosity.**

Page 3:

*RC#2: Seems redundant.*

**AC#2: The sentence is important as it describes one dominant feature, a man-made drainage system that lowers the water table below sea level, disturbs the "natural" flow system by enhancing the inflow of "young" ocean water.**

Page 4:

*RC#3: what method? Should be stated here*

**AC#3: we add the reference to Goode (1996).**

*RC#4: which equation is this? 2?*

**AC#4: Yes, we add the reference to equation 2.**

*RC#5: make a brief statement about the steady state assumption – over what time period; what is the evidence for steady state?*

***AC#5: The system, close to the coast, is not expected to be in steady-state over a very long period. Changes in sea level over the last thousands of years and human activity (drains and dikes) over the last centuries changed the hydraulic system, especially in the west, close to the sea. While upstream, to the east, where***

*most of our samples were taken, the system was more steady over the last thousands of years. We included a discussion about effects of transient conditions (see also our response to the short comment).*

*RC#6: what physical features do these boundaries correspond to?*

**AC#6: The physical features are delineated along flow lines and watershed boundaries; we add this to the description:**

**"No-flow boundaries were used along flow lines in the north and south and the water divide in the east and at the bottom, where the Palaeogene clay constitutes the base of the aquifer system."**

*RC#7: What about the assessment of the existing model calibration? Could it be mentioned briefly here so the reader knows how good the model is?*

**AC#7: We add the calibration results by Meyer et al. (2018) in a new figure 3 and change the sentence to:**

**"The steady-state MODFLOW flow solution (calibration results summarized in Figure 3) forms the basis for the advective transport simulation using MODPATH."**

[Figure]

**"Figure 3: Calibration results of steady-state groundwater flow model that forms the basis for the advective transport model (modified after Meyer et al. (2018a). Left: simulated versus observed hydraulic head; right: simulated versus observed stream discharge. ME=mean error, RMS=root mean square."**

*RC#8: Table 2 shows results; maybe save those for the result section.*

**AC#8: We move table 2 to the result section (section 4.2 calibration results).**

*RC#9: What is the explanation for the porosities, i.e. are they reasonable given the description of each formation. Why do the clay units have relatively small porosities? Probably the estimated porosity is an effective transport porosity; this should be noted.*

**AC#9: The reviewer is right, we are estimating effective porosities that is the reason why the estimated effective porosities for clay are relatively small. We check throughout the manuscript and specify where it is missing.**

*RC#10: or of structural error in the number and boundaries of zones chosen, boundary conditions, ect. – many more possible causes than unsimulated heterogeneity*

**AC#10: The reviewer is right. We extended the explanation of the mismatches to:**

**"…mismatches can be, e.g., a result of small scale heterogeneity below grid resolution, errors in the model structure or uncertainties of parameter."**

*RC#11: clay typically has a larger porosity than sand*

**AC#11: This is right for the total porosity. We are dealing with effective porosities (see also AC#9)**

*RC#12: does*

**AC#12: Misspelling corrected. "does"**

*RC#13: It seems that well C has several screen segments with short pathlines that should produce short travel times. It's not clear that only some results are excluded.*

**AC#13: We add screen numbers to be more precise on which wells were used for calibration.**

**"As mentioned above, only 14 C observations with an activity higher than 5pMC (Table 1) were used, which excludes results from well screens C1, C2, C3, D1, D2, F1 and F2."**

*RC#14: A little more discussion on how SV are applied and interpreted here.*

**AC#14:**

**SVD operates on the sensitivity matrix, the Jacobian that relates parameters to observations, and divides the parameter space into a solution space and a null-space. Hereby parameters that are informed by the**

observations are put in the solution space while those not informed by observations fall in the null-space. The truncation between these spaces is user-defined and should be at a level where observations do not further constrain parameters. The advantage of using SVD is that the number of estimated parameters is reduced and hence the inverse problem is well-posed. If too many singular values are included, the problem will be still ill-posed. If too few, the model fit might be unnecessarily poor. Singular values are ordered in a decreasing manner, meaning that a singular value of index 1 is more constrained by information contained in the observations than a singular value of index 2 (Anderson et al., 2015). In our study we truncated the SV at index 5 and in Figure 4 b) the identifiability of the parameters based on the SV index 5 is shown. For more details on singular value decomposition refer to, e.g., Anderson et al. (2015), Doherty and Hunt (2009) or Doherty (2015).

*RC#15: does this mean that estimated porosities for clay units are not reliable?*

**AC#15: The reviewer is right, that the estimated effective porosities with a higher identifiability are more reliable, because they are constrained by the observations, compared to those with a small identifiability. However, it does not necessarily mean that the estimates with a low identifiability are unreliable. They are rather more dependent, or constrained, on the regularization and hence on the expert knowledge than by the observations.**

*RC#16: Consider color-coding the well designations on FIgs 5 and 6 and Table 3. This will make it easier to compare the information on each of these.*

**AC#16: We have considered color coding as suggested, but we think it is more confusing. The well screens are all numbered throughout the figures and the tables, which allow comparison easily.**

*RC#17: One problem with this type of plot is that some of the data are always obscured. Consider plotting each subplot on one or more 2D graphs.*

**AC#17: We changed the graph to a 2D normalized frequency distribution based on the former histograms.**

[Figure]

**Figure 7 (before 6): Particle age distributions at sampling wells A-G (see Figure 1 for locations). a) and b) young waters (bin size = 50 years) show a narrow, unimodal distribution; c) old waters (bin size = 500 years) have broader and often multimodal distributions; d) multi-modal age distribution at sample location D1 (bin size = 1000 years), which shows the longest travel times.**

*RC#18: Consider shading as Table 2 to show which samples were used in model calibration.*

**AC#18: We changed the shading as suggested.**

*RC#19: mean [this relation does not hold for the median]*

**AC#19: We specified the description to:**

**"The younger waters (mean age <1000 years) …"**

*RC#20: Consider using horizontal and vertical lines to show your thresholds of 1000 years and 10 and 20km path lengths.*

**AC#20: We have considered this. But if we do so, these lines should be on each subfigure. The axes of the subfigures are chosen to show best the distribution of the data. For some of the subfigures (wells A, B, F, G) these lines would be outside the figure, therefore we choose not to add these lines.**

*RC#21: If you use alpha=0, you could have used particle tracking. This would avoid the complication of numerical dispersion and would allow you to talk about higher moments of the travel time distribution.*

**AC#21: We set alpha = 0 because the physical dispersion which we still have probably in a range of a few meters is overruled by numerical dispersion (see also AC#14 to comments by reviewer 1). We used particle tracking for the calibration at the sampled well location. But in order to get an idea of the age distribution in the entire model (1.2 mio cells) it was not feasible to produce ravel time distributions of 1000 particles in each cell (as we did for the cells where we had tracer samples). This is why we chose the direct mean age simulation to visualize the age structure in the entire aquifer system.**

*RC#22: Be clear this is mean age here and elsewhere in this section. Also, consider use the term travel time instead of age*

**AC#22: We checked the consistency and added 'mean' when it was missing. We considered using the term 'travel times'. To preserve the comparability between particle age and tracer-based apparent ages we chose the term 'mean age' instead of travel time.**

*RC#23: Consider explicitly explaining why section e and f are different at the western boundary.*

**AC#23: We thank for this remark and add a detailed explanation of the two cross-sections and their differences.**

**"The two cross-sections e ) and f) (Figure 10) differ in their geological connection to the sea-boundary (compare geological sections g) and h) Figure 10). In e) a buried valley connects the inland aquifer with the sea and here younger waters reach further inland due the relatively higher hydraulic conductivity and the inland head gradient as a result of the drainage system. Moreover, buried valleys constitute locations where the deep aquifer system, bearing old waters, connects with the shallow one and here upwelling of older water occurs due to the higher heads in the deep semi-confined (by the Maade aquitard) Miocene aquifer. In cross-section f) where the buried valley occurs further inland, the young ocean water penetrates the higher**

**permeable Miocene aquifer but is impeded in the low permeable sections and hence does not reach as far inland."**

*RC#24: Be consistent with color schemes across all figures – that help the reader understand your points easily. Considering a 1000 year line instead of 100 because 1000 years is used in the discussion.*

**AC#24: We chose to have a different color scale on a) in order to better resolve the younger ages close to the surface. In order to prevent misinterpretation we add an explanation to the caption of the figure.**

**"Be aware that the color scheme in a) is different in order to better resolve younger ages close to the surface."**

**We chose the 100 year line because this is approx. the time span over which human activity (e.g. contamination with fertilizers) heavily started and contaminated groundwater might be expected. This is on what we base our interpretation and discussion groundwater quality and quantity issues on (section 5.3.)**

*RC#25: It would be worth noting that regardless of human actions, stresses have not been steady over that period, either climate, sea level, or within earth's crust. If it takes that long to reach equilibrium under steady stress, the system is never in steady state.*

**AC#25: We thank for this remark. The reviewer is right, the system is over this period never in steady state. A similar remark was given in the short comment. We add the sentence here and further discuss this in the discussion.**

**"The steady state distribution of direct simulated mean groundwater age was reached after ~26000 years. Over this time span the system has been exposed to transient stresses from human activity, climatic changes (glacial cover, sea level, ect.). Therefore, the steady-state assumption is a notable simplification, which is further discussed in section 5.1."**

*RC#26: Review the porosity of Maade and how it was determined.*

**AC#26: The porosity of the Maade was estimated as 'Pleistocene clay (Maade formation)' e.g. Table 2 or Figure 4 (the new Figure 5).**

*RC#27: That seems to be older than what the pdf indicates.*

**AC#27: We have checked the mean groundwater ages derived from a moment analysis and the shown pdfs again. They are correct.**

*RC#28: The direct ages are a function of the age mass of the volume of the model cell whereas the advective ages are a function of the well screen position within the cell. You wouldn't necessarily expect them to match.*

**AC#28: We add a section about the commensurability to the discussion. Here we discuss the differences in observed tracer ages, particle-based simulated ages and directly simulated ages.**

**"The comparison of groundwater ages, estimated from tracer concentration in a water sample, and simulated groundwater ages, either derived by particle tracking or direct age modelling, bears the problem of commensurability, the comparison of a point measurement relative to the modelling scale. The water sample represents the age distribution in the direct surrounding of the well screen which only makes up a few percent of the water in one model cell.**

**The differences between mean advective ages and directly simulated mean ages as described in section 4.4 can be related to the simulation methods. While particle tracking neglects dispersion, but allows simulating an age distribution in a cell (by perturbing the measurement location so to speak), direct age modelling allows to account for dispersion/diffusion, resulting in only the mean age at a cell. The mismatches between advective and direct age can be related to the diffusion and dispersion processes (here represented by numerical dispersion as dispersivity was set to zero), which are included in the direct age approach, but neglected in simulating advective ages."**

*RC#29: the dashed lines are not clear on these maps.*

**AC#29: We enlarged the figure, now the lines are better visible.**

*RC#30: not clear what you mean by 'general behavior of the voxel system.' Maybe this could be reworded, for example, "properties averaged over hydrogeological units".*

**AC#30: We thank the reviewer for the suggestion and changed the sentenced accordingly to:**

**"The geology is highly complex and aquitard thickness and porosity distribution change spatially over the entire region, whereas the correction terms were based on the properties averaged over hydrogeological units."**

*RC#31: Particle tracking can also be used to calculate flux-weighted residence times. The difference is how you choose to weight particles, whether by volume or by flux.*

**AC#31: To prevent confusion, also based on comments by reviewer 1,  we decided to take this part out of the manuscript (see also AC#25 and 26 to reviewer 1)**

*RC#32: You can also assign weights to particles based on flux, which would give you a more comparable age to the direct method. You still have the difference that particles placed in a well screen have limited spatial distribution compared to those in a model cell.*

**AC#32: see AC#31.**

**References added:**

Anderson, M., Woessner, W.W., Hunt, R., 2015. Applied Groundwater Modeling: Simulation of Flow and Advective Transport, 2nd ed. Elsevier. https://doi.org/10.1016/B978-0-08-091638-5.00001-8

Doherty, J., 2016. Model-Independent Parameter Estimation II. Watermark Numer. Comput. 217.

Doherty, J., 2015. Calibration and uncertainty analysis for complex environmental models. Wartermark Numerical Computing.

Doherty, J., Hunt, R.J., 2009. Two statistics for evaluating parameter identifiability and error reduction. J. Hydrol. 366, 119–127. https://doi.org/10.1016/j.jhydrol.2008.12.018

Starn, J.J., Green, C.T., Hinkle, S.R., Bagtzoglou, A.C., Stolp, B.J., 2014. Simulating water-quality trends in public-supply wells in transient flow systems. Ground Water 52, 53–62. https://doi.org/10.1111/gwat.12230

---

## Author Comment (AC3) · 18 Jul 2018

*Meyer et al describe the estimation of the porosity parameter in a steady-state groundwater flow model of a coastal region in Denmark, using 14C dated groundwater samples as a calibration target. We find this a very interesting work and welcome the effort to calibrate groundwater models on targets other than groundwater heads. Such efforts are crucial to improve our understanding of groundwater flow in coastal environments.*

*However, their paper did spark two comments we could not resist to raise.*

*1. The groundwater flow system cannot be assumed stationary over the timescales considered Meyer et al use a stationairy groundwater flow model to calculate the age of groundwater at measurement locations, and compare this with corrected 14C dates at these locations. Their model represents the present-day groundwater system, and is forced with present-day boundary conditions. However, the historical trajectory of the measured water droplets has likely been far more complex than assumed in the stationary flow model. Sea level changes, shifting of coastlines, marine transgressions and subsequent infiltration of sea water, drainage of arable land, land subsidence, development of well fields all significantly alter groundwater flow patterns over the timescales considered. See e.g. Delsman et al. (2014) in HESS (sorry to cite our own work), where we show – in a very similar hydrogeological setting – massive changes in groundwater flow patterns occurring over millennia, and even over the very last decades. The authors do acknowledge the non-stationarity of groundwater flow patterns at larger timescales, by discarding all samples over 5 pMC of activity. But that still leaves samples with a corrected age of 1800 years in the data set, a timeframe in which a lot can happen. For example, as described by Meyer et al., "low-lying marsh areas (with elevations below mean sea level) in the west were reclaimed from the Wadden Sea."*

*With profound effects on groundwater flow patterns: "the large drainage network, established in the reclaimed terrain keeping the groundwater table constantly below the sea level, acts as a large sink for the entire area." And while this dominant flow-defining feature has only been present for the last 200 years, the analysis presented by Meyer et al assumes the present-day groundwater flow pattern to have existed for at least 1800 years. Furthermore, the North sea level has risen about 2 m over the past 1800 years (Van de Plassche, 1982). Given the very shallow local bathymetry, the coastline of 1800 years ago may have been located as far as 25 km westward of its present-day position. Such significant changes should in our opinion be accounted for (for instance by paleo-hydrogeological modeling) before attempting to use age data as a calibration target.*

 *2. Density effects may significantly affect groundwater flow patterns and should be addressed Our second point concerns the use of a constant-density groundwater flow model in the analysis. In this specific coastal groundwater system, saline groundwater has clearly been detected from an airborne electro-magnetic survey (Støvring Harbo, 2011; Jørgensen et al., 2012). This means density variations will significantly affect groundwater flow patterns and should have been addressed in the analysis, e.g. by using the computer code SEAWAT. Correcting the seaward boundary for density effects will unfortunately do little to improve modeled inland flow patterns (and hence calculated age distributions) affected by density variations. Simmons (2005) has a nice way of showing the importance of density variations, by equating a typical head gradient of 10-3 to the density effect caused by a density difference of only 1 kg/m3 (5% seawater). In addition, we wonder if the seaward boundary condition accurately represents the connection of the groundwater flow system to the*

*groundwater flow system underlying the North Sea. The boundary condition is located directly next to the system of interest, and seems to be applied without taking into account the likely seaward extension of the clay layers that are depicted in Figure 2. Therefore, given the issues outlined above, we wonder if the conceptualization of the groundwater flow model used by Meyer et al is indeed sufficient to accurately model groundwater age, and if the obtained results are not merely a case of "The right result, for the wrong reasons" (Beven, 1993)*

**AC:**

**Conceptually, we agree with the comments that non-stationary boundary conditions and density effects have an influence on the age distribution. However, we would like to emphasize that the aim of our study is the calibration of porosities and the subsequent analysis of groundwater ages in the entire regional aquifer system. Our previous study** (Meyer et al., 2018b) **showed that the area outside the marsh is relatively insensitive to changes in fluxes at the coastal boundary. Our data points, except at location A and B, are located quite far away from the marsh area and, more importantly, upstream from the coastal area where the boundary conditions change. Hence, we expect relative small changes in simulated ages, at least when compared to other uncertainties of our approach that are mostly related to measurement, analysis and correcting the C-14 derived ages.**

**Still, in order to investigate the influence of density and non-stationary boundary conditions we made a few test runs with the particle tracking model and a preliminary SEAWAT model. We compare the 200 years travel length for 12 particles released in each location A1, B1, C1, D1, E1, E3 and F3 for the original case, presented in this study and in the density-dependent system (SEAWAT) (similar to the one presented in Meyer 2018, PhD thesis). The SEAWAT model accounts for density effects and simplified palaeo-hydraulic conditions (lower sea levels according to Behre (2007), absence of drainage canals) over three stages for the precedent 6000 years. The results in the table below correspond to stage 1 (oldest), where the sea level was 2 m below today's level and represent the highest changes in the particle tracking path length ("worst" case). The results indicate that the relative change in particle travel length is higher closer to the marsh area (A1 and B1) compared to the other locations further upstream. The relative differences are, however, relatively small (especially for the wells located to the east where relative differences below 10% are found) compared to the other uncertainties in our model approach. Therefore, we believe that our approach is applicable. It allows us to estimate a distributed porosity field, which of course is prone to uncertainties of different kinds.**

**We added the reference "Delsman et al. (2014)" to the discussion of uncertainties which are, among others, due to non-stationarity of the system. Moreover we have added a short discussion about the uncertainties that arises from neglecting density effects, which are mainly expected to be relevant close to the coast, but less in the upstream area. Our study highlights the importance of the effective porosity as a transport parameter and the benefit of using the 7-effective porosity model compared to the uniform one.**

**Changes made:**

"The area close to the coast is not only affected by changing sea levels during the past thousands of years, but also by saltwater intrusion. In this study, the density effects on flow were accounted for in a simplified way by using a density-corrected constant head boundary at the coast. Both, sea level changes and density effects, would also have affected the age distribution. The impact on age calculations due to density effects would be largest close to the coast. However, most of the groundwater samples used for age estimations were collected several tens of km inland and are therefore expected to be affected to a minor extent. To quantify the impact of boundary conditions and saltwater intrusion on the particle tracking, the differences of particle travel path lengths for a 200 year period, investigated based on the present model and a preliminary density-driven model (SEAWAT) accounting for non-stationary and density effects (similar to the one presented in Meyer, 2018c) are computed. The relative differences are below 10% (except at location A and B). Also, the uncertainties introduced by simplifying the density boundary effects are likely less important compared to other uncertainties associated, e.g., with estimating the groundwater age by the procedures for correcting $^{14}$C activities. A solution would, of course, be to use a fully density-driven model such as SEAWAT as in (Meyer et al., 2018a) or Delsman (2014). But, the very long computer run times for these kinds of models and the need of several thousands of model runs during calibration made it infeasible to use a variable-density flow model."

| MODPATH average particle path length [m] for 200 years | A1 | B1 | C1 | D1 | E1 | E3 | F3 |
|---|---|---|---|---|---|---|---|
| steady-state MODFLOW | 2645 | 2714 | 173 | 120 | 1215 | 5685 | 2407 |
| sea level -2m, density-dependent | 2238 | 2364 | 164 | 117 | 1235 | 5235 | 2422 |
| rel diff (%) [abs((stage1-original)/original *100)] | 15 | 12 | 5 | 3 | 2 | 8 | 1 |

Additional references

Behre, K.-E.: A new Holocene sea-level curve for the southern North Sea, Boreas, 36(1), 82–102, doi:10.1080/03009480600923386, 2007.

Delsman, J. R., Hu-a-ng, K. R. M., Vos, P. C., de Louw, P. G. B., Oude Essink, G. H. P., Stuyfzand, P. J. and Bierkens, M. F. P.: Paleo-modeling of coastal saltwater intrusion during the Holocene: an application to the Netherlands, Hydrol. Earth Syst. Sci., 18(10), 3891–3905, doi:10.5194/hess-18-3891-2014, 2014.

Meyer, R., Engesgaard, P. and Sonnenborg, T. O.: Origin and dynamics of saltwater intrusions in regional aquifers; combining 3D saltwater modelling with geophysical and

geochemical data, Water Resour. Res., submitted, 2018a.

Meyer, R., Engesgaard, P., Høyer, A.-S., Jørgensen, F., Vignoli, G. and Sonnenborg, T. O.: Regional flow in a complex coastal aquifer system : Combining voxel geological modelling with regularized calibration, J. Hydrol., 562(May), 544–563, doi:10.1016/j.jhydrol.2018.05.020, 2018b.

---

## Author Comment (AC4) · 24 Jul 2018

[revised manuscript text omitted]

20  water resources model (Henriksen et al. 2003) and included as a specified flux condition. Internal specified boundaries included abstraction wells with a total flux of $26\text{x}10^6\,m^3/year$ (averaged over the years 2000-2010, corresponding to 4% of the total recharge), rivers and drains.

Horizontal hydraulic conductivities, one for each hydrogeological unit, two anisotropy factors (Kh/Kv), one for sand and one for clay units, as well as river and drain conductances were calibrated, using a multi-objective regularized inversion scheme

25  (PEST; Doherty, 2016a), using head and mean stream flow observations as targets. The resulting head distribution is shown in Figure 1. Horizontal hydraulic conductivities were estimated in a range of $K_h \in [1\,\text{m/d};\, 83\,\text{m/d}]$ for Pleistocene sand units, $K_h \in [0.028\,\text{m/d};\, 0.19\,\text{m/d}]$ for Pleistocene clay units, $K_h \in [0.008\,\text{m/d};\, 0.016\,\text{m/d}]$ for the Maade formation, $K_h \in [16\,\text{m/d};\, 46\,\text{m/d}]$ for Miocene Sand and $K_h \in [0.14\,\text{m/d};\, 0.23\,\text{m/d}]$ for Lower Miocene Clay. The vertical anisotropy factor ($K_h/K_v$) was estimated to 25 and 85 for sand and clay units, respectively.

30  The steady-state MODFLOW flow solution (calibration results summarized in Figure 3; Meyer et al. (2018a) also contains an identifiability and uncertainty analysis of the estimated parameters as well as an evaluation and discussion of the non-uniqueness of the flow model.) forms the basis for the advective transport simulation using MODPATH.

**Table 1.** Sampling wells, uncorrected and corrected groundwater ages. Gray shade indicates samples used for calibration. Note that lower numbers of the wells indicate deeper locations (m b.s. = meter below ground surface, std = standard deviation, pMC = percent Modern Carbon).

| well | DGU no. | filter screen depth [m b.s.] | aquifer geology | measured $^{14}$C [pMC] | uncorrected $^{14}$C [years] | $\Delta^{13}C_m$ [‰VDPD] | age corrected for dissolution and diffusion (std)[years] |
|------|---------|------------------------------|-----------------|--------------------------|------------------------------|---------------------------|----------------------------------------------------------|
| A1 | 166.761-1 | 246-252 | Buried valley | 46.44 | 6161 | -13.2 | 344 (59) |
| A2 | 166.761-2 | 204-210 | Buried valley | 49.95 | 5576 | -13 | 108 (19) |
| B1 | 166.762-1 | 160-166 | Buried valley | 49.84 | 5593 | -13.9 | 293(50) |
| B2 | 166.762-2 | 102-108 | Buried valley | 51.9 | 5268 | -13.2 | 46 (8) |
| C1 | 167.1545-1 | 306-312 | Buried valley | 0.48 | 42889 | -5.9 | 10429 (1789) |
| C2 | 167.1545-2 | 273-276 | Buried valley | 1.03 | 36755 | -7.7 | 9097 (1569) |
| C3 | 167.1545-3 | 215-218 | Buried valley | 0.16 | 51714 | -11 | 15038 (2593) |
| C4 | 167.1545-4 | 142-149 | Buried valley | 33.84 | 8703 | -13.2 | 1191 (205) |
| C5 | 167.1545-5 | 116-123 | Buried valley | 43.18 | 6746 | -13.1 | 518 (89) |
| D1 | 159.1335-1 | 290-295 | Miocene | 1.8 | 32271 | -7.9 | 7671 (1323) |
| D2 | 159.1335-2 | 277-282 | Miocene | 1.35 | 34582 | -10.6 | 9229 (1591) |
| E1 | 159.1444-1 | 194-200 | Buried valley | 31.34 | 9320 | -12 | 1141 (197) |
| E3 | 159.1444-3 | 81-87 | Buried valley | 40.29 | 7302 | -12.8 | 642 (111) |
| F1 | 168.1378-1 | 372-378 | Miocene | 46.12 | 6216 | -12.3 | 173 (30) |
| F2 | 168.1378-2 | 341-345 | Miocene | 2.85 | 28580 | -13.3 | 7836 (1351) |
| F3 | 168.1378-3 | 208-214 | Miocene | 25.73 | 10904 | -12.6 | 1800 (310) |
| G1 | 168.1546-1 | 110-120 | Miocene | 42.57 | 6860 | -12.3 | 388 (67) |
| G2 | 168.1546-2 | 74-84 | Pleistocene/ Miocene | 45.33 | 6355 | -12 | 153 (26) |

**3.3 Advective transport model**

Advective transport simulation was performed using MODPATH (Pollock, 2012) in particle back-tracking mode. Hereby, the travel time of a particle (t), released in a cell, is calculated based on the MODFLOW cell-by-cell flow rates (q). The advective travel time (t) along the travel paths in 3D ($\underline{x}$) is calculated as

$$\quad t(\underline{x}) = \int_{\underline{x_0}}^{\underline{x}} \frac{\mathbf{n}_e(\underline{x})}{\mathbf{q}(\underline{x})} d\underline{x} \tag{5}$$

[Figure]

**Figure 3.** Calibration results of steady-state groundwater flow model that forms the basis for the advective transport model (modified after Meyer et al. (2018a). Left: simulated versus observed hydraulic head; right: simulated versus observed stream discharge. ME=mean error, RMS=root mean square.

In addition to the input data required by MODFLOW to generate the flow solution, MODPATH requires a value for effective porosity ($n_e$) to calculate the seepage velocity.

The groundwater age can be seen as the backward integration of travel times along the travel path back to its recharge location. Hence, the simulated groundwater age is a function of the ratio of flux to effective porosity and the travel distance. In this study,

5    the total flux is controlled by prescribed recharge and heterogeneous distribution of hydrogeological parameters (e.g. hydraulic conductivity, porosity).

In order to ensure stability (Konikow et al., 2008), 1000 particles were distributed evenly in the cell of the well screen and their average simulated particle age was compared with apparent groundwater ages (derived from equation 3).

The corrected $^{14}C$ ages were used as targets in the objective function (see below) of the simulated average travel time during

10    calibration. ~~According to Sanford (2011), neglecting hydrodynamic dispersion in advective transport simulations on a regional scale is a reasonable approach when old-age tracers, such as $^{14}C$, are used as dispersion might not be crucial for these tracers. On the other hand, diffusion into stagnant zones can create a significant loss in old-age tracer concentration which was taken into account by correcting the $^{14}C$ (paragraph 3.1.1) before calibration.~~ The particle-based approach used in this study computes the kinematic age at a point. With 1000 particles released in each cell with a screen, we essentially get an age distribution of

15    kinematic ages by perturbing the measurement location within the cell reflecting the mixing of waters from different origins. The $^{14}C$ ages have also been diffusion-corrected (paragraph 3.1.1) so that dilution or mixing due to loss of $^{14}C$ into the stagnant zones have been accounted for.

[revised manuscript text omitted]

assumed that mixing at scales larger than 200 m is accounted for by the geological model. Therefore, the dispersivity should only describe the heterogeneity at flow scale of several hundred of meters which justifies the use of a relatively small $\alpha_L$.  In accordance with Gelhar et al. (1992)  flow scales of hundreds of meters result in $\alpha_L$  of magnitudes in the range of a few meters, which is also in line with studies in the Dutch polder system where dispersivity values of 2 m were applied in similar sized models (e.g. Oude Essink et al., 2010; Pauw et al., 2012). On the grid scale of 200 m to 400 m and with the standard difference solver for the advection-dispersion equation a substantial numerical dispersion is expected. Since there is no sensitivity for lower $\alpha_L$ (numerical dispersion dominates at this scale),  physical dispersivity was set to zero m in the following simulations of direct age. This does not imply that physical dispersion does not exist, only that physical dispersion is accounted for by numerical dispersion.

The directly simulated mean age distribution on a regional scale (Figure 10) shows a general age evolution from young water in the recharge area in the east towards older water in the west (Figure 10 b, e, f). Young water also enters the system through the coastal boundary in the west (Figure 10 b, e, f). The age distribution is strongly affected by the heterogeneity in flow and transport through the aquifers geology and is therefore in good agreement with the interpretation of the flow system by Meyer

[Figure]

**Figure 10.** Directly simulated mean ages and velocity vectors presented at: a) horizontal section at layer 2, also showing river network; b) horizontal section at a depth of 100 m a.s.l. (buried valleys indicated with dotted lines); c) horizontal section at the top of the Maade formation; d) extent and thickness of the Maade formation; e) cross-sections A-B and f) C-D; Pleistocene-Miocene boundary indicated with dashed lines (buried valleys), 100 year lines; g) and h) geological cross-section and i) horizontal geological section, main geological units indicated (a detailed geological description is given in Meyer et al. 2018a). Notice that the color scheme in a) is different in order to better resolve younger ages close to the surface.

et al. (2018a). Two main aquifers are present on a regional scale: a shallow Pleistocene sand aquifer and a deep Miocene sand aquifer, separated by the Maade formation and locally connected through buried valleys (conceptual model in Figure 2, Figure 10 g, h). The regional mean age distribution also reflects this system. Younger waters dominate the shallow Pleistocene aquifers (Figure 10 a, e, f), where the flow regime can be described as mostly local and intermediate (cf. Tóth, 1963). The separating Maade formation with its increasing thickness towards the west (Figure 10 d) acts as a stagnant zone where groundwater age increases (Figure 10 c). The underlying Miocene sand shows the mean age evolution from young water in the recharge areas in the east to older water towards the discharge zones in the west (Figure 10 b, e, f). Here the flow regime is dominated by regional flow (cf. Tóth 1963). Special features are the buried valleys where downward flow of young waters, upwelling of old waters and mixing occurs (Figure 10 e, f, g, h). At the coastal boundary in the west young water enters the system and due to the density-corrected head boundary a wedge is formed with young waters in the wedge and old water accumulating in the transition zone (Figure 10 e, f). The two cross-sections e ) and f) (Figure 10) differ in their geological connection to the sea-boundary (compare geological sections g) and h) in Figure 10). In e) a buried valley connects the inland aquifer with the sea and here younger waters reach further inland due the relatively higher hydraulic conductivity and the inland head gradient as a result of the drainage system. Moreover, buried valleys constitute locations where the deep aquifer system, bearing old waters, connects with the shallow one and here upwelling of older waters occurs due to the higher heads in the deep semi-confined (by the Maade aquitard) Miocene aquifer. In cross-section f) where the buried valley occurs further inland, the young ocean water penetrates the higher permeable Miocene aquifer but is impeded in the low permeable sections and hence does not reach as far inland. Another feature is the human land use change including an extensive drainage network with drain elevations below the sea level in the marsh area. There, old groundwater is forced upward, partly through buried valleys, before it could discharge into the sea.

**4.4.1 Direct simulated mean age distribution in geological units**

The steady state distribution of direct simulated mean groundwater age was reached after $\approx 26000$ years. Over this time span the system has been exposed to transient stresses from human activity and climatic changes (glacial cover, sea level, ect.). Therefore, the steady-state assumption is a notable simplification, which is further discussed in section 5.1.

In Figure 11 the normalized direct age distributions are shown for a) the whole model, b) the Pleistocene aquifer, c) the Maade clay formation that acts as an aquitard, and d) the Miocene sand aquifer (compare the geological setting with conceptual model in Figure 2). The directly simulated mean groundwater ages for the whole model, the Pleistocene sand, the Maade formation and the Miocene sand were determined by a moment analysis (Levenspiel and Sater 1966) as 2574 years, 1009 years, 3883 years, and 2087 years, respectively. The shape of the age distribution in these units varies significantly. The Pleistocene sand shows a unimodal distribution with one peak at $\approx 100$ years and a tail (Figure 11b). The age distribution is governed by recharge of young water and discharge through rivers and drains, which are fed by the upwelling older groundwater (Figure 10a). The age distribution in the Maade formation is multi-modal with five peaks at about 600 years, 1400 years, 3900 years, 6500 years and 7600 years (Figure 11c). Comparison of Figures 10c and 10d reveals a positive relation between age and thickness of the Maade formation. The age distribution in the underlying Miocene sand has one peak at 200 years followed by a plateau

[Figure]

**Figure 11.** Frequency distributions (bin size = 100 years) of directly simulated groundwater ages in a) the whole model, b) the shallow Pleistocene aquifer, c) the separating Miocene clay (Maade formation) and d) the deep Miocene aquifer.

between 1600 years and 3100 years and a small peak at 7800 years (Figure 11d). This distribution is controlled by the overlying and separating Maade formation in the west and the interlayering with Miocene clay.

**4.4.2 Advective and directly simulated ages**

The comparison of the advective ages with the direct simulated ages at the sampling well locations shows a good match for advective ages with a small variance and worsens when the variance increases (Figure 12). Older ages are generally associated with larger variances. Where the mismatch between advective and direct ages is large, the direct simulated mean ages are

consistently lower than mean ages derived from particle back tracking (see discussion below) because of diffusion into clay units. However, most of them lie within one standard deviation; but please observe that the standard deviation spans several thousands of years at some locations, where particle travel time distributions show a multi-modal shape.

[Figure]

**Figure 12.** Mean advective age (MODPATH (MP) particle backtracking) compared to directly simulated mean groundwater age at sampling well locations; error bars on advective age represent 1 standard deviation.

**4.4.3 Capture zones: effect of porosity**

5   Figure 11 shows the capture zones at the Abild well for 1500 years and 2000 years and for the virtual well (AW) for  1000 years, 2000 years and 3000 years for a constant effective porosity of 0.3 (solid line) and the calibrated distributed effective porosities model (dashed line), respectively. The capture zones of the two models vary both in extent and shape. The areas of the capture zone differ by up to 50%. Interestingly, it is not always the same effective porosity model that has the smaller capture zone, but it changes due to the heterogeneity in the geological model and the assigned effective porosities.

[revised manuscript text omitted]

As mentioned in the introduction, the apparent age (or radiometric age) is not equal to the mean particle-based kinematic age.

10 This introduces additional, but unknown uncertain. Ideally, one could develop an advection-dispersion equation for the second moment and solve for the variance of ages (Varni and Carrera, 1998) and use that together with the directly simulate mean age (or first moment) to establish a relation between radiometric and mean ages. This has not been pursued as we believe the benefits from this would be masked by uncertainty in age dating $^{14}C$ (i.e. uncertainty on analyses, and corrections for effects of geochemical and physical processes).

15 Finally, the calibration of effective porosity using an advective transport model relies on a calibrated 3D flow solution that already bears uncertainties with respect to structure and parameters, as addressed by Meyer et al. (2018a). The number and position of the released particles contribute to the uncertainty especially in heterogeneous systems as pointed out by Konikow et al. (2008) and Varni and Carrera (1998). The use of a high number of particles – here 1000 particles were distributed in one cell – generally reduces the uncertainty and enhances stability of the solution. The arithmetic mean of the 1000 released

20 particles evenly distributed in the sampling cells resulted in estimates of effective porosities in the range of 0.13 to 0.45 for sand and 0.043 to 0.1 for clay units, which is significantly different to porosities of 0.25 or 0.30 that are often used in porous media (e.g. Sonnenborg et al., 2016). The reliability of the estimated effective porosities was assessed through the identifiability that depends on the observation density (see section 4.2) and is high for four out of the seven estimated porosities.

**5.1.3 Commensurability**

25 The comparison of groundwater ages, estimated from tracer concentration in a water sample, and simulated groundwater ages, either derived by particle tracking or direct age modelling, bears the problem of commensurability, the comparison of a point measurement relative to the modelling scale. The water sample represents the age distribution in the direct surrounding of the well screen which only makes up a few percent of the water in one model cell.

The differences between mean advective ages and directly simulated mean ages as described in section 4.4 can be re-

30 lated to the simulation methods. While ~~the direct age corresponds to the flux-averaged mean, the particle tracking age is resident-averaged (Varni and Carrera, 1998). Hence, the age distribution of the 1000 simulated particles, especially when it is broad and multi-modal, shifts the mean age towards older ages. By using the harmonic mean of travel times of particles back-tracked from one cell (Konikow et al., 2008) more weight is given to younger ages which would more closely correspond to a flux-weighted mean. This approach improves the comparison (Figure 12; red stars), especially at wells, where the variances~~

are large. Nonetheless, this approach is empirical and do generally not guarantee a better result. Hence, there are still some mismatches that particle tracking neglects dispersion, but allows simulating an age distribution in a cell (by perturbing the measurement location so to speak), direct age modelling allows to account for dispersion/diffusion, resulting in only the mean age at a cell. The mismatches between advective and direct age 
[revised manuscript text omitted]

Campana, M.E., Simpson, E.S., 1984. Groundwater residence times and recharge rates using a discrete-state compartment model and 14C data. J. Hydrol. 72, 171–185. https://doi.org/10.1016/0022-1694(84)90190-2

Castro, M.C., Goblet, P., 2005. Calculation of ground water ages-a comparative analysis. Groundwater 43, 368-380. https://doi.org/10.1111/j.1745-6584.2005.0046.x

Castro, M.C., Goblet, P., 2003. Calibration of regional groundwater flow models: Working toward a better understanding of site-specific systems. Water Resour. Res. 39, 1172. https://doi.org/10.1029/2002WR001653

Cook, P.G., Herczeg, A.L., 2000. Environmental tracers in subsurface hydrology. Springer Science and Business. https://doi.org/10.1017/CBO9781107415324.004

Cornaton, F.J., 2012. Transient water age distributions in environmental flow systems: The time-marching Laplace transform solution technique. Water Resour. Res. 48, 1–17. https://doi.org/10.1029/2011WR010606

Delsman, J.R., Hu-a-ng, K.R.M., Vos, P.C., de Louw, P.G.B., Oude Essink, G.H.P., Stuyfzand, P.J., Bierkens, M.F.P., 2014. Paleo-modeling of coastal saltwater intrusion during the Holocene: an application to the Netherlands. Hydrol. Earth Syst. Sci. 18, 3891–3905. https://doi.org/10.5194/hess-18-3891-2014

de Dreuzy, J.-R., Ginn, T.R., 2016. Residence times in subsurface hydrological systems, introduction to the Special Issue. J. Hydrol. 543, 1–6. https://doi.org/10.1016/j.jhydrol.2016.11.046

[revised manuscript text omitted]

30    Maloszewski, P., Zuber, A., 1996. Lumped parameter modles for the interpretation of environmental tracer data, in: Manual on Mathematical Models in Isotope Hydrology, IAEA-TECDOC 910. pp. 9–50.

Manning, A.H., Solomon, D.K., Thiros, S.A., 2005.  3H/ 3He Age Data in Assessing the Susceptibility of Wells to Contamination. Ground Water 43, 353-367. https://doi.org/10.1111/ j.1745-6584.2005.0028.x

McCallum, J.L., Cook, P.G., Simmons, C.T., 2015. Limitations of the Use of Environmental Tracers to Infer Groundwater Age. Groundwater
35    53, 56-70. https://doi.org/10.1111/gwat. 12237

McCallum, J.L., Cook, P.G., Simmons, C.T., Werner, A.D., 2014. Bias of Apparent Tracer Ages in Heterogeneous Environments. Groundwater 52, 239-250. https://doi.org/10.1111/ gwat.12052

McMahon, P.B., Carney, C.P., Poeter, E.P., Peterson, S.M., 2010. Use of geochemical, isotopic, and age tracer data to develop models of groundwater flow for the purpose of water management, northern High Plains aquifer, USA. Appl. Geochemistry 25, 910-922. https://doi.org/10.1016/j.apgeochem.2010.04.001

Meyer, R., Engesgaard, P., Høyer, A.-S., Jørgensen, F., Vignoli, G., Sonnenborg, T.O., 2018a. Regional flow in a complex coastal aquifer system: combining voxel geological modelling with regularized calibration. J. Hydrol. 562, 544–563. https://doi.org/10.1016/j.jhydrol.2018.05.020

Meyer, R., Engesgaard, P., Sonnenborg, T.O., 2018b. Origin and dynamics of saltwater intrusions in regional aquifers; combining 3D saltwater modelling with geophysical and geochemical data. Water Resour. Res., submitted.

Meyer, R., 2018c. Large scale hydrogeological modelling of a low-lying complex coastal aquifer system. University of Copenhagen. 184p.

Molson, J.W., Frind, E.O., 2012. On the use of mean groundwater age, life expectancy and capture probability for defining aquifer vulnerability and time-of-travel zones for source water protection. J. Contam. Hydrol. 127, 76-87. https://doi.org/10.1016/j.jconhyd.2011.06.001

Morgan, L.K., Werner, A.D., Simmons, C.T., 2012. On the interpretation of coastal aquifer water level trends and water balances?: A precautionary note. J. Hydrol. 470-471, 280-288. https://doi.org/10.1016/j.jhydrol.2012.09.001

Oude Essink, G.H.P., Van Baaren, E.S., De Louw, P.G.B., 2010. Effects of climate change on coastal groundwater systems: A modeling study in the Netherlands. Water Resour. Res. 46, 1-16. https://doi.org/10.1029/2009WR008719

Park, J., Bethke, C.M., Torgersen, T., Johnson, T.M., 2002. Transport modeling applied to the interpretation of groundwater 36 Cl age. Water Resour. Res. 38, 1-15. https://doi.org/10.10 29/2001WR000399

Partington, D., Brunner, P., Simmons, C.T., Therrien, R., Werner, A.D., Dandy, G.C., Maier, H.R., 2011. A hydraulic mixing-cell method to quantify the groundwater component of streamflow within spatially distributed fully integrated surface water-groundwater flow models. Environ. Model. Softw. 26, 886–898. https://doi.org/10.1016/j.envsoft.2011.02.007

Pauw, P., De Louw, P.G.B., Oude Essink, G.H.P., 2012. Groundwater salinisation in the Wadden Sea area of the Netherlands: Quantifying the effects of climate change, sea-level rise and anthropogenic interferences. Geol. en Mijnbouw/Netherlands J. Geosci. 91, 373-383. https://doi.org/10.1017/S0016774600000500

[revised manuscript text omitted]

Woolfenden, L.R., Ginn, T.R., 2009. Modeled ground water age distributions. Ground Water 47, 547–557. https://doi.org/10.1111/j.1745-6584.2008.00550.x

---

## Author Response (AR1)

Dear Editor,

we thank you for your helpful comments. You find our responses (AC) in **bold, changes in quotation marks (" ")** and your comments (EC) in *italic*.

Kind regards,

Rena Meyer, on behalf of the authors

*EC#1*

*Comments to the Author:*
*Thank you for your thoughtful and fairly comprehensive responses to reviewers' comments. I believe your revised paper is much improved and will be of significant interest to the hydrologic community. In reading all the reviews, your responses and the revised manuscript, however, I have two concerns that I believe you can address fairly easily. (1) The statement on p. 19 line 9: "This does not imply that physical dispersion does not exist, only that physical dispersion is accounted for by numerical dispersion." is problematic because you do not even assert that the magnitude of numerical dispersion might similar to that of the sub-grid scale (200 m) dispersion. It sounds like you hope that the numerical dispersion will mimic the real thing, but you do not really know. Unless you have some evidence that the numerical dispersion might fortuitously mimic the real thing, I suggest you remove this sentence and discussion of sub-200-m scale dispersion, and rely on the argument that your resolution of 200-m scale heterogeneity accounts for the dominant dispersion on the scale of this problem and leave it at that. You could cite Weissmann et al. (2002) on this, as well as LaBolle & Fogg (2001) which is cited by Weissmann. The obvious fix would have been to use the more accurate MOC or TVD solution schemes in MT3DMS, and apply actual grid-scale dispersivities, rather than hoping numerical dispersion will cover it for you. If those methods created some other problems, such as excessive execution times, making them impractical, you should say so in the methods.*

**AC#1:**

**We agree and removed the sentence and instead rely on the argument that on our modelling scale dispersion is dominated by facies-scale heterogeneities that are accounted for in the geological model in a scale of 200m to 400m. We add two sentences on page 19:**

**"Similar to Weissmann et al. (2002) and LaBolle and Fogg (2001) the simulations showed little sensitivity to local scale dispersivity because at the modelling scale of tens of kilometers, dispersion is dominated by facies-scale heterogeneity which is captured by the detailed, highly resolved geological model."**

**"Choosing the TVD or MOC solver scheme for the advection-dispersion equation would have been more accurate in terms of less numerical dispersion, but would have required excessive running times which made it impractical to use in this study."**

*EC#2:*

*(2) As suggested by reviewer 2, the low clay porosities that you estimated will look quite implausible to most reviewers. A true clay will indeed always have higher porosity than a sand, unless perhaps the clay has been heavily lithified, say by compaction due to glacial loading, in which case they may be more like claystone formations than clay. Another possibility is that the clays have heterogeneities within, possibly including fractures, that provide preferential flow paths that would increase the apparent effective porosity. So if it really is plausible for the clay effective porosities to be that low, you should explain it through geologic arguments concerning the actual nature of those clays. My first thought, however, was that the calibration was forced to dramatically lower the clay porosity to compensate for too-high K values from your flow model calibration. The latter is also quite plausible because calibrating aquitard K values in a flow model is tough unless you have really good 3D head data. At this point, I just suggest you give one or two geologically plausible explanations for how the clay porosity could be so low.*

**AC#2: As we replied to reviewer 2, we calibrated effective porosities, defined as the pore space which allows the fluid to travel through. The effective porosity can be smaller for clay than for sand (e.g. Hölting and Coldewey, 2013, page 13 Fig. 4) because the pores might be less connected and the water adhesive to the clay minerals. However, we agree that the small effective porosity of the Miocene clay of deep marine origin might be due to compaction as a result of glacier load during several glaciations.**

**Moreover, we cannot rule out that there might be some compensation of the porosity values due to uncertainties in k values because we treat the flow and the advective transport calibration independently.**

**We add the argument to page 14:**

**"The relatively small effective porosities for clay units might be due to compaction as a result of glacial loading in the course of several glacial periods during the Pleistocene."**

**References**

Hölting, B., Coldewey, W.G., 2013. Hydrogeologie, 8th ed. Springer-Verlag, Berin, Heidelberg. https://doi.org/10.1007/978-3-8274-2354-2

LaBolle, E.M., Fogg, G.E., 2001. Role of Molecular Diffusion in Contaminant Migration and Recovery in an Alluvial Aquifer System. Transp. Porous Media 42, 155–179. https://doi.org/10.1023/A:1006772716244

**Reply to Reviewer 1**

Dear Reviewer 1,

We very much appreciate your thoroughly review. You find our responses (AC) in **bold, changes in quotation marks (" ")** and your comments (RC) in *italic*.

Kind regards,

Rena Meyer, on behalf of the authors

Hydrol. Earth Syst. Sci. Discuss., https://doi.org/10.5194/hess-2018-99-RC1, 2018
*The paper details a significant modeling effort demonstrating the importance of carbon-14 dating in the calibration of spatially-distributed porosity. The study utilizes a previously calibrated 3D groundwater flow model of the site and selects 11 of 18 carbon-14 data as targets. I have two major concerns and several other concerns about the implementation of the inverse method and the conceptualization of the apparent ages. The latter are detailed in the specific comments and the former are:*

*1. the model assumes the conductivity field inherited from the (unpublished, at the time of this review) Meyer*

*et al. (2018a, and b which is in preparation); and 2. the data are prefiltered (e.g., eliminated) based on their coherence with the inherited model prior to the analyses. While I highly respect the authors' work in the field and I believe this work has a substantive contribution in the rarely touched world of porosity estimation, I think there are important elements that require consideration and careful address in the discussion. Details of my concerns follow.*

*RC# 1:*

*The very significant reliance on the unpublished groundwater flow model, and its fixed hydraulic conductivities, raises concerns about the current study. The current study seeks to identify porosities of 7 geological units by fitting them so that the mean ("direct") ages match the apparent ages from carbon-14 corrected for dissolution and diffusion; however, there is no allowance for departures from the originally calibrated conductivities (from the unpublished Meyer et al. 2018a). Thus the porosities are treated as if they are independent of the hydraulic conductivities. This is not conventional and disagrees with current understanding of the properties of natural porous media, and needs to be addressed by the authors.*

**AC#1:**

**The groundwater flow model that forms the basis for this study is now published in Journal of Hydrology as "Meyer et al. 2018: Regional flow in a complex coastal aquifer system: combining voxel geological modelling with regularized calibration", DOI: 10.1016/j.jhydrol.2018.05.020."**

**The effective porosities of the 7 geological units were calibrated using an advective particle tracking model (MODPATH) in the way that the mean average particle tracking time, based on 1000 particles released in each of the 11 cells where C-14 observations were available, (not the mean "direct age"), match apparent ages from C-14. These estimated effective porosities were subsequently used in a "direct age" simulation to analyze the age distribution in the entire aquifer.**

**We decided to approach the calibration of this large groundwater model in two steps procedure to enhance stability and well-posedness of the inverse problem. Simultaneous estimation of both flow and transport parameters resulted in stability problems where small changes in the weighting resulted in large changes in parameter values (unrealistic parameter estimates). Hence, following e.g. Carrera et al. (2009) the number of parameters were reduced by dividing the estimation problem into two stages (flow and advective transport). Hereby, realistic parameter estimates were obtained and an acceptable match to the targets were found.**

**Moreover, in our setup, the fluxes are pre-determined by boundary conditions such as recharge and constant heads in streams, drains and ocean. As a consequence, changes in hydraulic conductivities would come along with changes in the gradient, but would not necessarily change the fluxes dramatically and hence not influence the advective age (based on particle tracking). On the other hand, changing the effective porosity would have a more direct influence on the age.**

*RC# 2:*

*Multiple aspects of the inversion done in Meyer et al. are important here since that work laid the foundation flow model; for instance, the vertical anisotropy factors assigned from that work are 25 for sand and 85 for clay units, which are quite high, and qualitatively at least would seem to restrict vertical migration of water in a way that would definitely affect age.*

**AC#2:**

**We agree that the flow model is of major importance and by now the article Meyer et al. 2018 is also published and available (see AC #1).**

*RC# 3:*

*A more robust approach would have been to do a wholistic inversion, where the conductivity (and other flow and transport parameters) were calibrated at the same time as the porosity (and other transport parameters, including the dispersivity, set to zero here based on a brief local sensitivity), to the collective head and apparent age data. Why this is not done, and the potential constraints on the resulting two-stage inverse, should be discussed.*

**AC #3: We agree that a holistic inversion would have been desirable. However, given the model size (millions of nodes) and the runtime this is not possible. This is also the reason why we decided to use a step wise approach and only calibrate effective porosity at this stage, based on a calibrated flow field** (Meyer et al., 2018)**, as this can be estimated using a particle model, which runs much faster than the full advective-dispersion model. It was not possible to perform a calibration on the full automated advective-dispersion model which requires several thousands of model runs. Of course our approach has limitations in a way that maybe information that is contained in the age observation about hydraulic conductivities is not fully exploited. Dispersion parameters are not possible to estimate using an advective transport model only. However, we think that our study still shows the benefit of estimating effective porosity instead of applying a uniform literature value.**

*RC #4:*

*There are no error plots from the prior head-inversion of Meyer et al so the success of the calibration of the flow equation is unknown. More importantly for a subsequent inversion for porosity, there is no indication of the uniqueness of that first inversion. Even if that inversion gave good results, it may be non-unique, and it seems that there may be a different set of hydraulic conductivities and porosities which together might fit both the available head and carbon-14 data.*

**AC #4: Error plots and an uncertainty analysis of parameters are now available in the published article by Meyer et al., 2018 (see AC#1).**

*RC #5:*

*The elimination of dispersivity appears not only somewhat arbitrary but also contradictory to the authors' overall argument for the importance of porosity (cf. specific comment on page 8 line 21). It appears they have replaced the modeling of mobile- immobile domain mass transfer in the model with the approximate diffusion-correction applied to the data. This could be justified based on pragmatic grounds but the discussion in this regard is lacking. The alternative to use effective mobile-immobile domain mass transfer seems potentially useful and pragmatic as well but is not discussed.*

**AC #5: We agree with the reviewer's argument that our approach is a simplification with regard to dispersivity and exchange between flow and stagnant zones. We base the calibration of the distributed effective porosity field on a steady state flow field and use a particle tracking scheme. This approach was needed to keep the computational effort down for the calibration (several thousands of runs). Even with our approach we gain still an important insight in the age distribution in a large scale complex multi-layer aquifer system. And it is shown that choosing a simple porosity estimation scheme is still beneficial compared with applying an uniform porosity.**

*RC 6#:*

*Very important is the unsupported elimination of 7 pesky carbon-14 data (cf specific comment on page 8 line 30). The focus only on the data which are consistent with the already partly calibrated model brings the entire study into question.*

**AC #6: See AC#15**

*RC 7#:*

*Why the recently developed methods for full distribution of age (e.g., several articles in J Hydrology, December 2016) are not used is not described; however, this may be attributed to the reliance on single radiometric tracer (carbon-14) concentration measurements, which precludes any inferrence of age distribution.*

**AC #7: Due to the complex nature of our hydrogeological model and the limitations to only one age tracer, as correctly identified by the reviewer. Moreover, our article focuses on the need to include effective porosity into groundwater model calibration which we demonstrate by the use of groundwater ages.**

**"The groundwater science community (de Dreuzy and Ginn, 2016) has a continued interest in the topic of residence time distributions (RTD) in the subsurface."**

**"It would have been optimal to use RTD analysis (de Dreuzy and Ginn, 2016) to compare modelled and inferred groundwater ages in this study. But, due to the rather complex nature of our hydrogeological flow model, the inherent uncertainties associated with inferring an apparent age to $^{14}$C, and the long computer**

**runtimes, we have chosen to use the particle-based kinematic approach of simulating a mixed age at the well screen (or numerical cell with a screen)."**

*Specific comments.*

*RC 8#:*

*page 2 line 4. "Three different apporaches with specific benefits and disadvantages are commonly applied to simulate groundwater age..." The given list of commonly used methods is not complete (there are also the lumped-parameter approach, and the mixing cell model approach), and equally important are the new methods which are generally more robust [solving the actual equation of groundwater age, either by the Laplace method of Cornaton (WRR 2012) or by using reduced dimensions as in Woolfenden and Ginn (Groundwater, 2009)]. The review by Turnadge and Smerdon (JHydrology 2014) provides a more complete listing and assessment.*

**AC #8: We agree with the reviewer that there are more methods to calculate groundwater ages and their distribution. We add a sentence and include other methods.**

**"Turnadge and Smerdon (2014) reviewed different methods for modelling environmental tracers in groundwater including lumped parameter models (e.g. Maloszewski and Zuber, 1996), mixing-cell models (e.g. Campana and Simpson, 1984; Partington et al., 2011) and direct age models (e.g. Cornaton, 2012; Goode, 1996; Woolfenden and Ginn, 2009). Here, we explain three different approaches with specific benefits and disadvantages that are commonly applied to simulate groundwater age in 3D distributed groundwater flow and transport models (Castro and Goblet 2005; Sanford et al. 2017)."**

*RC 9#:*

*page 2 line 12. "A comparison of ages simulated using any of these methods with ages determined from tracer observations, referred to as apparent ages is desireable..." This is true but omits the very important point that "ages determined from tracer observations" are not equal to mean ages, especially as in the present case of decaying environmental tracers (e.g., carbon-14). The rest of this paragraph summarizes part of the way that "apparent ages" are misled by old carbonate dissolution, by diffusion, and by heterogeneity, following McCallum's work; however, it should also point out the fundamental difference between mean ages and radiometric ages described explicitly by equation 16 of Varni and Carerra (WRR 1998), and the general relation between distribution of age and the radiometric age given in Massoudieh and Ginn (WRR 2011).*

**AC #9:Thank you for this comment, we have added text;**

**"It is important here to distinguish between mean and radiometric ages as for example defined by Varni and Carrera (1998). The only way they can be directly compared in reality is if no mixing is taking place, i.e., if the flow field can be regarded as pure piston flow, which will give the kinematic age. " (introduction)**

**"The particle-based approach used in this study computes the kinematic age at a point. With 1000 particles released in each cell with a screen, we essentially get an age distribution of kinematic ages by perturbing the measurement location within the cell reflecting the mixing of waters from different origins. The C-14 ages have also been diffusion-corrected so that dilution or mixing due to loss of C-14 into the stagnant zones have been accounted for." (Methods page 8)**

*RC 10#:*

*page 2 line 238. "Bethke and Johnson (2002) concldued that the groundwater age exchange... is only a function of the volume of stored water." This is misleading because it is valid only for the mean groundwater age, and requires steady-state as detailed in Ginn et al. (Tranpsort in Porous Media, 2009). Also this point is made earlier and more precisely in Varni and Carerra (op. cit., page 3272), who points out that it is actually a result of Haggerty. The overall point by the authors that porosity is important to age modeling is valid.*

**AC #10: We agree and specify more clearly under which assumptions this is valid and include the suggested references:**

**"However, for steady state flow (Ginn et al., 2009) in a layered aquifer system, Bethke and Johnson (2002) concluded that the mean groundwater age exchange between flow and stagnant zones is only a function of the volume of stored water (Harvey and Gorelick, 1995; Varni and Carrera, 1998)."**

*RC 11#:*

*page 3 line 1. "neglecting dispersion effects seemed to be acceptable at large scale" is unsupported for the present application, results of cited Sanford and later Gelhar notwithstanding. See comments below (re: page 8 line 21 and the reliance on Sanford; page 10 lines 14-17 and Figure 8) for more discussion.*

**AC #11: see AC#14 and AC#18**

*RC 12#:*

*page 6 line 27. "Meyer et al. (2018b) simulated ....further details can be found in Meyer et al. (2018a)." Actually they cannot because Meyer et al. (2018a) is in submitted state (page 30 line 28). This is quite important because the present authors have chosen to rely upon the hydraulic conductivity field previously calibrated in that work, and here do not allow the conductivity values to be modified in the inversion using carbon-14 inferred ages (page 8 line 26).*

**AC #12:**

**The study Meyer et al. 2018 is published and available now (see AC#1).**

*RC#13:*

*page 8 line 2. "The resulting head distribution is shown in Figure 1." Figure 1 shows (it seems to me) only the shallow aquifer heads. It is well-known that the quality of an inversion of the flow equation (to determine hydraulic conductivities) depends on a broad spatial distribution of the heads, and it is unclear that such head data were available to Meyer et al. Also, there are no error plots showing the goodness-of-fit of the flow inversion to the measured heads, so it is impossible for the reader to evaluate how good was the flow equation inversion. Also it is impossible for the reader to evaluate the uniqueness of the flow equation inversion, which is commonly very poor.*

**AC#13:**

**More than 1000 head observations from different depths and aquifers were available and the information can be found in the published article Meyer et al. 2018. Calibration performance of the flow model in terms of goodness-of-fit, ME, RMS. Meyer et al. also contains an identifiability and uncertainty analysis of the estimated parameters as well as an evaluation and discussion of the non-uniqueness of the flow model.**

*RC#14:*

*page 8 line 21 "According to Sanford (2011), neglecting hydrodynamic dispersion... on a regional scale is a reasonable approach when old-age tracers, such as carbon-14, are used as dispersion might not be crucial for these tracers." This sentiment is unclear because it suggests that there is something particular to the carbon molecule that frees it from dispersion, which is quite incorrect. It is also directly in opposition with the authors' claim (page 2 line 28ff) that porosity is important for groundwater mean age determination because "groundwater age exchange between flow and stagnant zones is only a function of the volume of stored water."*

**AC#14: In order to prevent any confusion we take the reference to Sanford out. Our intention was not to argue that we do not have physical dispersion in our system, but that we have a relative high resolution of geological heterogeneities resulting in flow scales of few hundreds of meters and hence physical dispersion of few meters (Gelhar). At the same time we expect some numerical dispersion due to the solver we used (standard finite difference) and the grid size. Hence, the numerical dispersion could overrule the physical one. This is why we set the physical dispersion to 0m.**

*RC#15:*

*page 8 line 30ff. The authors removed 7 data from their 18 carbon-14 measurements becaue the values did not match their conceptual model; 6 were deleted because the carbon-14 activities were below 5pmc, and one due to proximity to another sample with different value. The justification given for the first 6 is "it was assumed that*

*the boundary conditions of the flow model ... were not representative for pre-Holocene conditions." This justification is unclear at best; the model is steady state so the initial conditions do not matter, and the boundary conditions are necessarily (by the steady-state assumption) constant. Thus the elimination of the low carbon data is unsupported. The elimination of the 7th datum is only weakly justified, as there appears to be nothing wrong with it other than its troubling value.*

**AC#15: It is right that the model is steady-state. However, the boundary conditions represent modern conditions. The eastern part of area has been affected by the Scandinavian Ice Sheet during the Weichselian. This ice cap probably induced a high hydraulic pressure with dramatic influence of the hydraulic system** (e.g. Piotrowski, 1997) **and the boundary conditions in the East. We believe that the 6 C-14 measurements with C-14 activities below 5pMC might be influenced by these conditions and eventually recharged outside the modern eastern boundary. Therefore we decided to calibrate the model only based on the measurements that were recharged during similar hydraulic conditions as today.**

**The 7[th] data point was excluded because there is an age inversion in the observations which might be a result of local heterogeneity and it would probably not be possible to reproduce this by the model with the current cell size. The age observations are located in neighboring cells, the younger one directly below the old one. This would have caused troubles during the calibration. Therefore, we decided to exclude the data point from the calibration.**

*RC#16:*

*page 9 line 4-6. The weights on the data used in the inversion were all the same. They were based on an average uncertainly of apparent ages of ~102 years, as per "average of the standard deviation of the diffusion correction for the selected 11 samples..." This defeats the purpose of calculating individual standard deviations for individual data in the first place. The individual standard deviations (Table 1, last column) show a range of 8 to 310 years, so individual weights based on these values would have led to significantly different weights. Individualized weighting is rarely possible in groundwater flow model inversion but is often possible in transport inversion, and it seems to me that the authors have unintentionally limited the inversion by assigning equal weights to all apparent age data. The importance and utility of weighting is amply described in the books by John Doherty and Mary Hill, and could have been used to condition the data per their individual certainties; moreover it could have been used to condition - perhaps to good end - the pesky 7 data that were eliminated instead. In fact, the standard deviations of the 6 eliminated data range from 1323 to 2593 years, which would have led to quite significant reduction in the importance of these data as the weights are generally taken as the reciprocals.*

**AC#16: We had several calibration experiments including individual weights. However, the fit to the older ages, having larger standard deviations was worse, while the one to the younger ages not significantly improved. By applying a uniform weight we intentionally gave higher weight to the older ages than to the younger ones. We decided to not include the 6 data points as justified in AC#15.**

*RC#17:*

*page 9 line 27. "mean groundwater age is simulated in analogy to solute transport as an "age mass" (Bethke and Johnson 2008)." This "age mass" requires mathematical and physical definition; as pointed out in Ginn et al (2009, op. cit., section 2.2) the definitions of Goode and of Bethke and Johnson are not clear or consistent. The example of Bethke and Johnson involves an aquifer and an aquiclude with only immobile water, so that diffusion is the only mechanism by which exchange can take place. If it is eliminated, then the argument collapses.*

**AC#17: We are not sure what the reviewer means. Essentially eq. 8 is identical to the one in Goode (1996), which we refer to, or for that matter, Varni and Carrera (1998). We did not eliminate diffusion but physical dispersion (see AC#18).**

*RC#18:*

*page 10 lines 14-17 The numerical experiments to evaluate dispersion effects, described here, with results summarized on page 15 lines 10ff and in Figure 8, are apparently done on one model, that is, on one assignment of hydraulic conductivities and porosities. It is not clear which porosities were used. In any case, this is at best a local parameter sensitivity analysis and it would be more accurate to include the dispersivity values in the inversion. The argument that the 200mx200m grid cell size is sufficiently resolved to allow ignoring dispersion is unconvincing, because there are multiple modeling exercises where the effective dispersivity is proportional to the grid cell size, not zero. Figure 8 does not tell how the errors grew but only the total error - did the errors go biased ? If one were to guess, I would bet they did, because the dispersion would allow mass transfer laterally, causing generally older ages.*

**AC#18: The estimated porosities using the regularized inversion scheme and shown in Table 2 were used. The dispersity values were not possible to include into the inversion scheme as we used an advective particle tracking model (MODPATH) for the automated calibration due to the long run times of the full advective-dispersion model (MT3DMS). To still investigate the effect of physical dispersion we did as correctly mentioned by the reviewer a local sensitivity analysis. As we used the standard finite difference scheme, we expect a some numerical dispersion with our grid size. From figure 8 one can approximate a numerical dispersion in the order of several tens of meters. Our geological modelling approach and the transformation into a hydrogeological mode as detailed explained in Meyer et al 2018, resolves geological heterogeneities on a grid scale which is 200m to 400m and the flow at a similar scale. Hence, we assumed that physical dispersion at regional scale is accounted for by including a detailed description of the geological heterogeneity (≈ 200 m scale). Therefore, only local scale mixing processes needs to be described by the dispersivity concept, as larger scale processes are taken care of by a detailed description of geology. According to Gelhar et al., (1992) physical dispersion would at this flow scale range from one meter to several meters, which is also in accordance with studies in the Dutch polder system where dispersivity values of 2m are applied in similar sized models (e.g. Oude Essink et al., 2010; Pauw et al., 2012). In our system, we**

assume that numerical dispersion is in the same order of magnitude and therefore is sufficient to account for the local scale mixing processes not accounted for by the heterogeneities build into the model.

In the figure below the errors for the individual wells are illustrated as a function of dispersivity. The error is generally constant for dispersivities up to 50 m, while it increase when the dispersivity is increased to 500 m.

We changed the sentences:

"The very detailed voxel geological model resolves heterogeneities at a scale of 200m x 200m. Hence, it is assumed that mixing at scales larger than 200m is accounted for by the geological model. Therefore, the dispersivity should only describe the heterogeneity at a flow scale of several hundred of meters which justifies the use of a relatively small $\alpha_L$. In accordance with Gelhar et al. (1992) flow scales of hundreds of meters result in $\alpha_L$ of magnitudes in the range of a few meters, which is also in line with studies in the Dutch polder system where dispersivity values of 2m were applied in similar sized models (e.g. Oude Essink et al., 2010; Pauw et al., 2012). On the grid scale of 200m to 400m and with the standard difference solver for the advection-dispersion equation a substantial numerical dispersion is expected. Since there is no sensitivity for lower $\alpha_L$ (numerical dispersion dominates at this scale), a physical dispersivity was set to zero m in the following simulations of direct age. This does not imply that physical dispersion does not exist, only that physical dispersion is accounted for by numerical dispersion."

[Figure]

*RC#19:*

*page 11 line 6 "...as porosity does not impact the trajectory of the particle path..." this is true only via the assumption that the porosity and hydraulic conductivity are independent, which is not common.*

**AC#19: As correctly mentioned by the reviewer, our description is valid and limited to our assumption that in the approach we choose the hydraulic conductivity field is constant (see AC#1). To avoid misunderstanding we add these limitations to our description.**

**"Given that the hydraulic conductivity field is unchanged, no differences in the area of the whole capture zone are expected as porosity does not impact the trajectory of the particle path (Hill and Tiedeman 2007) and only affects the travel time."**

*RC#20:*

*page 13 Figure 4a. The plot demonstrates in my view limited improvement for two reasons. First, the 5 older water samples (with carbon-14 corrected ages greater than 500 years) show significantly improved fitting in 3 cases, with one getting worse. Second, the plot is absent of confidence intervals (compare for instance to Figure 11) which could be it seems to me estimated based on the standard deviations of the corrected carbon-14 ages (Table 1), with additional uncertainty based on equation 16 of Varni and Carrera (op. cit.). The recognized uncertainty in the apparent ages should it seems be used to condition the results of Figure 4a.*

**AC#20: The calculation of uncertainty using equation 16 of Varni and Carrera requires the 2$^{nd}$ moment of the direct simulated age distribution which we do not have. In Figure 4 the age is calculated based on particle tracking not on direct age modelling (advection-dispersion equation), hence it is not possible to calculate uncertainty using equation 16 of Varni and Carrera as this requires the 2$^{nd}$ moment of the direct simulated age distribution using the advection-dispersion equation. However, we add the standard deviation derived from particle-based pdf at a well screen and age correction of the measurements to Figure 4a. As mentioned in the text (on page 11) we achieve a significant reduction in both ME and RMS compared to the uniform-porosity field model.**

*RC#21:*

*page 16 line 4ff "Hence, the dispersivity only describes the effect of heterogeneity at the grid scale, 200m. In accordance with Gelhar et al. (1992) this results in (dispersivity) with a magnitude of a few meters." I am unaware that Gelhar suggested this dispersivity value given (only) the size of the grid, please provide the page. Also in the intervening 25 years there has been extensive research and articles on the effective dispersivity for regional groundwater models, and more up to date referencing is called for. Notably, the model (including its effective parameters) at the 200m grid block scale tells only the expected or mean concentration in the grid block, that is, the concentration in the model is treated as a constant on the 200m x 200m x 5m grid block, while*

*the carbon-14 data are collected from sampling wells on much smaller spatial scales - this issues should also be addressed or at least noted.*

**AC#21: We agree with the reviewer that the Gelhar plots refer to the flow scale and not the grid scale. This is actually what we meant. Thanks to the high resolution of the description of geological heterogeneities (cf. (Meyer et al., 2018) we reach flow scales in the order of several hundred meters. According to Gelhar physical dispersivities would be in the order of several meters at this flow scale. To be more precise we changed the wording (compare AC#18).**

**The problem of commensurability, the problem of comparing point measurements with a mean value for a large volume, is added to the discussion.**

**"The comparison of groundwater ages, estimated from tracer concentration in a water sample, and simulated groundwater ages, either derived by particle tracking or direct age modelling, bears the problem of commensurability, the comparison of a point measurement relative to the modelling scale. The water sample represents the age distribution in the direct surrounding of the well screen which only makes up a few percent of the water in one model cell.**

**The differences between mean advective ages and directly simulated mean ages as described in section 4.4 can be related to the simulation methods. While particle tracking neglects dispersion, but allows simulating an age distribution in a cell (by perturbing the measurement location so to speak), direct age modelling allows to account for dispersion/diffusion, resulting in only the mean age at a cell. The mismatches between advective and direct age can be related to the diffusion and dispersion processes (here represented by numerical dispersion as dispersion was set to zero), which are included in the direct age approach, but neglected in simulating advective ages."**

*RC#22:*

*page 17 line 1. "The age distribution is strongly affected by geology and is therefore in good agreement with the interpretation of the flow system by Meyer et al. (2018)." This statement is unclear: the age distribution is always strongly affected by geology. Figure 10 caption "Normalized probability distributions..." These are frequency distributions because there is no randomness in the model or its parameters.*

**AC#22: We agree and specify that "the age distribution is strongly affected by the heterogeneity in flow and transport through the aquifers geology". The description of Figure 10 is changed as suggested to "frequency distribution of…"**

*RC#23:*

*page 21 line 12 (regarding Figure 11) "However, most of them lie within one standard deviation." Seven of the standard deviations here span several thousands of years while the means for all but one are less than 7000 years, so this is not a comforting result.*

**AC#23: In order to not give the impression to the reader that these fits are perfect we follow the reviewers comment and extend the description of the results to "However, most of them lie within one standard deviation; but please observe that the standard deviation spans several thousands of year at some locations, where particle travel time distributions show a multi-modal shape."**

*RC#24:*

*page 23 section 5.1.2.  This discussion clearly identifies the ways that individual particle path history of exposure to different geologic units differentiates the actual true correction of the carbon-14 from the simplified correction done in the paper; however, it still does not tell about the fundamental difference between the apparent age and the mean age (cf. comment on page 2 line 12). That is, even if the correction were perfect, the apparent age would not equal the mean age.*

**AC#24: Thank you. We have extended the discussion to reflect this;**

**"As mentioned in the introduction, the apparent age (or radiometric age) is not equal to the mean particle-based kinematic age. This introduces some extra, but unknown uncertain. Ideally, one could develop an advection-dispersion equation for the 2. Moment and solve for the variance of ages (Varni and Carrera, 1998) and use that together with the directly simulate mean age (or first moment) to establish a relation between radiometric and mean ages. This has not been pursued as we believe the benefits from this would be masked by uncertainty in age dating C-14 (laboratory uncertainty and dilution-diffusion-correction)." (page 24 )**

*RC#25:*

*page 24 line 6.  "While direct age corresponds to the flux-averaged mean, the particle tracking age is resident-averaged (Varni and Carrera, 1998)." I do not see where this statement is given in the cited reference, please clarify if so; furthermore, I do not believe the statement is correct.  The mean age of the model of Goode is an Eulerian quantity, just like a solute resident concentration.  The relation between resident and flux-averaged concentrations is given in a number of papers by Parker and van Genuchten and coworkers (1984) but the governing equations that result are mainly restricted to 1D cases.*

**AC#25: We take this part out from the manuscript as it has no further relevance for the overall study and leads to confusion.**

*RC#26:*

*page 24 line 9. The use of harmonic mean for particle ages is absent of a rational basis other than it seems to fit the data well, and a generic reference to Konikow (2008). The specific manner of averaging the particle ages should be physically-based and independent of how well it fits the data in a particular setting.*

**AC#26: We take this part out from the manuscript as it has no further relevance for the overall study and leads to confusion.**

*RC#4: which equation is this? 2?*

**AC#4: Yes, we add the reference to equation 2.**

*RC#5: make a brief statement about the steady state assumption – over what time period; what is the evidence for steady state?*

**AC#5: The system, close to the coast, is not expected to be in steady-state over a very long period. Changes in sea level over the last thousands of years and human activity (drains and dikes) over the last centuries**

*changed the hydraulic system, especially in the west, close to the sea. While upstream, to the east, where most of our samples were taken, the system was more steady over the last thousands of years. We included a discussion about effects of transient conditions (see also our response to the short comment).*

*RC#6: what physical features do these boundaries correspond to?*

**AC#6: The physical features are delineated along flow lines and watershed boundaries; we add this to the description:**

**"No-flow boundaries were used along flow lines in the north and south and the water divide in the east and at the bottom, where the Palaeogene clay constitutes the base of the aquifer system."**

*RC#7: What about the assessment of the existing model calibration? Could it be mentioned briefly here so the reader knows how good the model is?*

**AC#7: We add the calibration results by Meyer et al. (2018) in a new figure 3 and change the sentence to:**

**"The steady-state MODFLOW flow solution (calibration results summarized in Figure 3) forms the basis for the advective transport simulation using MODPATH."**

[Figure]

**"Figure 3: Calibration results of steady-state groundwater flow model that forms the basis for the advective transport model (modified after Meyer et al. (2018a). Left: simulated versus observed hydraulic head; right: simulated versus observed stream discharge. ME=mean error, RMS=root mean square."**

*RC#8: Table 2 shows results; maybe save those for the result section.*

**AC#8: We move table 2 to the result section (section 4.2 calibration results).**

*RC#9: What is the explanation for the porosities, i.e. are they reasonable given the description of each formation. Why do the clay units have relatively small porosities? Probably the estimated porosity is an effective transport porosity; this should be noted.*

**AC#9: The reviewer is right, we are estimating effective porosities that is the reason why the estimated effective porosities for clay are relatively small. We check throughout the manuscript and specify where it is missing.**

*RC#10: or of structural error in the number and boundaries of zones chosen, boundary conditions, ect. – many more possible causes than unsimulated heterogeneity*

**AC#10: The reviewer is right. We extended the explanation of the mismatches to:**

**"…mismatches can be, e.g., a result of small scale heterogeneity below grid resolution, errors in the model structure or uncertainties of parameter."**

*RC#11: clay typically has a larger porosity than sand*

**AC#11: This is right for the total porosity. We are dealing with effective porosities (see also AC#9)**

*RC#12: does*

**AC#12: Misspelling corrected. "does"**

*RC#13: It seems that well C has several screen segments with short pathlines that should produce short travel times. It's not clear that only some results are excluded.*

**AC#13: We add screen numbers to be more precise on which wells were used for calibration.**

**"As mentioned above, only 14 C observations with an activity higher than 5pMC (Table 1) were used, which excludes results from well screens C1, C2, C3, D1, D2, F1 and F2."**

*RC#14: A little more discussion on how SV are applied and interpreted here.*

**AC#14:**

**SVD operates on the sensitivity matrix, the Jacobian that relates parameters to observations, and divides the parameter space into a solution space and a null-space. Hereby parameters that are informed by the**

observations are put in the solution space while those not informed by observations fall in the null-space. The truncation between these spaces is user-defined and should be at a level where observations do not further constrain parameters. The advantage of using SVD is that the number of estimated parameters is reduced and hence the inverse problem is well-posed. If too many singular values are included, the problem will be still ill-posed. If too few, the model fit might be unnecessarily poor. Singular values are ordered in a decreasing manner, meaning that a singular value of index 1 is more constrained by information contained in the observations than a singular value of index 2 (Anderson et al., 2015). In our study we truncated the SV at index 5 and in Figure 4 b) the identifiability of the parameters based on the SV index 5 is shown. For more details on singular value decomposition refer to, e.g., Anderson et al. (2015), Doherty and Hunt (2009) or Doherty (2015).

*RC#15: does this mean that estimated porosities for clay units are not reliable?*

**AC#15: The reviewer is right, that the estimated effective porosities with a higher identifiability are more reliable, because they are constrained by the observations, compared to those with a small identifiability. However, it does not necessarily mean that the estimates with a low identifiability are unreliable. They are rather more dependent, or constrained, on the regularization and hence on the expert knowledge than by the observations.**

*RC#16: Consider color-coding the well designations on FIgs 5 and 6 and Table 3. This will make it easier to compare the information on each of these.*

**AC#16: We have considered color coding as suggested, but we think it is more confusing. The well screens are all numbered throughout the figures and the tables, which allow comparison easily.**

*RC#17: One problem with this type of plot is that some of the data are always obscured. Consider plotting each subplot on one or more 2D graphs.*

**AC#17: We changed the graph to a 2D normalized frequency distribution based on the former histograms.**

[Figure]

**Figure 7 (before 6): Particle age distributions at sampling wells A-G (see Figure 1 for locations). a) and b) young waters (bin size = 50 years) show a narrow, unimodal distribution; c) old waters (bin size = 500 years) have broader and often multimodal distributions; d) multi-modal age distribution at sample location D1 (bin size = 1000 years), which shows the longest travel times.**

*RC#18: Consider shading as Table 2 to show which samples were used in model calibration.*

**AC#18: We changed the shading as suggested.**

*RC#19: mean [this relation does not hold for the median]*

**AC#19: We specified the description to:**

**"The younger waters (mean age <1000 years) …"**

*RC#20: Consider using horizontal and vertical lines to show your thresholds of 1000 years and 10 and 20km path lengths.*

**AC#20: We have considered this. But if we do so, these lines should be on each subfigure. The axes of the subfigures are chosen to show best the distribution of the data. For some of the subfigures (wells A, B, F, G) these lines would be outside the figure, therefore we choose not to add these lines.**

*RC#21: If you use alpha=0, you could have used particle tracking. This would avoid the complication of numerical dispersion and would allow you to talk about higher moments of the travel time distribution.*

**AC#21: We set alpha = 0 because the physical dispersion which we still have probably in a range of a few meters is overruled by numerical dispersion (see also AC#14 to comments by reviewer 1). We used particle tracking for the calibration at the sampled well location. But in order to get an idea of the age distribution in the entire model (1.2 mio cells) it was not feasible to produce ravel time distributions of 1000 particles in each cell (as we did for the cells where we had tracer samples). This is why we chose the direct mean age simulation to visualize the age structure in the entire aquifer system.**

*RC#22: Be clear this is mean age here and elsewhere in this section. Also, consider use the term travel time instead of age*

**AC#22: We checked the consistency and added 'mean' when it was missing. We considered using the term 'travel times'. To preserve the comparability between particle age and tracer-based apparent ages we chose the term 'mean age' instead of travel time.**

*RC#23: Consider explicitly explaining why section e and f are different at the western boundary.*

**AC#23: We thank for this remark and add a detailed explanation of the two cross-sections and their differences.**

**"The two cross-sections e ) and f) (Figure 10) differ in their geological connection to the sea-boundary (compare geological sections g) and h) Figure 10). In e) a buried valley connects the inland aquifer with the sea and here younger waters reach further inland due the relatively higher hydraulic conductivity and the inland head gradient as a result of the drainage system. Moreover, buried valleys constitute locations where the deep aquifer system, bearing old waters, connects with the shallow one and here upwelling of older water occurs due to the higher heads in the deep semi-confined (by the Maade aquitard) Miocene aquifer. In cross-section f) where the buried valley occurs further inland, the young ocean water penetrates the higher**

**permeable Miocene aquifer but is impeded in the low permeable sections and hence does not reach as far inland."**

*RC#24: Be consistent with color schemes across all figures – that help the reader understand your points easily. Considering a 1000 year line instead of 100 because 1000 years is used in the discussion.*

**AC#24: We chose to have a different color scale on a) in order to better resolve the younger ages close to the surface. In order to prevent misinterpretation we add an explanation to the caption of the figure.**

**"Be aware that the color scheme in a) is different in order to better resolve younger ages close to the surface."**

**We chose the 100 year line because this is approx. the time span over which human activity (e.g. contamination with fertilizers) heavily started and contaminated groundwater might be expected. This is on what we base our interpretation and discussion groundwater quality and quantity issues on (section 5.3.)**

*RC#25: It would be worth noting that regardless of human actions, stresses have not been steady over that period, either climate, sea level, or within earth's crust. If it takes that long to reach equilibrium under steady stress, the system is never in steady state.*

**AC#25: We thank for this remark. The reviewer is right, the system is over this period never in steady state. A similar remark was given in the short comment. We add the sentence here and further discuss this in the discussion.**

**"The steady state distribution of direct simulated mean groundwater age was reached after ~26000 years. Over this time span the system has been exposed to transient stresses from human activity, climatic changes (glacial cover, sea level, ect.). Therefore, the steady-state assumption is a notable simplification, which is further discussed in section 5.1."**

*RC#26: Review the porosity of Maade and how it was determined.*

**AC#26: The porosity of the Maade was estimated as 'Pleistocene clay (Maade formation)' e.g. Table 2 or Figure 4 (the new Figure 5).**

*RC#27: That seems to be older than what the pdf indicates.*

**AC#27: We have checked the mean groundwater ages derived from a moment analysis and the shown pdfs again. They are correct.**

*RC#28: The direct ages are a function of the age mass of the volume of the model cell whereas the advective ages are a function of the well screen position within the cell. You wouldn't necessarily expect them to match.*

**AC#28: We add a section about the commensurability to the discussion. Here we discuss the differences in observed tracer ages, particle-based simulated ages and directly simulated ages.**

**"The comparison of groundwater ages, estimated from tracer concentration in a water sample, and simulated groundwater ages, either derived by particle tracking or direct age modelling, bears the problem of commensurability, the comparison of a point measurement relative to the modelling scale. The water sample represents the age distribution in the direct surrounding of the well screen which only makes up a few percent of the water in one model cell.**

**The differences between mean advective ages and directly simulated mean ages as described in section 4.4 can be related to the simulation methods. While particle tracking neglects dispersion, but allows simulating an age distribution in a cell (by perturbing the measurement location so to speak), direct age modelling allows to account for dispersion/diffusion, resulting in only the mean age at a cell. The mismatches between advective and direct age can be related to the diffusion and dispersion processes (here represented by numerical dispersion as dispersivity was set to zero), which are included in the direct age approach, but neglected in simulating advective ages."**

*RC#29: the dashed lines are not clear on these maps.*

**AC#29: We enlarged the figure, now the lines are better visible.**

*RC#30: not clear what you mean by 'general behavior of the voxel system.' Maybe this could be reworded, for example, "properties averaged over hydrogeological units".*

**AC#30: We thank the reviewer for the suggestion and changed the sentenced accordingly to:**

**"The geology is highly complex and aquitard thickness and porosity distribution change spatially over the entire region, whereas the correction terms were based on the properties averaged over hydrogeological units."**

*RC#31: Particle tracking can also be used to calculate flux-weighted residence times. The difference is how you choose to weight particles, whether by volume or by flux.*

**AC#31: To prevent confusion, also based on comments by reviewer 1, we decided to take this part out of the manuscript (see also AC#25 and 26 to reviewer 1)**

*RC#32: You can also assign weights to particles based on flux, which would give you a more comparable age to the direct method. You still have the difference that particles placed in a well screen have limited spatial distribution compared to those in a model cell.*

**AC#32: see AC#31.**

**References added:**

**Anderson, M., Woessner, W.W., Hunt, R., 2015. Applied Groundwater Modeling: Simulation of Flow and Advective Transport, 2nd ed. Elsevier. https://doi.org/10.1016/B978-0-08-091638-5.00001-8**

**Doherty, J., 2016. Model-Independent Parameter Estimation II. Watermark Numer. Comput. 217.**

[revised manuscript text omitted]

    The groundwater science community (de Dreuzy and Ginn, 2016) has a continued interest in the topic of residence time distributions (RTD) in the subsurface. Turnadge and Smerdon (2014) reviewed different methods for modelling environmental tracers in groundwater including lumped parameter models (e.g. Maloszewski and Zuber, 1996), mixing-cell models (e.g. Campana and Simpson, 1984; Partington et al., 2011) and direct age models (e.g. Cornaton, 2012; Goode, 1996; Woolfenden

10   and Ginn, 2009). Here, we focus on three different approaches with specific benefits and disadvantages that are commonly applied to simulate groundwater age in 3D distributed groundwater flow and transport models (Castro and Goblet 2005; Sanford et al. 2017).

[revised manuscript text omitted]

20    water resources model (Henriksen et al. 2003) and included as a specified flux condition. Internal specified boundaries included abstraction wells with a total flux of $26 \times 10^6\,m^3/year$ (averaged over the years 2000-2010, corresponding to 4% of the total recharge), rivers and drains.

Horizontal hydraulic conductivities, one for each hydrogeological unit, two anisotropy factors (Kh/Kv), one for sand and one for clay units, as well as river and drain conductances were calibrated, using a multi-objective regularized inversion scheme

25    (PEST; Doherty, 2016a), using head and mean stream flow observations as targets. The resulting head distribution is shown in Figure 1. Horizontal hydraulic conductivities were estimated in a range of $K_h \in [1\,\mathrm{m/d};\ 83\,\mathrm{m/d}]$ for Pleistocene sand units, $K_h \in [0.028\,\mathrm{m/d};\ 0.19\,\mathrm{m/d}]$ for Pleistocene clay units, $K_h \in [0.008\,\mathrm{m/d};\ 0.016\,\mathrm{m/d}]$ for the Maade formation, $K_h \in [16\,\mathrm{m/d};\ 46\,\mathrm{m/d}]$ for Miocene Sand and $K_h \in [0.14\,\mathrm{m/d};\ 0.23\,\mathrm{m/d}]$ for Lower Miocene Clay. The vertical anisotropy factor ($K_h/K_v$) was estimated to 25 and 85 for sand and clay units, respectively.

30    The steady-state MODFLOW flow solution (calibration results summarized in Figure 3; Meyer et al. (2018a) also contains an identifiability and uncertainty analysis of the estimated parameters as well as an evaluation and discussion of the non-uniqueness of the flow model.) forms the basis for the advective transport simulation using MODPATH.

**Table 1.** Sampling wells, uncorrected and corrected groundwater ages. Gray shade indicates samples used for calibration. Note that lower numbers of the wells indicate deeper locations (m b.s. = meter below ground surface, std = standard deviation, pMC = percent Modern Carbon).

| well | DGU no. |  screen depth [m b.s.] | aquifer geology | measured $^{14}$C [pMC] | uncorrected $^{14}$C [years] | $\Delta^{13}C_m$ [‰VDPD] | age corrected for dissolution and diffusion (std)[years] |
|------|---------|------|------|------|------|------|------|
| A1 | 166.761-1 | 246-252 | Buried valley | 46.44 | 6161 | -13.2 | 344 (59) |
| A2 | 166.761-2 | 204-210 | Buried valley | 49.95 | 5576 | -13 | 108 (19) |
| B1 | 166.762-1 | 160-166 | Buried valley | 49.84 | 5593 | -13.9 | 293(50) |
| B2 | 166.762-2 | 102-108 | Buried valley | 51.9 | 5268 | -13.2 | 46 (8) |
| C1 | 167.1545-1 | 306-312 | Buried valley | 0.48 | 42889 | -5.9 | 10429 (1789) |
| C2 | 167.1545-2 | 273-276 | Buried valley | 1.03 | 36755 | -7.7 | 9097 (1569) |
| C3 | 167.1545-3 | 215-218 | Buried valley | 0.16 | 51714 | -11 | 15038 (2593) |
| C4 | 167.1545-4 | 142-149 | Buried valley | 33.84 | 8703 | -13.2 | 1191 (205) |
| C5 | 167.1545-5 | 116-123 | Buried valley | 43.18 | 6746 | -13.1 | 518 (89) |
| D1 | 159.1335-1 | 290-295 | Miocene | 1.8 | 32271 | -7.9 | 7671 (1323) |
| D2 | 159.1335-2 | 277-282 | Miocene | 1.35 | 34582 | -10.6 | 9229 (1591) |
| E1 | 159.1444-1 | 194-200 | Buried valley | 31.34 | 9320 | -12 | 1141 (197) |
| E3 | 159.1444-3 | 81-87 | Buried valley | 40.29 | 7302 | -12.8 | 642 (111) |
| F1 | 168.1378-1 | 372-378 | Miocene | 46.12 | 6216 | -12.3 | 173 (30) |
| F2 | 168.1378-2 | 341-345 | Miocene | 2.85 | 28580 | -13.3 | 7836 (1351) |
| F3 | 168.1378-3 | 208-214 | Miocene | 25.73 | 10904 | -12.6 | 1800 (310) |
| G1 | 168.1546-1 | 110-120 | Miocene | 42.57 | 6860 | -12.3 | 388 (67) |
| G2 | 168.1546-2 | 74-84 | Pleistocene/ Miocene | 45.33 | 6355 | -12 | 153 (26) |

**3.3 Advective transport model**

Advective transport simulation was performed using MODPATH (Pollock, 2012) in particle back-tracking mode. Hereby, the travel time of a particle , released in a cell, is calculated based on the MODFLOW cell-by-cell flow rates (q). The advective travel time (t) along the travel paths in 3D ($\underline{x}$) is calculated as

$$\quad t(\underline{x}) = \int_{\underline{x_0}}^{\underline{x}} \frac{\mathbf{n}_e(\underline{x})}{\mathbf{q}(\underline{x})} d\underline{x} \tag{5}$$

[Figure]

**Figure 3.** Calibration results of steady-state groundwater flow model that forms the basis for the advective transport model (modified after Meyer et al. (2018a). Left: simulated versus observed hydraulic head; right: simulated versus observed stream discharge. ME=mean error, RMS=root mean square.

In addition to the input data required by MODFLOW to generate the flow solution, MODPATH requires a value for effective porosity ($n_e$) to calculate the seepage velocity.

The groundwater age can be seen as the backward integration of travel times along the travel path back to its recharge location. Hence, the simulated groundwater age is a function of the ratio of flux to effective porosity and the travel distance. In this study,

5   the total flux is controlled by prescribed recharge and heterogeneous distribution of hydrogeological parameters (e.g. hydraulic conductivity, porosity).

In order to ensure stability (Konikow et al., 2008), 1000 particles were distributed evenly in the cell of the well screen and their average simulated particle age was compared with apparent groundwater ages (derived from equation 3).

The corrected $^{14}C$ ages were used as targets in the objective function (see below) of the simulated average travel time during

10   calibration. ~~According to Sanford (2011), neglecting hydrodynamic dispersion in advective transport simulations on a regional scale is a reasonable approach when old-age tracers, such as $^{14}C$, are used as dispersion might not be crucial for these tracers. On the other hand, diffusion into stagnant zones can create a significant loss in old-age tracer concentration which was taken into account by correcting the $^{14}C$ (paragraph 3.1.1) before calibration.~~ The particle-based approach used in this study computes the kinematic age at a point. With 1000 particles released in each cell with a screen, we essentially get an age distribution of

15   kinematic ages by perturbing the measurement location within the cell reflecting the mixing of waters from different origins. The $^{14}C$ ages have also been diffusion-corrected (paragraph 3.1.1) so that dilution or mixing due to loss of $^{14}C$ into the stagnant zones have been accounted for.

[revised manuscript text omitted]

Results from the distributed effective porosity model were compared to those from a uniform effective porosity model with

[Figure]

**Figure 5.** Calibration results: a)  'x' show apparent ages simulated with MODPATH (MP) and a porosity of 0.3 as often used in porous media models and  '+' are MP ages simulated based on the 7 calibrated porosities (Table 2); standard deviations based on MP and correction terms (see section 3.1.1) are shown. b) parameter identifiability of effective porosities (warmer colors correspond to singular values (SV) of a lower index, cooler color to SV of higher index) of the different geological formations; the identifiability of the Maade porosity is close to zero.

5   an effective porosity of 0.3 which is a typical textbook value for porous media (Holting and Coldewey, 2013; Anderson et al., 2015) and often used in groundwater modelling studies (e.g. Sonnenborg et al., 2016). The calibrated distributed effective porosity model is able to match all the observations reasonably. This is not the case for the single effective porosity model where especially one sample is poorly simulated with an estimate of more than 5500 years whereas the corresponding observation only reach about 1200 years. The ME and RMS of the calibrated distributed effective porosity model were -2.3 years

10  and 267 years, respectively, which correspond to a reduction in ME of 99% and RMS of 82% compared to the single effective porosity model. Considering the uncertainties involved in estimation of apparent age, see uncertainty estimates in Table 1, column to the right, the match is found acceptable. Comparing the average uncertainty on apparent ages used for calibration of 102 years with the achieved RMS of 267 years indicate that no overfitting occurred and mismatches can be a result of small scale heterogeneity below grid resolution, errors in the model structure or uncertainties of parameters..

**Table 2.** Calibration settings  and results: parameters with initial, preferred and estimated values for effective porosity.

| parameter ($n_e$) | Initial/preferred value | estimated value | % of cells | objective function | |
|---|---|---|---|---|---|
| Pleistocene sand 1 | 0.3 | 0.130 | 24.4 | PHIMLIM | 60 |
| Pleistocene sand 2 | 0.3 | 0.263 | 2.5 | PHIMACCEPT | 100 |
| Pleistocene clay 1 | 0.1 | 0.085 | 11.6 | $\phi_m$ achieved | 74 |
| Pleistocene clay 2 | 0.05 | 0.043 | 4.8 | | |
| Miocene sand | 0.3 | 0.450 | 15.1 | | |
| Miocene clay | 0.1 | 0.102 | 22.8 | | |
| Miocene clay | 0.05 | 0.049 | 18.8 | | |
| (Maade formation) | | | | | |

The estimated effective porosities of the seven hydrogeological units are listed in Table 2. Realistic values are found for all parameters and the values of the sand units are generally higher than those of the clay units. However, the effective porosity estimate of 0.13 for Pleistocene sand 1 is relatively low. This may be explained by the fact that this unit does not represent sand exclusively everywhere. The Pleistocene deposits in the area are highly heterogeneous (Jørgensen et al., 2015) and it is

5    therefore difficult to identify units exclusively composed of sand, partly due to the difficulties in using AEM data to guide the distinction between sand and clay at a relatively small scale. Hence, Pleistocene sand 1 may to some extent represent a mixture of sand and clay. The relatively small effective porosities for clay units might be due to compaction as a result of glacial loading in the course of several glacial periods during the Pleistocene. 
[revised manuscript text omitted]

scale. The very detailed voxel geological model  resolves heterogeneities at a scale of  200 m x 200 m. Hence, it is assumed that mixing at scales larger than 200 m is accounted for by the geological model. Therefore, the dispersivity should only describe the heterogeneity at flow scale of several hundred of meters which justifies the use of a relatively small $\alpha_L$.  In accordance with Gelhar et al. (1992)

5  flow scales of hundreds of meters result in $\alpha_L$  of magnitudes in the range of a few meters, which is also in line with studies in the Dutch polder system where dispersivity values of 2 m were applied in similar sized models (e.g. Oude Essink et al., 2010; Pauw et al., 2012). Similar to Weissmann et al. (2002) and LaBolle and Fogg (2001) the simulations showed little sensitivity to local scale dispersivity because at the modelling scale of tens of kilometers, dispersion is dominated by facies-scale heterogeneity which is captured by the detailed, highly resolved geological model. On the grid scale of 200 m

10 to 400 m and with the standard difference solver for the advection-dispersion equation a substantial numerical dispersion is expected. Choosing the TVD or MOC solver scheme for the advection-dispersion equation would have been more accurate in terms of less numerical dispersion, but would have required excessive running times which made it impractical to use in this study. Since there is no sensitivity for lower $\alpha_L$ (numerical dispersion dominates at this scale),

physical dispersivity was set to zero m in the following simulations of direct age.

The directly simulated mean age distribution on a regional scale (Figure 10) shows a general age evolution from young water in the recharge area in the east towards older water in the west (Figure 10 b, e, f). Young water also enters the system through the coastal boundary in the west (Figure 10 b, e, f). The age distribution is strongly affected by the heterogeneity in flow and transport through the aquifers geology and is therefore in good agreement with the interpretation of the flow system by Meyer et al. (2018a). Two main aquifers are present on a regional scale: a shallow Pleistocene sand aquifer and a deep Miocene sand aquifer, separated by the Maade formation and locally connected through buried valleys (conceptual model in Figure 2, Figure 10 g, h). The regional mean age distribution also reflects this system. Younger waters dominate the shallow Pleistocene aquifers (Figure 10 a, e, f), where the flow regime can be described as mostly local and intermediate (cf. Tóth, 1963). The separating Maade formation with its increasing thickness towards the west (Figure 10 d) acts as a stagnant zone where groundwater age increases (Figure 10 c). The underlying Miocene sand shows the mean age evolution from young water in the recharge areas in the east to older water towards the discharge zones in the west (Figure 10 b, e, f). Here the flow regime is dominated by regional flow (cf. Tóth 1963). Special features are the buried valleys where downward flow of young waters, upwelling of old waters and mixing occurs (Figure 10 e, f, g, h). At the coastal boundary in the west young water enters the system and due to the density-corrected head boundary a wedge is formed with young waters in the wedge and old water accumulating in the transition zone (Figure 10 e, f). The two cross-sections e ) and f) (Figure 10) differ in their geological connection to the sea-boundary (compare geological sections g) and h) in Figure 10). In e) a buried valley connects the inland aquifer with the sea and here younger waters reach further inland due the relatively higher hydraulic conductivity and the inland head gradient as a result of the drainage system. Moreover, buried valleys constitute locations where the deep aquifer system, bearing old waters, connects with the shallow one and here upwelling of older waters occurs due to the higher heads in the deep semi-confined (by the Maade aquitard) Miocene aquifer. In cross-section f) where the buried valley occurs further inland, the young ocean water penetrates the higher permeable Miocene aquifer but is impeded in the low permeable sections and hence does not reach as far inland. Another feature is the human land use change including an extensive drainage network with drain elevations below the sea level in the marsh area. There, old groundwater is forced upward, partly through buried valleys, before it could discharge into the sea.

**4.4.1   Direct simulated mean age distribution in geological units**

The steady state distribution of direct simulated mean groundwater age was reached after ≈ 26000 years. Over this time span the system has been exposed to transient stresses from human activity and climatic changes (glacial cover, sea level, ect.). Therefore, the steady-state assumption is a notable simplification, which is further discussed in section 5.1.
In Figure 11 the normalized direct age distributions are shown for a) the whole model, b) the Pleistocene aquifer, c) the Maade clay formation that acts as an aquitard, and d) the Miocene sand aquifer (compare the geological setting with conceptual model in Figure 2). The directly simulated mean groundwater ages for the whole model, the Pleistocene sand, the Maade formation and the Miocene sand were determined by a moment analysis (Levenspiel and Sater 1966) as 2574 years, 1009 years, 3883 years,

[revised manuscript text omitted]

  The area close to the coast is not only affected by changing sea levels during the past thousands of years, but also by saltwater

15   intrusion. In this study, the density effects on flow were accounted for in a simplified way by using a density-corrected constant head boundary at the coast. Both, sea level changes and density effects, would also have affected the age distribution. The impact on age calculations due to density effects would be largest close to the coast. However, most of the groundwater samples used for age estimations were collected several tens of kilometers inland and are therefore expected to be affected to a minor extent. To quantify the impact of boundary conditions and saltwater intrusion on the particle tracking, the differences of particle

20   travel path lengths for a 200 year period, investigated based on the present model and a preliminary density-driven model (SEAWAT) accounting for non-stationary and density effects (similar to the one presented in Meyer, 2018c) are computed. The relative differences are below 10% (except at location A and B). Also, the uncertainties introduced by simplifying the density boundary effects are likely less important compared to other uncertainties associated, e.g., with estimating the groundwater age by the procedures for correcting $^{14}C$ activities. A solution would, of course, be to use a fully density-driven model such

25   as SEAWAT as in Meyer (2018c) or Delsman (2014). But, the very long computer run times for these kinds of models and the need of several thousands of model runs during calibration made it infeasible to use a variable-density flow model.

**5.1.2 Apparent age as calibration target**

  Uncertainties in the use of $^{14}C$ as a groundwater dating tool and as calibration target arise at different levels. First, sampling of well screens with a length of  6 m ~-10 m would encompass a range of groundwater ages as a result of mixing of

30   groundwater of different ages. Hereby younger waters, corresponding to DIC with a higher $^{14}C$ content, would dominate older ages (Park et al. 2002). The $^{14}C$ content is measured in the DIC of the groundwater. In order to obtain a reliable age estimate, the origin of DIC in groundwater is important. For the different processes that can affect the DIC and change its $^{14}C$ content

(e.g dissolution, precipitation, isotopic exchange) a variety of correction models exists (see overview of correction models in IAEA 2013). For the investigated system, corrections for carbonate dissolution and diffusion were applied, but it cannot be ruled out that also other chemical processes might have changed the $^{14}C$ content over the past thousands of years. The $^{14}C$ correction for diffusion into stagnant zones is sensitive to aquifer porosity, aquitard thickness and diffusion constant. The geology is highly complex and aquitard thickness and porosity distribution change spatially over the entire region, whereas the correction terms were based on the properties averaged over hydrogeological units. Hence, average values of diffusion corrections were applied with parameters varying in ranges realistic for an aquifer system at this scale. However, in reality a groundwater particle would have been exposed to a variety of aquifer/aquitard thicknesses and porosities along its flow path implying smaller or larger diffusion. The correction results show that both carbonate dissolution and diffusion into stagnant zones reduce the apparent groundwater age considerably, both at a similar magnitude as observed by Scharling (2011) and Hinsby et al. (2001).

As mentioned in the introduction, the apparent age (or radiometric age) is not equal to the mean particle-based kinematic age. This introduces additional, but unknown uncertain. Ideally, one could develop an advection-dispersion equation for the second moment and solve for the variance of ages (Varni and Carrera, 1998) and use that together with the directly simulated mean age (or first moment) to establish a relation between radiometric and mean ages. This has not been pursued as we believe the benefits from this would be masked by uncertainty in age dating $^{14}C$ (i.e. uncertainty on analyses, and corrections for effects of geochemical and physical processes).

Finally, the calibration of effective porosity using an advective transport model relies on a calibrated 3D flow solution that already bears uncertainties with respect to structure and parameters, as addressed by Meyer et al. (2018a). The number and position of the released particles contribute to the uncertainty especially in heterogeneous systems as pointed out by Konikow et al. (2008) and Varni and Carrera (1998). The use of a high number of particles – here 1000 particles were distributed in one cell – generally reduces the uncertainty and enhances stability of the solution. The arithmetic mean of the 1000 released particles evenly distributed in the sampling cells resulted in estimates of effective porosities in the range of 0.13 to 0.45 for sand and 0.043 to 0.1 for clay units, which is significantly different to porosities of 0.25 or 0.30 that are often used in porous media (e.g. Sonnenborg et al., 2016). The reliability of the estimated effective porosities was assessed through the identifiability that depends on the observation density (see section 4.2) and is high for four out of the seven estimated porosities.

**5.1.3 Commensurability**

The comparison of groundwater ages, estimated from tracer concentration in a water sample, and simulated groundwater ages, either derived by particle tracking or direct age modelling, bears the problem of commensurability, the comparison of a point measurement relative to the modelling scale. The water sample represents the age distribution in the direct surrounding of the well screen which only makes up a few percent of the water in one model cell.

The differences between mean advective ages and directly simulated mean ages as described in section 4.4 can be related to the simulation methods. While

is broad and multi-modal, shifts the mean age towards older ages. By using the harmonic mean of travel times of particles back-tracked from one cell (Konikow et al., 2008) more weight is given to younger ages which would more closely correspond to a flux-weighted mean. This approach improves the comparison (Figure 12; red stars), especially at wells, where the variances are large. Nonetheless, this approach is empirical and do generally not guarantee a better result. Hence, there are still some

5 mismatches that particle tracking neglects dispersion, but allows simulating an age distribution in a cell (by perturbing the measurement location so to speak), direct age modelling allows to account for dispersion/diffusion, resulting in only the mean age at a cell. The mismatches between advective and direct age 
[revised manuscript text omitted]